# Keep Your Friends Close, and Your Enemies Farther: Distance-aware Voxel-wise Contrastive Learning for Semi-supervised Multi-organ Segmentation

## Abstract

Voxel-wise contrastive learning (VCL) is a prominent approach in semi-supervised medical image segmentation. Based on the initially generated pseudo-labels, VCL pulls voxels with the same pseudo-labels toward their prototypes while pushes those with different labels apart, thereby learns effective representations for the segmentation task. However, in multi-organ segmentation (MoS), the complex anatomical structures of certain organs often lead to many unreliable pseudo-labels. Directly applying VCL can introduce confirmation bias, resulting in poor segmentation performance. A common practice is to first transform these unreliable pseudo-labels into more reliable complementary ones, which represent classes that voxels are least likely to belong to, and then push voxels away from the prototypes of their complementary labels. However, we find that in this approach, if voxels with unreliable pseudo-labels are originally close in feature space, they can end up far apart after being pushed away from their complementary prototypes. This disruption of the semantic relationships among voxels can be detrimental to the MoS task. In this paper, we propose DVCL, a novel distance-aware VCL method for semi-supervised MoS. DVCL is based on the observation that voxels close to each other in the feature space ('neighbors') likely belong to the same semantic category, while distant ones ('outsiders') likely belong to different categories. In DVCL, we first identify neighbors and outsiders for all voxels with unreliable pseudo-labels, and then pull their neighbors into the same clusters while pushing outsiders away. In this way, neighbors of unreliable voxels remain their neighbors and outsiders remain outsiders. This approach helps maintain useful semantic relationships among unreliable voxels while still enjoying the advantages of VCL. We conduct extensive experiments on four datasets to validate the effectiveness. Extensive experiments on four datasets demonstrate the superior performance of DVCL compared to state-of-the-art methods.

## 1 Introduction

Medical image segmentation is a critical task in computer-aided diagnosis (Meng et al., 2022; Zhao et al., 2022). However, considering the large amount of effort and expertise required for labeling data (Greenspan et al., 2016), semi-supervised learning (SSL) is often employed. In SSL, voxel-wise contrastive learning (VCL) (Chaitanya et al., 2023; You et al., 2024; Wu et al., 2024) has proven highly effective. Based on the pseudo-labels generated by the initial model, VCL pulls voxels with the same pseudo-labels toward their prototypes while pushing those with different pseudo-labels away from each other, thereby learning effective representations from unlabeled voxels for the segmentation task.

However, existing VCL methods can encounter difficulty in multi-organ segmentation (MoS). MoS is more challenging than single-organ segmentation due to complex anatomical structures, including large size variations (e.g., the liver is over 200 times larger than the right adrenal gland), different shapes (e.g., the stomach has a more complex shape than the liver despite similar volumes), and overlapping organs (e.g., the pancreas often overlaps with adjacent organs, resulting in unclear

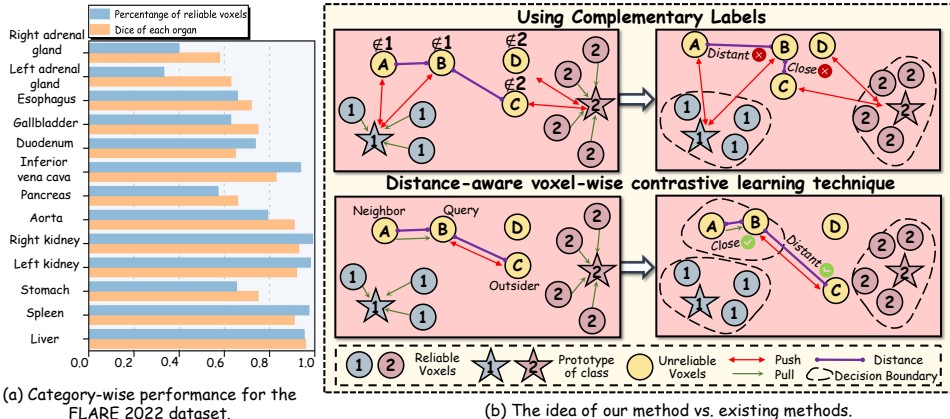

Figure 1: (a) Category-wise performance for the FLARE 2022 dataset. The orange bars represent the Dice similarity coefficient (referred to as Dice) for each organ,and the blue bars denote the percentange of reliable voxels. (b) The idea of our method vs. existing methods. DVCL maintains useful semantic relationships among unreliable voxels in the feature space.

boundaries). Consequently, segmenting these organs becomes considerably more difficult, resulting in many unreliable pseudo-labels, especially at the initial stage. These unreliable pseudo-labels generally lead to confirmation bias in VCL, ultimately lowering the segmentation performance on these organs. Figure 1(a) shows the segmentation performance of different organs on the FLARE 2022 dataset. The blue bars denote the percentage of voxels with reliable pseudo labels for each organ, and the orange bars represent the corresponding performance. We can see that some organs, such as the left and right adrenal glands and the pancreas, have many more unreliable voxels. Correspondingly, the segmentation performance of these organs is lower than others.

To address these challenges in VCL, some methods choose not to use voxels with unreliable pseudo-labels (Su et al., 2024; Chaitanya et al., 2023; Wang et al., 2022a). However, given the amount of unreliable voxels, this approach wastes the valuable information contained in them. To leverage information contained in a large number of unreliable pseudo-labels, a popular practice is to use complementary labels (Wang et al., 2022c; Du et al., 2023; Feng et al., 2024; Deng et al., 2024). The complementary labels of a voxel represent classes that the voxel is least likely to belong to, and therefore can be more reliable than the original pseudo-label. We can then push the voxel away from the prototypes of its complementary labels. An example of using complementary labels is shown in the upper left of Fig. 1(b). The two yellow nodes $A$ and $B$, labeled with $\notin 1$, are the two voxels least likely to belong to class 1 and therefore are pushed away from the prototype of class 1 (shown as the star in the left corner) during VCL. Likewise, voxel $C$ and $D$, labeled with $\notin 2$ are pushed away from the prototype of its complementary class 2. Experiments demonstrate that VCL based on complementary labels helps to learn good voxel representations(Wang et al., 2022c; Du et al., 2023; Feng et al., 2024; Deng et al., 2024).

Despite the successes of this technique, we have identified a hidden drawback. This approach can disrupt the relationships among some unreliable voxels, and these relationships are helpful when learning features. An example is shown in the upper two figures of Fig. 1(b). The left figure shows the feature space of some voxels before using complementary labels. The four voxels with unreliable pseudo-labels are shown as yellow nodes denoted as $A$, $B$, $C$, and $D$, and their complementary labels are $\notin 1$, $\notin 1$, $\notin 2$ and $\notin 2$, respectively. We can see that before using VCL, the two voxels $A$ and $B$ are close to each other and are likely to belong to the same category. On the other hand, voxel $C$ should belong to a different class. However, after being pushed away from prototype 1, voxel $A$ and $B$ become far apart from each other, suggesting that they end up belonging to different categories. Similarly, after being pushed away from prototype 2, voxel $C$ moves closer to $B$, suggesting that $B$ and $C$ appear to belong to the same category. This disruption of the semantic relationships among voxels can be detrimental to the segmentation task.

Then the critical question is, among a large number of voxels with unreliable pseudo-labels, which kind of information should be maintained during VCL? Our answer to this questions is: *neighbors*

*remain neighbors, and outsiders remain outsiders*. Correspondingly, we propose a novel distance-aware VCL (DVCL) technique. The rationale behind DVCL is simple: for a voxel with an unreliable pseudo-label, we wish to maintain two types of information. Its close voxels in feature space (referred to as neighbors), which likely belong to the same semantic category, should still be its neighbors after implementing DVCL, while its far-away voxels (referred to as outsiders), which likely belong to different categories, should remain its outsiders after DVCL. Specifically, in DVCL, we first identify, for each voxel with an unreliable pseudo-label, its neighbors and outsiders. During contrastive learning, for all unreliable voxels, we pull their neighbors into the same clusters while pushing outsiders away. This approach helps maintain useful semantic relationships among unreliable voxels while still enjoying the advantages of contrastive learning.

An example of using DVCL is shown in the bottom figure of Fig.1(b). Take voxel $B$ as an example (Query). Before implementing DVCL, voxel $A$ is one of its neighbors, and voxel $C$ is one of its outsiders. Then, after DVCL, voxel $A$ is pulled to the same cluster of $B$, while voxel $C$ is pushed away from it. In this way, the neighbors and outsiders of this query voxel $B$ are kept the same. One interesting thing we should emphasize is related to voxel $D$, which is neither a neighbor nor an outsider for the query voxel $B$. However, this does not mean it remains in the same location after DVCL. As voxel $D$, in most conditions, belongs to the neighbor or an outsider for some other voxels, it will be moved accordingly to maintain their relationships.

Additionally, in the image space, an entropy-based selection module (ESM) is designed to adaptively separate all pseudo-labels into two groups, *i.e.,* a reliable one and an unreliable one. Furthermore, reliable pseudo-labels are assigned higher weights, while unreliable pseudo-labels receive lower weights, effectively reweighting the unsupervised loss. We conduct extensive experiments on four datasets to demonstrate the effectiveness of DVCL. Compared to state-of-the-art methods, our proposed DVCL achieves superior performance. Additionally, we believe that DVCL can be applied to other segmentation tasks where there are a large number of unreliable pseudo-labels. Overall, our contributions can be summarized as follows:

- We propose a novel Distance-aware Voxel-based Contrastive Learning technique. By maintaining useful semantic relationships among unreliable voxels, discriminative features for segmentation can be learned.

- We propose an entropy-based learning approach to efficiently utilize pseudo-labels that have low confidence yet are accurate in image space.

- Extensive experiments on four medical multi-organ benchmarks demonstrate the effectiveness of our method and show that we advance the state-of-the-art approaches.

## 2 RELATED WORK

**Semi-supervised multi-organ segmentation.** Recently, semi-supervised methods for MoS have emerged (Zhou et al., 2019; Xia et al., 2020; Ma et al., 2023c). For example, Wen et al. Wen et al. (2024) proposed a DCL-Net to utilize global and local VCL to enhance model performance. For a more detailed summary of semi-supervised MoS methods, please refer to Appendix A.

**Semi-supervised contrastive learning.** Recently, VCL has achieved significant success in extracting powerful features from unlabeled data in SSL (Wang et al., 2021; 2022b; Feng et al., 2024; Deng et al., 2024). For example, Wang et al. (2021) proposed a VCL algorithm for SSL semantic segmentation that enhances similarity between embeddings of same-class voxels while distinguishing different-class voxels using pseudo-labels. However, unreliable pseudo-labels can introduce confirmation bias in VCL, ultimately reducing task performance. Wang et al. (2022b) proposed an uncertainty-guided contrastive learning method that discard unreliable pseudo-labels based on threshold. More recently, works such as (Wang et al., 2022c; Du et al., 2023; Feng et al., 2024; Deng et al., 2024) have focused on leveraging unreliable pseudo-labels for learning rather than filtering them out, often employing complementary labels. For a more detailed summary of complementary label VCL methods, please refer to Appendix A. However, this approach can disrupt the relationships among some unreliable voxels, and these relationship are helpful when learning features. In contrast, our method emphasizes preserving these semantic relationships among unreliable voxels. This capability is a key reason for the superiority of our method.

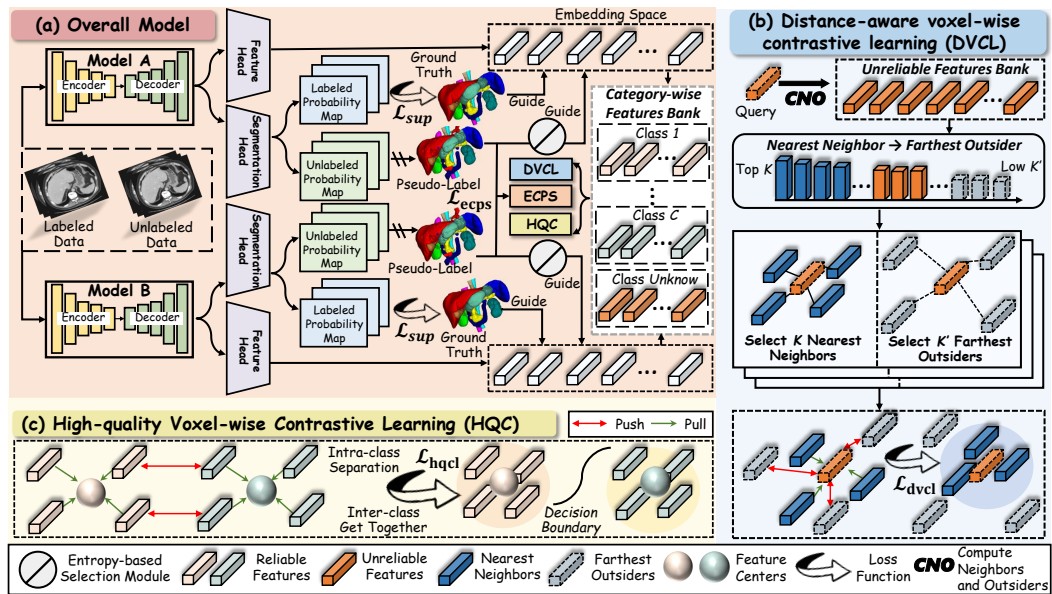

Figure 2: **Pipeline overview.** Overview of our framework, which consists of two networks. Labeled data is directly fed into both networks for supervised training. For unlabeled data, we separate the initially generated pseudo-labels into reliable and unreliable ones based on ESM. Reliable features (features with reliable pseudo-labels and ground-truth masks) are used to compute the $\mathcal{L}_{\mathrm{hqc}}$. Unreliable features (features with unreliable pseudo-labels) are used to compute the $\mathcal{L}_{\mathrm{dvcl}}$. Additionally, the $\mathcal{L}_{\mathrm{ecps}}$ is computed using the probability maps and all pseudo-labels.

**Self-supervised Learning.** The self-supervised training has been widely applied in the computer vision tasks (Wu et al., 2018; Yang et al., 2021; Chongjian et al.; Wu et al., 2023). For example, Debidatta et al. (Dwibedi et al., 2021) used an memory bank for the purpose of nearest neighbor mining (NNCLR). For a more detailed summary of self-supervised training methods, please refer to Appendix A.

## 3 METHOD

### 3.1 PROBLEM DEFINITION AND FRAMEWORK OVERVIEW

**Problem definition.** The training dataset, denoted as $\mathcal{S}$, consists of both labeled and unlabeled data, expressed as $\mathcal{S} = \mathcal{S}^l \cup \mathcal{S}^u$. Here, $\mathcal{S}^l = \{(x_i, y_i)\}_{i=1}^N$ constitutes the labeled data, and $\mathcal{S}^u = \{x_i\}_{i=1}^M$ represents the unlabeled data. ($M \gg N$ in most cases). In this context, $x_i \in \mathbb{R}^{D \times H \times W}$ refers to the input volume, and $y_i \in \mathbb{R}^{C \times D \times H \times W}$ denotes the ground-truth mask, where $C$ is the number of classes, including the background. $H$,$W$, and $D$ represent the height, width, and depth of the input medical volume, respectively. The goal is to train a segmentation network based on $\mathcal{S}^l$ and $\mathcal{S}^u$ that correctly predicts labels for unseen data.

**Framework overview.** An overview of the proposed framework is illustrated in Fig.2, which is based on the Cross Pseudo Supervision (CPS) framework (Chen et al., 2021) and utilizes two parallel segmentation networks: $model\ A$ and $model\ B$. Both networks share the same structure but are initialized with different weights. Each model consists of four main components: an encoder, a decoder, a segmentation head, and a feature head (*i.e.,* the sequentially-connected MLP layers). Full details of the framework are provided in Appendix B. The pseudo-code of our framework is summarized in Algorithm 1 (Appendix B). The framework incorporates two primary training strategies: i) An entropy-based learning approach to effectively utilize the pseudo-labels in the image space, which is described in Section.3.2. ii) Distance-aware voxel-wise contrastive learning to leverage feature interactions between unreliable voxels in the feature space, allowing for comprehensive mining of unreliable pseudo-label information. This is detailed in Section 3.3.

Figure 3: *Left*: The calculation process of the entropy-based Learning. *Right*: An illustration of the positive and negative candidates in the $\mathcal{L}_{hqc}$ and neighbors and outsiders in the $\mathcal{L}_{dvcl}$.

## 3.2 ENTROPY-BASED LEARNING

In the CPS method, the CPS loss is computed at the image level by considering all pseudo-labels without discrimination. However, overfitting to incorrect pseudo-labels can result in confirmation bias. To address this issue, existing methods Shen et al. (2023) use a confidence thresholding mechanism to filter out unreliable and potentially incorrect pseudo-labels during training. Unfortunately, this approach results in the underutilization of pseudo-labels, causing a significant loss of semantic information. Thus, we propose an entropy-based learning approach to mitigate confirmation bias and effectively utilize the pseudo-labels, as shown in Fig.3 left, including entropy-based selection module (ESM) and entropy-weighted cross pseudo supervision (ECPS) loss.

**Entropy-based Selection Module.** We follow Wang et al. (2022c) utilize voxel-wise entropy $e_{ij}^{(\cdot)}$ as the selection indicator between reliable and unreliable pseudo-labels in unlabeled data as:

$$e_{ij}^A = \sum_{c=1}^{C} -p_{ij}^A(c) \log\left(p_{ij}^A(c)\right), e_{ij}^B = \sum_{c=1}^{C} -p_{ij}^B(c) \log\left(p_{ij}^B(c)\right), \tag{1}$$

where $p_{ij}^{(\cdot)}(c)$ is the value of $p_{ij}^{(\cdot)}$ at $c$-th dimension. We then define $M_r^{(\cdot)}[i,j]$ and $M_u^{(\cdot)}[i,j]$ as the masks of low and high entropy for selecting reliable and unreliable pseudo-labels respectively. These masks can be expressed as:

$$\begin{cases} M_r^A[i,j] = \mathbb{I}(e_{ij}^A < \tau_e^A), M_u^A[i,j] = \mathbb{I}(e_{ij}^A > \tau_e^A), \\ M_r^B[i,j] = \mathbb{I}(e_{ij}^B < \tau_e^B), M_u^B[i,j] = \mathbb{I}(e_{ij}^B > \tau_e^B), \end{cases} \tag{2}$$

where $\tau_e^{(\cdot)}$ is the selection threshold of the entropy mask as follows:

$$\tau_e^A = e_{at}^A + \alpha \cdot e_{st}^A, \tau_e^B = e_{at}^B + \alpha \cdot e_{st}^B, \tag{3}$$

where $e_{at}^{(\cdot)}$ is the average entropy of voxels at training iteration $t$, *i,e*, $e_{at}^{(\cdot)} = \frac{\sum_{i=1}^{|\mathcal{B}|} \sum_{j=1}^{D \times H \times W}}{\mathcal{B} * D \times H \times W} e_{ij}^{(\cdot)}$ and $e_{st}^{(\cdot)}$ is the variance, *i,e*, $e_{st}^{(\cdot)} = \sqrt{\frac{\sum_{i=1}^{|\mathcal{B}|} \sum_{j=1}^{D \times H \times W}}{\mathcal{B} * D \times H \times W} \left(e_{ij}^{(\cdot)} - e_{at}^{(\cdot)}\right)^2}$. $\alpha$ is factor and is set to 0.5 in our experiments. This new selection strategy has the following benefits: i) ESM allows the model to adapt its confidence across different training stages, thereby enabling a clearer distinction between reliable and unreliable pseudo-labels. ii) ESM compares confidence against a threshold, $\tau_e$, without needing to sort all training voxels (Wang et al., 2022c; Ma et al., 2023a; Wang et al., 2023). This reduces the time complexity from $O(\log N)$ to $O(1)$, making the strategy more efficient when applied to large-scale datasets.

**Entropy-weighted Cross Pseudo Supervision loss.** Based on the segmented regions of reliable and unreliable pseudo-labels, we utilize an exponential function (Chen et al., 2023b) to fit the confidence

distribution. The weights are calculated as follows:

$$\mathcal{W}^A = \begin{cases} \mathcal{W}_{\max} \exp\left(\frac{-e_{ij}^A}{\log_2 C}\right), & \text{if } x_{ij} \in M_u^A[i,j], \\ \mathcal{W}_{\max}, & \text{otherwise,} \end{cases} \quad \mathcal{W}^B = \begin{cases} \mathcal{W}_{\max} \exp\left(\frac{-e_{ij}^B}{\log_2 C}\right), & \text{if } x_{ij} \in M_u^B[i,j], \\ \mathcal{W}_{\max}, & \text{otherwise,} \end{cases} \quad (4)$$

where $\mathcal{W}_{\max}$ is the weight for reliable pseudo-labels and is set to 1 in our experiments. The weights of the unreliable pseudo-labels are calculated using an exponential function, where high uncertainty corresponds to small weights, and low uncertainty corresponds to large weights. Therefore, this approach increases the utilization of pseudo-labels by assigning lower weights to unreliable pseudo-labels. Based on the designed weights, ECPS loss is defined as follows:

$$\mathcal{L}_{\text{ecps}} = \frac{\sum_{i=1}^{\mathcal{B}_u} \sum_{j=1}^{D \times H \times W}}{|\mathcal{B}_u| \times D \times H \times W} \left(\mathcal{W}^B \mathcal{L}_{\text{ce}}(p_{ij}^A, \hat{y}_{ij}^B) + \mathcal{W}^A \mathcal{L}_{\text{ce}}(p_{ij}^B, \hat{y}_{ij}^A)\right), \quad (5)$$

where $\hat{y}_{ij}^{(\cdot)} = \arg\max(p_i^{(\cdot)})$ is the pseudo-labels of the $j$-th voxel of $i$-th unlabeled image. $\mathcal{L}_{\text{ce}}$ indicates the cross-entropy loss.

### 3.3 DISTANCE-AWARE VOXEL-WISE CONTRASTIVE LEARNING

To effectively maintain useful semantic relationships among unreliable voxels while still enjoying the advantages of VCL, we propose a distance-aware VCL (DVCL) technique. Using the selection of reliable and unreliable pseudo-labels from Section 3.2, we employ the high-quality VCL (HqC) loss for voxels with reliable pseudo-labels and ground-truth masks. For voxels with unreliable pseudo-labels, we apply the DVCL loss.

**High-quality Voxel-wise Contrastive Learning.** During network training, anchor voxels are sampled for each class present in the current mini-batch. For labeled data, we select voxels with accurate predictions based on ground-truth masks, ensuring the quality of the anchor voxels. The feature sets of the labeled anchor voxels for class $c$ are selected as:

$$\mathcal{R}_c^{Al} = \{\mathbf{r}_{ij}^A \mid y_{ij} = c, \arg\max p_{ij}^A(c) = c\}, \mathcal{R}_c^{Bl} = \{\mathbf{r}_{ij}^B \mid y_{ij} = c, \arg\max p_{ij}^B(c) = c\} \quad (6)$$

where $\mathbf{r}_{ij}^{(\cdot)} \in \mathbb{R}^d$ represents the feature of the $j$-th voxel of $i$-th labeled image given by feature head. $y_{ij}$ denotes the ground-truth mask. Similar to Singh (2021), the pseudo-labels obtained from the segmentation head are utilized as class labels for the unlabeled voxels. The feature sets of anchor voxels with reliable pseudo-labels are selected as:

$$\mathcal{R}_c^{Au} = \left\{\mathbf{r}_{ij}^A \mid \hat{y}_{ij}^A = c, x_{ij} \in M_r^A[i,j]\right\}, \mathcal{R}_c^{Bu} = \left\{\mathbf{r}_{ij}^B \mid \hat{y}_{ij}^B = c, x_{ij} \in M_r^B[i,j]\right\}, \quad (7)$$

where $M_r^{(\cdot)}[i,j]$ denotes the reliable pseudo-labels by following Eq.2. The set of all qualified anchor voxels for class $c$ is defined as follows:

$$\mathcal{R}_c = \mathcal{R}_c^{Al} \bigcup \mathcal{R}_c^{Bl} \bigcup \mathcal{R}_c^{Au} \bigcup \mathcal{R}_c^{Bu}. \quad (8)$$

To prevent limited anchors leading to an overly localized and unstable sample center, we store $\mathcal{R}_c$ by using a memory bank. When the memory bank is saturated, we remove old features to leave enough space to store the latest features. Inspired by (Miao et al., 2023), based on the anchor voxels defined in Eq.8, the positive candidate for class $c$ is set as the feature center of all anchor voxels:

$$\mathcal{P}_c = \frac{1}{|\mathcal{R}_c|} \sum_{\mathbf{r}_c \in \mathcal{R}_c} \mathbf{r}_c. \quad (9)$$

We consider labeled voxels that are misclassified as class $c$ as negative candidates for that class, helping to guide these voxels away from misclassified categories in the feature space. Consequently, the feature sets of negative candidates for class $c$ are defined as follows:

$$\mathcal{G}_c^{Al} = \{\mathbf{r}_{ij}^A \mid \arg\max p_{ij}^A(c) = c, y_{ij} \neq c\}, \mathcal{G}_c^{Bl} = \{\mathbf{r}_{ij}^B \mid \arg\max p_{ij}^B(c) = c, y_{ij} \neq c\}. \quad (10)$$

Based on the anchor voxels, positive candidates, and negative candidates, we use the popular InfoNCE loss function (Oord et al., 2018) to calculate the contrastive loss. The HqC loss $\mathcal{L}_{\text{hqc}}$ is defined as:

$$\mathcal{L}_{\text{hqc}} = -\frac{\sum_{c=1}^C \sum_{m=1}^M}{C \times M} \log \frac{\exp\left(\langle \mathbf{r}_{cm}, \mathcal{P}_c \rangle / \tau\right)}{\exp(\langle \mathbf{r}_{cm}, \mathcal{P}_c \rangle / \tau) + \sum_{n=1}^N \exp\left(\langle \mathbf{r}_{cm}, \mathbf{r}_{cmn}^- \rangle / \tau\right)}, \quad (11)$$

where $M$ is the number of anchor voxels, and $\mathbf{r}_{cm}$ denotes the feature vectors of the $m$-th anchor voxels of class $c$. Each anchor voxel is followed by a positive candidate $\mathcal{P}_c$ and $N$ negative candidates, $\{\mathbf{r}_{cmn}^-\}_{n=1}^{N}$. $\langle\cdot,\cdot\rangle$ denotes the cosine similarity between features. In this study, we set $M$ =256, $N$ =50, and $\tau = 1$.

**Distance-aware Voxel-wise Contrastive Learning.** From the perspective that *features that are close or distant in the feature space should yield consistent or inconsistent predictions*, we propose a DVCL technique to maintain useful semantic relationships among unreliable voxels. Inspired by Goldberger et al. (2004); Wu et al. (2018), we define $\mathcal{O}_{m,n}$ as the probability that the feature $\mathbf{r}_m$ has same prediction to feature $\mathbf{r}_n$ :

$$\mathcal{O}_{m,n} = \frac{e^{p_m^T p_n}}{\sum_{\mathbf{r}_q \in U_\mathcal{R}} e^{p_m^T p_q}},\tag{12}$$

where $U_\mathcal{R} = M_u^A \bigcup M_u^B$ denotes the feature set of all voxels with unreliable pseudo-labels by following Eq.2. $\mathbf{r}_m$ and $\mathbf{r}_n$ denote the features of two voxels from $U_\mathcal{R}$, while having corresponding predict probabilities $p_m$ and $p_n$. We then define two sets for each feature $\mathbf{r}_m$ (quary) in $U_\mathcal{R}$: close neighbor set $\mathcal{C}_m$ (features potentially from same classes) and distant outsider set $\mathcal{D}_m$ (features potentially from different classes). Inspired by (Yang et al., 2021), $\mathcal{C}_m$ is formed by selecting the $K$-Nearest Neighbors(KNN) (Cover & Hart, 1967) of $\mathbf{r}_m$ from $U_\mathcal{R}$ using cosine similarity as the distance metric. Conversely, $\mathcal{D}_m$ is constructed using the $K'$- Furthest Neighbors of $\mathbf{r}_m$:

$$\mathcal{C}_m = \{\mathbf{r}_n | top\text{-}K(cos\,(\mathbf{r}_m, \mathbf{r}_n)\,, \forall \mathbf{r}_n \in U_\mathcal{R})\}, \mathcal{D}_m = \{\mathbf{r}_k | low\text{-}K'(cos\,(\mathbf{r}_m, \mathbf{r}_k)\,, \forall \mathbf{r}_k \in U_\mathcal{R})\}.\tag{13}$$

Following Dwibedi et al. (Dwibedi et al., 2021), we use queues as candidate sets $\mathcal{C}_m$ and $\mathcal{D}_m$, respectively, where each element represents a feature. During network training with sample batches, we store features in $\mathcal{C}_m$ and $\mathcal{D}_m$, updating the candidate sets using a first-in-first-out strategy.

Returning to our motivation, for each feature $\mathbf{r}_m$, the features in $\mathcal{D}_m$ should have different predictions than those in $\mathcal{C}_m$. This process involves moving features in $\mathcal{C}_m$ towards $\mathbf{r}_m$ while pushing away features in $\mathcal{D}_m$. To achieve this, we first define two likelihood functions:

$$\mathcal{N}\,(\mathcal{C}_m | \theta_\mathcal{S}) = \prod_{\mathbf{r}_n \in \mathcal{C}_m} \mathcal{O}_{m,n} = \prod_{\mathbf{r}_n \in \mathcal{C}_m} \frac{e^{p_m^T p_n}}{\sum_{\mathbf{r}_q \in U_\mathcal{R}} e^{p_m^T p_q}}, \mathcal{N}\,(\mathcal{D}_m | \theta_\mathcal{S}) = \prod_{\mathbf{r}_k \in \mathcal{D}_m} \mathcal{O}_{m,n} = \prod_{\mathbf{r}_k \in \mathcal{D}_m} \frac{e^{p_m^T p_k}}{\sum_{\mathbf{r}_q \in U_\mathcal{R}} e^{p_m^T p_q}},\tag{14}$$

where $\theta_\mathcal{S}$ denotes the parameters of the segmentation head in our network. We then propose to simultaneously maximize the likelihood of the close neighbor set and minimize the likelihood of the distant outsider set, denoted as

$$\psi(\mathcal{C}_m, \mathcal{D}_m) = -\log \frac{\mathcal{N}\,(\mathcal{C}_m \mid \mathbf{r}_m, \theta_\mathcal{S})}{\mathcal{N}\,(\mathcal{D}_m \mid \mathbf{r}_m, \theta_\mathcal{S})},\tag{15}$$

One problem optimizing Eq.15 is that it requires the participation of all features in $U_\mathcal{R}$ to compute Eq.14, which, in practice, can result in a significant waste of time and resources. Here we resort to get an upper-bound of Eq.15:

$$\begin{aligned}
\psi(\mathcal{C}_m, \mathcal{D}_m) &= -\log \frac{\mathcal{N}\,(\mathcal{C}_m \mid \theta_\mathcal{S})}{\mathcal{N}\,(\mathcal{D}_m \mid \theta_\mathcal{S})} \\
&= -\sum_{\mathbf{r}_n \in \mathcal{C}_m} p_m^T p_n + \sum_{\mathbf{r}_k \in \mathcal{D}_m} p_m^T p_k + (|\mathcal{C}_m| - |\mathcal{D}_m|) \log(\sum_{\mathbf{r}_q \in U_\mathcal{R}} e^{p_m^T p_q}) \\
&\leq -\sum_{\mathbf{r}_n \in \mathcal{C}_m} p_m^T p_n + \sum_{\mathbf{r}_k \in \mathcal{D}_m} p_m^T p_k + (|\mathcal{C}_m| - |\mathcal{D}_m|)(\log |U_\mathcal{R}| + \sum_{\mathbf{r}_q \in U_\mathcal{R}} \frac{p_m^T p_q}{|U_\mathcal{R}|}) \\
&< -\sum_{\mathbf{r}_n \in \mathcal{C}_m} p_m^T p_n + \sum_{\mathbf{r}_k \in \mathcal{D}_m} p_m^T p_k + (|\mathcal{C}_m| - |\mathcal{D}_m|)(\log |U_\mathcal{R}| + 1) \\
&= \overline{\psi}(\mathcal{C}_m, \mathcal{D}_m).
\end{aligned}\tag{16}$$

Details for our proof can be found in Appendix.J.

Finally, our DVCL loss is defined as follows:

$$\mathcal{L}_{\text{dvcl}} = \frac{1}{|U_\mathcal{R}|} \sum_{\mathbf{r}_m \in U_\mathcal{R}} \overline{\psi}(\mathcal{C}_m, \mathcal{D}_m).\tag{17}$$

**Overall loss.** Finally, our overall contrastive loss can be formulated as:

$$\mathcal{L}_{\text{contra}} = \mathcal{L}_{\text{hqc}} + \beta \mathcal{L}_{\text{dvcl}}\tag{18}$$

where the scalar $\beta = 0.3$ is used to balance the two loss functions.

Table 1: Quantitative results on two settings of FLARE 2022 dataset. 'w/o VCL ' or 'w/ VCL' indicates whether the SSL methods combined with VCL or not. Prevalent SSL methods, including DAN (Zhang et al., 2017), MT (Tarvainen & Valpola, 2017), UA-MT (Yu et al., 2019), SASSnet (Li et al., 2020), DTC (Luo et al., 2021a), CPS (Chen et al., 2021), CLD (Lin et al., 2022), DHC (Wang & Li, 2023), MagicNet (Chen et al., 2023a), UGPCL (Chen et al., 2021), U$^2$PL (Wang et al., 2022c), BaCon (Feng et al., 2024), and CCL (Deng et al., 2024). Best results are boldfaced.

| Methods | Liv | Spl | Sto | L.kid | R.kid | Aor | Pan | IVC | Duo | Gal | Eso | RAG | LAG | Mean Dice | Mean Jaccard |
|---|---|---|---|---|---|---|---|---|---|---|---|---|---|---|---|
| Fully | 94.21 | 88.32 | 49.40 | 91.68 | 91.33 | 89.89 | 48.83 | 79.24 | 52.10 | 56.24 | 61.13 | 44.87 | 42.12 | 68.41±0.58 | 56.97±0.42 |
| *50% labeled data (labeled:unlabeled=42:42)* | | | | | | | | | | | | | | | |
| DAN (w/o VCL) | 96.50 | 89.74 | 62.04 | 93.63 | 93.24 | 90.76 | 61.71 | 80.09 | 66.60 | 70.07 | 67.20 | 58.04 | 43.47 | 74.86±0.69 | 63.73±0.62 |
| MT (w/o VCL) | 96.97 | 89.64 | 66.63 | 93.66 | 92.58 | 91.39 | 68.65 | 82.08 | 60.96 | 69.89 | 71.68 | 57.51 | 62.21 | 77.22±0.42 | 65.94±0.83 |
| UA-MT (w/o VCL) | **97.21** | 88.85 | 71.66 | 94.00 | 93.50 | 92.41 | 70.60 | 82.92 | 64.85 | 76.82 | 72.05 | 60.02 | 60.85 | 78.91±0.89 | 68.01±1.21 |
| SASSnet (w/o VCL) | 95.25 | 92.03 | 66.48 | 92.47 | 93.79 | 90.03 | 63.61 | 79.94 | 60.14 | 65.57 | 70.83 | 59.98 | 62.78 | 76.38±0.61 | 65.20±0.60 |
| DTC (w/o VCL) | 96.47 | 91.33 | 65.94 | 94.46 | 93.57 | 92.52 | 64.88 | 83.77 | 65.58 | 75.80 | 68.53 | 68.87 | 61.35 | 78.70±0.79 | 67.64±1.05 |
| CPS (w/o VCL) | 96.77 | 91.23 | 72.63 | 93.38 | 93.70 | 92.35 | 70.34 | 83.30 | 65.48 | 78.81 | 72.37 | 58.34 | 61.47 | 79.24±0.56 | 67.99±0.56 |
| CLD (w/o VCL) | 94.27 | 88.54 | 74.88 | 94.48 | 93.33 | 91.51 | 71.51 | 83.21 | 68.15 | 76.68 | 71.03 | 64.57 | 63.98 | 79.47±0.27 | 67.95±0.28 |
| DHC (w/o VCL) | 92.54 | 90.81 | 76.96 | 93.39 | 92.18 | 91.84 | 74.65 | 83.25 | **69.44** | 84.60 | 72.91 | 64.52 | 55.88 | 80.23±1.07 | 68.90±0.26 |
| MagicNet (w/o VCL) | 96.49 | 88.84 | 80.33 | 90.84 | 93.06 | 91.72 | 69.46 | 81.98 | 67.44 | 82.81 | 75.79 | 63.40 | 59.98 | 80.16±0.33 | 69.08±0.12 |
| UGPCL (w/ VCL) | 96.00 | 90.63 | 79.75 | **94.88** | 94.01 | 90.86 | 73.39 | 82.26 | 66.28 | 81.02 | 71.42 | 61.81 | 65.44 | 80.59±0.85 | 69.76±0.85 |
| U$^2$PL (w/ VCL) | 96.83 | 90.67 | 77.19 | 94.25 | **94.33** | **92.73** | 74.36 | 84.22 | 68.37 | 82.99 | 75.57 | 66.83 | 67.27 | 81.97±0.31 | 71.69±0.31 |
| BaCon (w/ VCL) | 95.54 | 90.81 | 76.89 | 93.78 | 93.49 | 91.71 | 74.68 | 83.77 | 68.67 | 83.33 | 73.18 | 68.80 | 66.54 | 81.63±0.56 | 70.82±0.57 |
| CCL (w/ VCL) | 96.25 | 91.65 | 76.14 | 93.04 | 93.48 | 92.15 | 73.44 | 83.19 | 67.35 | 82.49 | 75.54 | 68.05 | 64.39 | 81.32±0.32 | 70.68±0.51 |
| **Ours (w/ VCL)** | 96.75 | **94.06** | **83.72** | 94.43 | 93.43 | 92.31 | **75.84** | **84.28** | 69.36 | **86.44** | **75.88** | **69.36** | **68.31** | **83.40±0.32** | **72.65±0.47** |
| *10% labeled data (labeled:unlabeled=42:420)* | | | | | | | | | | | | | | | |
| DAN (w/o VCL) | 95.89 | 84.15 | 67.29 | 91.96 | 91.35 | 91.38 | 63.12 | 79.33 | 66.48 | 77.29 | 67.82 | 50.41 | 48.41 | 75.10±0.69 | 63.83±0.23 |
| MT (w/o VCL) | 96.49 | 91.54 | 74.64 | 93.78 | 92.77 | 92.17 | 69.23 | 82.71 | 66.68 | 73.49 | 70.66 | 61.88 | 41.26 | 77.49±0.48 | 66.45±0.39 |
| UA-MT (w/o VCL) | 96.42 | 91.98 | 79.91 | 92.74 | 92.83 | 92.33 | 71.43 | 83.10 | 67.72 | 77.26 | 72.41 | 64.04 | 46.18 | 79.10±0.38 | 68.21±0.48 |
| SASSnet (w/o VCL) | 96.21 | 90.40 | 67.12 | 94.00 | 92.85 | 91.61 | 67.89 | 79.59 | 65.47 | 71.44 | 72.41 | 52.07 | 57.83 | 76.77±0.30 | 65.89±0.44 |
| DTC (w/o VCL) | 96.63 | 92.91 | 72.76 | 92.68 | 92.40 | 91.87 | 66.82 | 81.47 | 65.76 | 78.38 | 69.39 | 59.74 | 59.10 | 78.45±0.82 | 67.05±1.02 |
| CPS (w/o VCL) | 96.62 | 92.16 | 77.02 | 92.70 | 92.71 | 92.25 | 69.39 | 81.91 | 65.94 | 75.12 | 72.78 | 63.56 | 58.96 | 79.32±0.46 | 68.14±0.61 |
| CLD (w/o VCL) | 94.63 | 89.74 | 73.20 | 91.76 | 92.97 | 91.61 | 70.27 | 83.12 | 68.13 | 84.15 | 72.69 | 67.89 | 55.27 | 79.65±0.17 | 68.22±0.49 |
| DHC (w/o VCL) | 93.17 | 90.64 | 80.56 | 93.13 | 92.89 | 91.38 | 72.22 | 83.75 | 69.73 | 82.47 | 73.25 | 67.12 | 56.19 | 80.50±0.43 | 69.38±0.63 |
| MagicNet (w/o VCL) | 97.04 | 88.04 | 81.51 | 92.18 | 92.95 | 91.75 | 71.15 | 81.01 | 69.61 | 84.36 | **77.07** | 63.34 | 60.33 | 80.79±0.75 | 70.23±0.96 |
| UGPCL (w/ VCL) | 96.81 | 92.11 | 75.48 | 94.02 | **94.79** | 92.07 | 68.75 | 82.46 | 68.60 | 77.50 | 71.09 | 63.54 | 58.10 | 79.64±0.72 | 69.09±0.58 |
| U$^2$PL (w/ VCL) | 96.34 | 91.45 | 80.09 | 94.30 | 93.81 | 91.17 | 74.53 | 83.13 | 69.88 | 84.40 | 72.70 | 67.25 | 68.12 | 82.09±0.32 | 71.34±0.51 |
| BaCon (w/ VCL) | 95.88 | 90.35 | 77.17 | 94.35 | 93.98 | 92.10 | 73.32 | 83.75 | 69.67 | 81.15 | 74.87 | 69.44 | 69.55 | 81.97±0.33 | 71.04±0.40 |
| CCL (w/ VCL) | 95.77 | 91.44 | 76.81 | 92.71 | 93.54 | 92.02 | 73.59 | 83.58 | 67.17 | 81.19 | 75.45 | 70.15 | 69.99 | 81.80±0.11 | 70.91±0.32 |
| **Ours (w/ VCL)** | **97.37** | **93.53** | **83.47** | **94.86** | 93.69 | **92.52** | **76.98** | **84.43** | **70.97** | **87.71** | 75.69 | **70.60** | **71.67** | **84.11±0.08** | **73.49±0.13** |

# 4 EXPERIMENTS

## 4.1 DATASET AND IMPLEMENTATION DETAILS

We evaluate our method using four widely recognized MoS datasets, *i.e.*, FLARE 2022 dataset (Ma et al., 2023b), AMOS dataset (Ji et al., 2022), MMWHS dataset (Zhuang & Shen, 2016), and BTCV dataset (Landman et al., 2015). For more details, please refer to the Appendix.C. We implement the proposed framework with PyTorch, using 2 NVIDIA A100 GPUs. For more implementation details, *i.e.*, data preprocessing, network parameters, data augmentation, learning rate policy, batch sizes, *etc.,* please refer to the Appendix.D. In our experiments, the segmentation performances of the different methods are evaluated using two standard evaluation metrics: the Dice and the Jaccard index (referred to as Jaccard). To reduce the randomness of network training, experiments are calculated in triplicate for all methods and the mean and standard deviation (SD) of the Dice and Jaccard values are calculated. Model weights are determined based on performance on the validation set, while comparisons of different methods are made using segmentation metrics on the test set.

## 4.2 COMPARISON TO SOTA METHODS

We demonstrate that our method achieves superior performance across all datasets and label ratios (*i.e.*, 10%, 50%). As shown in Table.1, Table.5 ( Appendix.F ), Table.6 ( Appendix.G ), and Table.7 (Appendix.H), our methods consistently outperform all the compared SSL-based methods (For more methods details, please refer to the Appendix.E) by a considerable margin across all datasets and label ratios. This validates the superior performance of our proposed methods in terms of segmentation accuracy and label efficiency. For example, compared to the second-best results, our method under {10%, 50%} label ratios achieves {2.02%↑, 1.43%↑}, {5.28%↑, 2.13%↑}, {2.01%↑,

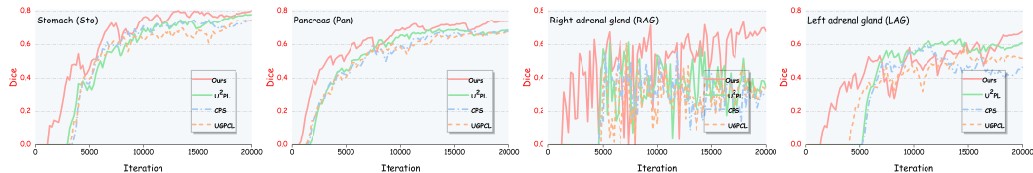

Figure 4: Dice curves generated by different methods on validation set of FLARE 2022 during network training.

1.04%↑}, {2.53%↑, 3.84%↑} in mean Dice across FLARE 2022, AMOS, MMWHS, and BTCV, respectively. These results indicate that our method is well segmented relative to other methods. We provide qualitative illustrations of FLARE 2022, AMOS, and MMWHS in Fig.5, Fig.6, Fig.7( Appendix.G ), respectively.

Furthermore, As shown in Fig.4, we plot the Dice curves of four challenging organs (*i.e.*, Sto, Pan, RAG, and LAG) obtained by different methods on the validation set during network training. In the case of these challenging organs, our method is able to learn organ knowledge earlier than other methods. For instance, in the curve for the RAG, our approach achieves this at least 4000 iterations earlier than others. Furthermore, UGPCL (discard unreliable pseudo-labels) exhibits lower performance compared to the baseline (CPS), indicating that discarding unreliable voxels from these challenging organs leads to significant information loss and, consequently, a decline in performance. In contrast, our algorithm outperforms the $U^2PL$ (using complementary labels) in terms of performance.

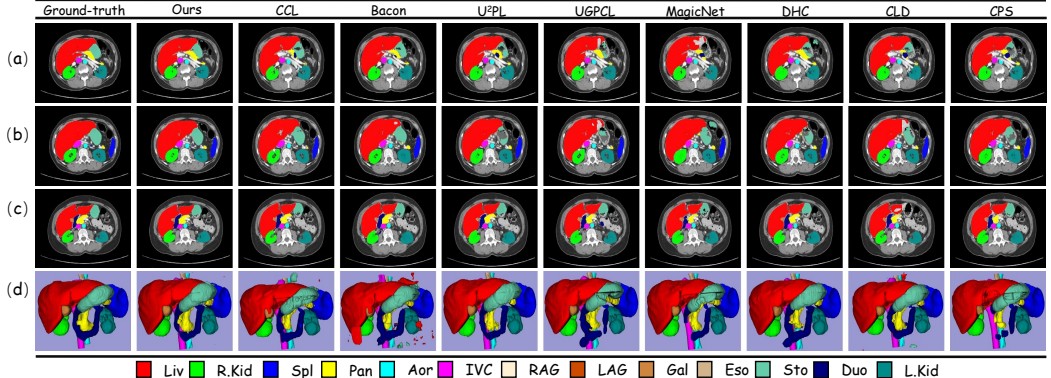

Figure 5: Visualization of the segmentation results for the FLARE 2022 dataset. (a-c) Segmentation results for one case of three transverse sections and (d) 3D segmentation views.

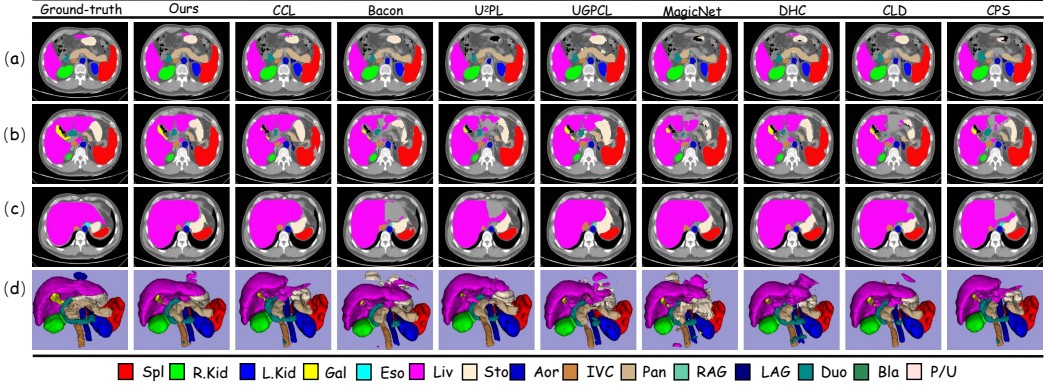

Figure 6: Visualization of the segmentation results for the AMOS dataset. (a-c) Segmentation results for one case of three transverse sections and (d) 3D segmentation views.

Table 2: Ablation on component aspect: (1) EBL; (2) DVCL. Best results are boldfaced.

| Baseline | EBL | DVCL | Mean Dice for each organ | | | | | | | | | | | | | Mean Dice | Mean Jaccard |
|---|---|---|---|---|---|---|---|---|---|---|---|---|---|---|---|---|---|
| | | | Liv | Spl | Sto | L.kid | R.kid | Aor | Pan | IVC | Duo | Gal | Eso | RAG | LAG | | |
| ✔ | | | 96.92 | 91.86 | 77.02 | 92.70 | 92.71 | 92.25 | 69.39 | 81.91 | 65.94 | 75.12 | 72.78 | 63.56 | 58.96 | 79.32±0.46 | 68.14±0.61 |
| ✔ | ✔ | | 96.58 | 90.56 | 82.8 | 93.77 | 93.35 | 92.23 | 76.59 | 83.93 | 69.04 | 82.7 | 73.43 | 67.04 | 68.72 | 82.36±0.28 | 71.4±0.48 |
| ✔ | | ✔ | 96.84 | **95.04** | 82.65 | 94.69 | 93.43 | 92.14 | 76.83 | 83.6 | 70.5 | 85.72 | 74.17 | **70.65** | 68.08 | 83.41±0.31 | 72.5±0.54 |
| ✔ | ✔ | ✔ | **97.37** | 93.53 | **83.47** | **94.86** | **93.69** | **92.52** | **76.98** | **84.43** | **70.97** | **87.71** | **75.69** | 70.60 | **71.67** | **84.11±0.08** | **73.49±0.13** |

Table 3: Analysis of the number of neighbors $K$ and outsiders $K'$. Best results are boldfaced.

| K neighbors | K' outsiders | Mean Dice | Mean Jaccard |
|---|---|---|---|
| 5 | 5 | 82.38±0.27 | 71.07±0.25 |
| 5 | 10 | 82.92±0.45 | 71.81±0.34 |
| 10 | 10 | 83.24±0.17 | 72.24±0.17 |
| 10 | 15 | **84.11±0.08** | **73.49±0.13** |
| 15 | 15 | 83.81±0.12 | 73.12±0.09 |

Table 4: Performance comparisons of DVCL and current nearest neighbor VCL.

| Methods | Mean Dice | Mean Jaccard |
|---|---|---|
| NCCL (Wu et al., 2023) | 80.98±0.79 | 70.99±0.82 |
| NNCLR (Dwibedi et al., 2021) | 83.17±0.34 | 72.09±0.41 |
| SNCLR (Chongjian et al.) | 83.67±0.71 | 72.74±0.67 |
| **DVCL (ours)** | **84.11±0.08** | **73.49±0.13** |

## 4.3 ABLATION STUDY

In this subsection, we conduct various ablations to better understand our design choices. For all the ablation experiments the models are trained on FLARE 2022 dataset with 10% labeled ratio.

**Importance of Each Components.** We analyze the effectiveness of different components of our method. At the core of our approach is the combination of two strategies: an entropy-based learning (EBL) approach and distance-aware voxel-wise contrastive learning (DVCL). We deactivate each component and then evaluate the resulting models, as shown in Table 2. As illustrated, both EBL and DVCL significantly enhance performance, particularly for challenging organs (*i.e.,* Sto; Pan; LAG; RAG). Furthermore, the integration of both EBL and DVCL yields the highest mean Dice (84.11%) and mean Jaccard (73.49%).

**Importance of Neighbors and Outsiders.** Although voxels from different organs might become neighbors due to similar features in the feature space, by carefully choosing the value for the number of neighbors,$K$, we can effectively reduce such misclassifications and ensure that the selected neighbors are as likely as possible to belong to the same class. As shown in Table 3, the optimal values are $K = 10$ and $K' = 15$, which yield the highest mean Dice and mean Jaccard. When $K$ is small, the query voxel has fewer neighbors. Although these neighbors have high similarity, they may only contain limited semantic information. Conversely, when $K$ is large, the number of neighbors increases, potentially introducing misclassified samples with high similarity but belonging to different categories.

**Analysis on The Nearest-neighbor Voxel-wise Contrastive Learning.** In Table 4, we compare DVCL with existing nearest neighbor VCL methods (Wu et al., 2023; Dwibedi et al., 2021; Chongjian et al.) and find that DVCL exhibits superior performance. This improvement can be attributed to two main factors. First, for unreliable voxels, DVCL not only considers similar neighbors but also introduces outsiders, which provide valuable inter-class contrastive information. Second, DVCL transcends traditional positive and negative sample pair comparisons by employing contrastive learning between two feature sets. Moreover, the results of ablation studies of the contrastive learning strategycan be found in Appendix.M.

## 5 CONCLUSION

In this work, we introduce a semi-supervised VCL method for MoS, termed DVCL, to make sufficient use of voxels with unreliable pseudo-labels. Contrary to the previous works that use complementary labels, DVCL helps maintain useful semantic relationships among unreliable voxels while still enjoying the advantages of VCL. Additionally, we propose an entropy-based learning approach to efficiently utilize pseudo-labels that have low confidence yet are accurate in image space. Comparative experiments against state-of-the-art methods and extensive ablation studies demonstrate the effectiveness of our method.

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

# Appendix

The appendix is organized as follows:

## A  RELATED WORK

**Semi-supervised Multi-organ Segmentation.** Accurately segmenting multiple organs in CT scans is crucial for a variety of medical procedures (Gao et al., 2023). Lee et al. (Lee et al.) proposed a lightweight volumetric ConvNet for multi-organ segmentation in CT scans called 3D UX-Net for segmenting 13 organs from CT scans. 3D UX-Net is a supervised network that relies on the quantity and quality of labeled data. Recently, a number of semi-supervised learning methods were developed for multi-organ segmentation in CT scans (Zhou et al., 2019; Xia et al., 2020; Ma et al., 2023c). To alleviate the class-imbalance problem, Lin et al. Lin et al. (2022) proposed a a novel framework to leverage label distribution to encourage the network to put more effort into small organ. Wang et al. Wang & Li (2023) proposed a dual-debiased heterogeneous co-training (DHC) framework that leverages the pseudo-labels dynamically to guide the model to solve data and learning biases. Chen et al. Chen et al. (2023a) designed a data augmentation strategy (MagicNet) that encourages unlabeled images to learn an organ's semantic information regarding its position relative to labeled images. Furthermore, Wen et al. Wen et al. (2024) proposed a two-stage dual contrastive learning network (DCL-Net) to utilize global and local contrastive learning to strengthen the relations among images and classes. In this work, we also use contrastive learning to fully utilize unlabeled data and propose a feature interaction method for unreliable voxels to deeply explore their semantic relationships, thereby improving the segmentation performance of difficult organs.

**Complementary Label VCL.**

- U$^2$PL (Wang et al., 2022c): Instead of discarding unreliable pixels, they are effectively treated as negative samples for the least likely categories. This enables a semi-supervised semantic segmentation framework that leverages these uncertain predictions to enhance learning.

- Lion (Du et al., 2023): They introduce a negative learning module to extract knowledge from complementary labels, thereby maximizing the utility of underestimated, low-confidence unlabeled data. Furthermore, they propose an innovative NC-Loss to enhance the model's performance.

- BaCon (Feng et al., 2024): To address the issue of class imbalance across multiple categories, they build upon the complementary label approach and propose a Balanced Distribution-related Temperature Adjusting (BTA) method. This approach dynamically modulates the class-wise temperature of positive pairs in VCL.

- CCL (Deng et al., 2024): They propose a Contrastive Complementary Labeling (CCL) method, which constructs reliable negative pairs using the complementary labels of low-confidence data. This approach supplies discriminative information for subsequent contrastive learning while effectively avoiding the introduction of potential noise.

**Self-supervised Learning.** A particular kind of self-supervised training – known as instance discrimination has become popular. For example, Wu et al. (Wu et al., 2018) proposed an unsupervised feature learning approach by maximizing distinction between instances via a novel non parametric softmax formulation. Debidatta Dwibedi et al. (Dwibedi et al., 2021) used an explicit support set (also known as a memory queue/bank) for the purpose of nearest neighbor mining(NNCLR). Furthermore, Yang et al. (Yang et al., 2021) only use a few neighbors from the feature bank to cluster the target features with a consistency regularization. Recently, Ge et al. (Chongjian et al.) computed cross-attention scores to explore soft neighbors during contrastive learning (SNCLR). Building on these approaches, Wu et al. (Wu et al., 2023) proposed a neighbor-guided consistent and contrastive learning framework for semi-supervised setting (NCCL). However, this approach can disrupt the relationships among some unreliable voxels, and these relationship are helpful when learning features. However, these existing methods select only semantically similar positive samples based on neighbors, while we introduce not only similar samples but also dissimilar ones, providing valuable inter-class contrastive information.

## B  FRAMEWORK OVERVIEW

This framework encompasses two parallel segmentation networks, $model\ A$ and $model\ B$. These networks share a common structure, but they are initialized differently; however, both are trained on the same input image. The purpose of this approach is to ensure that both networks produce consistent outputs. For the unlabeled data, each segmentation network can generate a pseudo-label, which serves as an additional signal to supervise the other segmentation network.

At each training step, $\mathcal{B}_l$ labeled data and $\mathcal{B}_u$ unlabeled data are sampled and fed into the $model\ A$ and the $model\ B$, respectively. For the labeled data, the supervised loss function is applied, guiding each segmentation head to generate a prediction mask that closely aligns with the ground-truth mask:

$$\mathcal{L}_{\text{sup}} = \frac{1}{\mathcal{B}_l} \sum_{i=1}^{\mathcal{B}_l} \left[ \mathcal{L}_s(p_i^A, y_i) + \mathcal{L}_s(p_i^B, y_i) \right], \tag{19}$$

where $\mathcal{L}_s = \frac{1}{2}[\mathcal{L}_{\text{Dice}} + \mathcal{L}_{\text{ce}}]$, $\mathcal{L}_{\text{Dice}}$ and $\mathcal{L}_{\text{ce}}$ represent the Dice and cross-entropy losses, respectively. $p_i^{(\cdot)}$ is the output probability map. For all data, the ECPS loss and the contrastive loss are employed. Our optimization target is to minimize the overall loss, which can be formulated as:

$$\begin{aligned} \mathcal{L}_{\text{total}} &= \mathcal{L}_{\text{sup}} + \lambda_u \mathcal{L}_{\text{ecps}} + \lambda_c \mathcal{L}_{\text{contra}}, \\ \mathcal{L}_{\text{contra}} &= \mathcal{L}_{hqc} + \beta \mathcal{L}_{\text{dvcl}} \end{aligned} \tag{20}$$

where $\mathcal{L}_{\text{sup}}$, $\mathcal{L}_{\text{ecps}}$, $\mathcal{L}_{hqc}$, and $\mathcal{L}_{\text{dvcl}}$ denote the supervised, ECPS, HqC, and DVCL losses, respectively. Additionally, $\lambda_u$ and $\lambda_c$ are the hyperparameters to control the contribution of each loss term. $\lambda_u$ is set to 0.1 and used the CPS weight ramp-up functionYu et al. (2019). The value of $\lambda_c$ is set to 0.1. The scalar $\beta = 0.3$ is used to balance the two loss functions. The training procedure is presented in Algorithm 1.

## C  DATASETS

**FLARE 2022.** This dataset is from the MICCAI 2022 Challenge Fast and Low GPU memory Abdominal Organ Segmentation (Ma et al., 2023b) and comprises 70 3D CT volumes accompanied by a corresponding ground-truth mask. In addition, it includes 2000 unlabeled 3D CT volumes. The dataset is designed to segment 13 abdominal organs: the liver (Liv), spleen (Spl), pancreas (Pan), right kidney (R.kid), left kidney (L.kid), stomach (Sto), gallbladder (Gal), esophagus (Eso), aorta (Aor), inferior vena cava (IVC), right adrenal gland (RAG), left adrenal gland (LAG), and duodenum (Duo). For dataset partitioning, we divide the labeled data into training, validation, and test sets at a ratio of 6:2:2. In addition, we incorporate the unlabeled data into the training set, resulting in labeled data proportions of 50% (42 labeled cases and 42 unlabeled cases) and 10% (42 labeled cases and 378 unlabeled cases) within two training sets.

**AMOS.** This dataset originates from the Multi-Modality Abdominal Multi-Organ Segmentation Challenge 2022 (Ji et al., 2022). The AMOS dataset is comprised of 300 CT images, which are annotated at the pixel level for 15 distinct abdominal organs, including two additional organs not

---

**Algorithm 1: Training Procedure of Our Framework**

---

**input** : Segmentation networks $\{f(\cdot; \theta_i)\}$, maximum epoch $E_{\max}$, batchsize $\mathcal{B}$, labeled training data and their ground-truth masks $\mathcal{S}_l = \{(x_i, y_i)_{i=1}^N\}$, unlabeled training data $\mathcal{S}_u = \{(x_i)_{i=1}^M\}$.

**output:** Trained weights of $model\ A\ f(\cdot; \theta_A)$ and $model\ B\ f(\cdot; \theta_B)$;

1   Initialize network parameters of $model\ A\ f(\cdot; \theta_A)$ and $model\ B\ f(\cdot; \theta_B)$;

2   **for** $epoch \in [1, E_{\max}]$ **do**

3     **for** $batch\ \mathcal{B}$ **do**

4        Get probabilities: $p_i^A \leftarrow$ segmentation head $A(x_i), p_i^B \leftarrow$ segmentation head $B(x_i)$;

5        Get features: $\mathbf{r}_i^A \leftarrow$ feature head $A(x_i), \mathbf{r}_i^B \leftarrow$ feature head $B(x_i)$;

6        **for** $x_i \in \mathcal{S}_l$ **do**

7           Calculate $\mathcal{L}_{\text{sup}} \leftarrow \{p_i^1, p_i^2, y_i\}$ via Eq.19 ; // Compute supervised loss

8        **end**

9        **for** $x_i \in \mathcal{S}_u$ **do**

10          Get pseudo-labels: $\hat{y}_i^A \leftarrow \arg\max(p_i^A), \hat{y}_i^B \leftarrow \arg\max(p_i^B)$;

11          Get reliable pseudo-labels $M_r^A(x_i)$ and $M_r^B(x_i)$ via Eq.2;

12          Get unreliable pseudo-labels $M_u^A(x_i)$ and $M_u^B(x_i)$ via Eq.2;

13          Get weights $\mathcal{W}^1$ and $\mathcal{W}^2$ via Eq.4;

14          Calculate $\mathcal{L}_{\text{ecps}} \leftarrow \{\hat{y}_i^1, \hat{y}_i^2, \mathcal{W}^1, \mathcal{W}^2\}$via Eq.5; // Compute ECPS loss

15        **end**

16        Initialize $\mathcal{L}_{contra} \leftarrow 0, \mathcal{L}_{\text{hqc}} \leftarrow 0, \mathcal{L}_{\text{dvcl}} \leftarrow 0$;

17        **for** $x_i \in \mathcal{S}_l \cup \mathcal{S}_u$ **do**

18          **for** $x_i \in \mathcal{S}_l \cup M_r^A(x_i) \cup M_r^B(x_i)$ **do**

19             **for** $c \leftarrow 1$ **to** $C$ **do**

20                Get feature sets of anchor voxels $\mathcal{R}_c$ via Eq.8;

21                Random sample $M$ anchor voxels: $\{\mathbf{r}_m\}_{m=1}^M \in \mathcal{R}_c$;

22                Get positive candidates $\mathcal{P}_c$ via Eq.9;

23                Get feature sets of negative candidates $\mathcal{G}_c = \mathcal{G}_c^{Al} \bigcup \mathcal{G}_c^{Bl}$ via Eq.10;

24                Random sample $N$ negative candidates: $\{\mathbf{r}_n\}_{n=1}^N \in \mathcal{G}_c$;

25                Calculate $\mathcal{L} \leftarrow \{\{\mathbf{r}_m\}_{m=1}^M, \{\mathbf{r}_n\}_{n=1}^N, \mathcal{P}_c\}$ ; // Compute HqC loss

26                $\mathcal{L}_{\text{hqc}} \leftarrow \mathcal{L}_{\text{hqc}} + \mathcal{L}$;

27             **end**

28          **end**

29          **for** $x_i \in M_u^A(x_i) \cup M_u^B(x_i)$ **do**

30             Get $\{\mathcal{O}_{i,j}\}_{j=1}^{|M_u^A(x_i) \cup M_u^B(x_i)-1|}$ via Eq.12;

31             Get the close neighbor set $\mathcal{C}_i$ via Eq.13;

32             Get the distant outsider set $\mathcal{D}_i$ via Eq.13;

33             Calculate $\mathcal{L}_{\text{dvcl}} \leftarrow \{\mathbf{r}_i, \mathcal{C}_i, \mathcal{D}_i\}$ via Eq.17; // Compute DVCL loss

34          **end**

35        **end**

36        Calculate $\mathcal{L}_{\text{contra}} \leftarrow \mathcal{L}_{\text{hqc}} + \beta\mathcal{L}_{\text{dvcl}}$ via Eq.18 ;

37        Update $model\ A\ f(\cdot; \theta_A)$ and $model\ B\ f(\cdot; \theta_B)$ by decreasing the stochastic gradient on $\mathcal{L}_{\text{sup}}$, $\mathcal{L}_{\text{ecps}}$, and $\mathcal{L}_{\text{contra}}$ via Eq.20 ;

38     **end**

39     $epoch = epoch + 1$;

40   **end**

---

found in the FLARE 2022 dataset: the bladder (Bl) and prostate/uterus (P/U). In our experiments, we divide the dataset into training, validation, and test sets using a ratio of 6:2:2. For the training set, the proportion of labeled data is set to 10% and 50%.

**BTCV.** This dataset is from MICCAI Multi-Atlas Labeling Beyond Cranial Vault-Workshop Challenge Landman et al. (2015) which contains 30 CT images with 13 organs annotation. In contrast to the FLARE 2022 dataset, the Duo has been replaced with the portal vein and splenic vein (P&S). Each CT scan was acquired with contrast enhancement in portal venous phase and consists of 80 to 225 slices with 512×512 pixels and slice thickness ranging from 1 to 6 mm. In our experiments, we divide the dataset into training, validation, and test sets using a ratio of 6:2:2. For the training set, the proportion of labeled data is set to 10% and 50%.

**MMWHS.** This dataset is from the Multi-Modality Whole Heart Segmentation Challenge 2017 (Zhuang & Shen, 2016) . The dataset contains 20 3D CT volumes with corresponding annotations of seven cardiac structures, *i.e.,* the myocardium of the left ventricle (MYO), left atrium blood cavity (LAC), right atrium blood cavity (RAC), left ventricle blood cavity (LVC), right ventricle blood cavity (RVC), ascending aorta (AA), and pulmonary aorta (PA). In our experiments, we divide the dataset into training, validation, and test sets at a ratio of 6:2:2. For the training set, the proportion of labeled data is set to 10%and 50%.

## D  DATA PREPROCESSING AND IMPLEMENTATION DETAILS

**Data Preprocessing.** We apply hierarchical steps for data preprocessing before network training. 1) the orientation of all CT scans is standardized in the left-posterior-inferior (LPI) direction. 2) The voxel values are clipped into the range of [–325, 325] Hounsfield units (HU) to enhance the contrast of the foreground organs and suppress background interference. 3) The voxel spacing is standardized to [1.25, 1.25, 2.5]. 4) Min-max normalization is implemented using the formula $(x - x_{0.5})/(x_{99.5} - x_{0.5})$, where $x_{0.5}$ and $x_{99.5}$ represent the 0.5th and 99.5th percentiles of $x$, respectively.

**Implementation Details.** All experiments are conducted using the PyTorch platform and trained on 2 NVIDIA A100 GPUs. For network training, we use the stochastic gradient descent (SGD) optimizer with a weight decay of 0.0001 and momentum of 0.9 (Luo et al., 2021b). To mitigate the overfitting problem, we apply basic data augmentation, including random cropping and flipping (Wang & Li, 2023). The model uses randomly cropped patches as inputs, with a patch size of $64 \times 128 \times 128$. During network training, we adopt a polynomial learning rate policy that scales the initial learning rate by $(1 - \frac{iteration}{max\_iteration})^{0.9}$ (Chen et al., 2023a), where *iteration* and *max_iteration* represent the current iteration and maximum number of iterations, respectively. For the MMWHS dataset, the segmentation model is trained with a batch size of 4, comprising an equal split of 2 labeled and 2 unlabeled instances, over a total of 10,000 iterations with an initial learning rate of 0.03. For the FLARE 2022, AMOS and BTCV datasets, we opt for a batch size of 8, including 4 labeled and 4 unlabeled instances, and conduct training for 20,000 iterations with an initial learning rate of 0.1. Note that, for all the evaluated methods, we make no additional modifications during the training process for fair evaluations. During inference, the final volumetric segmentation is generated using a sliding-window strategy. The stride used is $32 \times 80 \times 80$, and the sliding-window approach employs a patch size of $64 \times 160 \times 160$.

## E  SEMI-SUPERVISED SEGMENTATION METHODS IN COMPARISON

For experiments, we compare our method to thirteen state-of-the-art semi-supervised segmentation methods: (1) deep adversarial networks (DAN) (Zhang et al., 2017), (2) MT (Tarvainen & Valpola, 2017), (3) uncertainty-aware mean teacher (UA-MT) (Yu et al., 2019), (4) the shape-aware semi-supervised network (SASSnet) (Li et al., 2020), (5) the dual-task consistency (DTC) framework (Luo et al., 2021a), (6) CPS (Chen et al., 2021), (7) calibrating label distribution (CLD) for segmentation (Lin et al., 2022), (8) the dual-debiased heterogeneous co-training (DHC) framework (Wang & Li, 2023), (9) MagicNet (Chen et al., 2023a), (10) Uncertainty-guided pixel contrastive learning (UGPCL) framework  (Chen et al., 2021), (11) Using unreliable pseudo-labels (U$^2$PL) framework (Wang et al., 2022c), (12) Balanced feature-level contrastive learning (BaCon) framework  (Feng et al., 2024), (13) Contrastive complementary labeling (CCL) framework (Deng et al., 2024), and the fully supervised 3D U-Net (Çiçek et al., 2016). Note that among all the above evaluated methods, several methods use a contrastive learning objective, including UGPCL (Chen et al., 2021), U$^2$PL (Wang et al., 2022c), BaCon (Feng et al., 2024), and CCL (Deng et al., 2024). For all semi-supervised methods, we utilize 3D U-Net as the backbone.

## F  MORE EXPERIMENT RESULTS ON AMOS.

Table.5 shows the qualitative results, where our method provides better segmentation performance compared to all other approaches. This clearly demonstrates the superiority of our model.

Table 5: Quantitative results on two settings of AMOS dataset. 'w/o VCL' or 'w/ VCL' indicates whether the SSL methods combined with VCL or not. Best results are boldfaced.

| | Methods | Liv | Sto | Spl | L.kid | R.kid | Aor | Bla | IVC | Pan | Duo | P/U | Gal | Eso | RAG | LAG | Mean Dice | Mean Jaccard |
|---|---|---|---|---|---|---|---|---|---|---|---|---|---|---|---|---|---|---|
| | *10% labeled data (labeled:unlabeled=18:162)* | | | | | | | | | | | | | | | | | |
| w/o VCL | 3D U-Net (Çiçek et al., 2016) | 85.99 | 40.08 | 82.19 | 79.57 | 80.24 | 75.51 | 11.63 | 59.86 | 26.80 | 20.22 | 22.40 | 15.06 | 37.05 | 32.46 | 9.68 | 45.24±0.65 | 39.06±0.42 |
| | DAN (Zhang et al., 2017) | 86.54 | 50.00 | 83.98 | 86.97 | 85.76 | 85.46 | 54.81 | 67.26 | 48.97 | 43.84 | 52.23 | 33.08 | 40.28 | 28.00 | 18.09 | 57.80±0.88 | 47.38±0.89 |
| | MT (Tarvainen & Valpola, 2017) | 89.26 | 56.87 | 84.27 | 84.42 | 85.98 | 85.57 | 51.22 | 70.13 | 48.91 | 48.04 | 39.46 | 42.17 | 50.71 | 43.92 | 30.46 | 61.44±1.28 | 51.00±1.10 |
| | UA-MT (Yu et al., 2019) | 88.25 | 52.49 | 86.39 | 86.34 | 87.73 | 86.14 | 68.22 | 70.76 | 47.19 | 42.79 | 49.54 | 32.49 | 52.87 | 43.98 | 37.76 | 61.73±1.12 | 50.97±0.90 |
| | SASSnet (Li et al., 2020) | 90.33 | 48.61 | 86.87 | 87.66 | **88.17** | 87.09 | 43.55 | 73.85 | 50.05 | 55.85 | 65.61 | 36.15 | 43.18 | 41.36 | 28.85 | 58.35±1.42 | 51.15±0.83 |
| | DTC (Luo et al., 2021a) | 89.81 | 50.49 | 87.48 | 85.20 | 85.84 | 85.83 | 64.49 | 72.72 | 43.44 | 47.36 | 39.19 | 38.62 | 50.56 | 42.53 | 37.02 | 60.81±1.27 | 50.84±1.24 |
| | CPS (Chen et al., 2021) | 88.52 | 55.52 | 83.25 | 86.30 | 87.97 | 85.36 | 60.53 | 71.71 | 50.11 | 46.05 | 60.33 | 37.95 | 52.37 | 46.33 | 37.48 | 63.52±0.36 | 51.82±0.49 |
| | CLD (Lin et al., 2022) | 88.43 | 63.71 | 84.90 | 85.85 | 86.07 | 85.16 | 64.15 | 75.56 | 55.21 | 49.67 | 60.62 | 39.47 | 56.71 | 50.91 | 40.56 | 65.81±1.24 | 54.00±1.69 |
| | DHC (Wang & Li, 2023) | 83.27 | 63.39 | 83.60 | 84.11 | 85.66 | 84.40 | **74.52** | 74.88 | 56.02 | 51.89 | 65.47 | 47.53 | 43.21 | 48.28 | 42.59 | 65.17±1.47 | 52.46±1.30 |
| | MagicNet (Chen et al., 2023a) | 88.99 | 61.20 | 83.52 | **88.39** | 87.24 | 83.69 | 62.47 | 74.83 | 54.11 | 51.18 | 54.62 | 56.69 | 55.68 | 46.87 | 43.16 | 65.31±1.31 | 54.89±0.78 |
| w/ VCL | UGPCL (Chen et al., 2021) | 90.31 | 60.14 | 86.78 | 86.62 | 87.87 | 86.69 | 56.14 | 71.14 | 36.27 | 44.16 | 46.04 | 38.21 | 48.82 | 38.20 | 35.09 | 61.48±1.02 | 50.99±1.61 |
| | U²PL (Wang et al., 2022c) | 90.23 | 54.80 | 85.80 | 87.97 | 89.10 | 87.95 | 59.31 | 74.39 | 52.07 | 51.45 | 54.92 | 36.64 | 56.67 | 49.77 | 44.54 | 64.73±1.35 | 54.46±1.01 |
| | BaCon (Feng et al., 2024) | 89.84 | 56.57 | 86.36 | 89.41 | 88.78 | 88.35 | 38.96 | 73.62 | 50.98 | 44.82 | 50.47 | 43.36 | 58.67 | 50.81 | 42.81 | 64.40±1.50 | 54.61±0.72 |
| | CCL (Deng et al., 2024) | 90.58 | 55.51 | 85.13 | 88.01 | 89.11 | 88.33 | 62.34 | 73.72 | 50.73 | 45.13 | 58.34 | 40.11 | 60.09 | 45.33 | 33.20 | 64.00±0.16 | 53.33±0.70 |
| | **Ours** | **90.76** | **70.37** | **89.62** | **89.74** | 89.21 | **88.48** | 72.14 | **77.59** | 55.85 | 55.20 | 63.79 | 55.58 | 47.14 | | | **71.09±0.51** | **59.34±0.46** |
| | *50% labeled data (labeled:unlabeled=90:90)* | | | | | | | | | | | | | | | | | |
| w/o VCL | 3D U-Net (Çiçek et al., 2016) | 89.25 | 55.60 | 84.23 | 87.40 | 88.58 | 87.32 | 53.49 | 73.71 | 48.56 | 48.21 | 52.68 | 38.43 | 50.27 | 38.48 | 30.30 | 61.29±1.74 | 51.62±1.35 |
| | DAN (Zhang et al., 2017) | 90.49 | 55.91 | 89.63 | 90.08 | 88.74 | 86.71 | 47.44 | 72.09 | 54.98 | 50.33 | 53.04 | 39.13 | 58.34 | 29.57 | 6.49 | 61.39±1.16 | 52.06±1.45 |
| | MT (Tarvainen & Valpola, 2017) | 92.08 | 62.02 | 89.83 | 90.23 | 89.24 | 89.12 | 63.05 | 78.11 | 53.46 | 52.85 | 40.93 | 51.63 | 59.64 | 45.41 | 37.35 | 66.17±0.75 | 57.06±1.00 |
| | UA-MT (Yu et al., 2019) | 90.86 | 58.55 | 88.92 | 88.93 | 88.83 | 88.49 | 54.86 | 74.28 | 51.88 | 54.54 | 54.67 | 40.99 | 58.58 | 51.31 | 41.78 | 65.48±0.80 | 55.62±1.10 |
| | SASSnet (Li et al., 2020) | 91.65 | 53.00 | **91.54** | 89.61 | 89.72 | 88.50 | 50.43 | 74.87 | 46.34 | 52.48 | 55.92 | 37.93 | 60.57 | 45.62 | 39.17 | 63.77±1.13 | 54.68±0.55 |
| | DTC (Luo et al., 2021a) | 91.25 | 56.49 | 90.68 | 88.88 | 89.30 | 89.16 | 67.37 | 76.50 | 48.13 | 54.67 | 54.23 | 41.88 | 62.49 | 47.67 | 42.91 | 66.93±1.78 | 55.92±1.78 |
| | CPS (Chen et al., 2021) | 90.94 | 61.90 | 89.97 | 90.25 | 89.67 | 88.77 | 65.03 | 75.27 | 52.34 | 45.15 | 54.76 | 42.87 | 62.44 | 49.96 | 47.74 | 66.65±1.24 | 56.56±0.54 |
| | CLD (Lin et al., 2022) | 91.23 | 66.18 | 89.34 | 89.50 | **89.86** | 88.85 | 66.40 | 76.97 | 55.63 | 53.35 | 58.82 | 45.78 | 62.93 | 54.24 | 43.79 | 69.09±1.14 | 57.99±1.14 |
| | DHC (Wang & Li, 2023) | 86.68 | 58.39 | 86.62 | 85.57 | 87.48 | 87.28 | 67.04 | 74.38 | 60.88 | 56.91 | 58.87 | 53.75 | 54.14 | 51.59 | 51.03 | 68.60±0.56 | 56.05±0.51 |
| | MagicNet (Chen et al., 2023a) | 91.69 | 66.33 | 88.59 | 90.28 | 89.64 | 86.80 | 61.80 | 74.39 | 59.94 | 52.88 | 57.28 | **58.83** | 59.53 | 52.74 | 42.35 | 68.94±0.56 | 58.33±0.52 |
| w/ VCL | UGPCL (Chen et al., 2021) | 90.84 | 66.10 | 90.04 | 89.86 | 89.92 | 89.09 | 72.45 | 76.95 | 56.10 | 57.52 | 60.73 | 53.87 | 65.19 | 51.67 | 48.38 | 70.71±0.17 | 59.42±0.31 |
| | U²PL (Wang et al., 2022c) | 86.09 | 62.61 | 87.08 | 87.18 | 87.48 | 87.35 | 73.57 | 76.57 | 60.32 | 56.67 | 65.75 | 58.30 | 61.75 | 55.24 | 43.55 | 69.97±0.25 | 57.42±0.20 |
| | BaCon (Feng et al., 2024) | 88.51 | 64.40 | 88.78 | 88.58 | 89.22 | 88.27 | 73.17 | 77.69 | 54.44 | 55.58 | 44.46 | 53.14 | 62.29 | 51.34 | **51.15** | 70.07±0.35 | 57.94±0.53 |
| | CCL (Deng et al., 2024) | 88.70 | 67.89 | 88.44 | 88.80 | 89.48 | 88.93 | **75.28** | 76.96 | 61.22 | 56.86 | 64.47 | 51.07 | 62.65 | 37.07 | 34.13 | 68.80±1.65 | 57.28±1.02 |
| | **Ours** | **92.23** | **73.23** | 91.14 | **90.41** | 89.76 | **89.32** | 71.87 | **78.90** | **61.26** | 57.97 | **71.24** | 55.42 | **66.57** | 55.34 | 49.25 | **72.84±0.23** | **61.38±0.23** |

Table 6: Quantitative results on two settings of MMWHS dataset. 'w/o VCL' or 'w/ VCL' indicates whether the SSL methods combined with CL or not. Best results are boldfaced.

| | Methods | MYO | LAC | RAC | LVC | RVC | AA | PA | Mean Dice | Mean Jaccard |
|---|---|---|---|---|---|---|---|---|---|---|
| | *10% labeled data (labeled:unlabeled=2:10)* | | | | | | | | | |
| w/o VCL | DAN (Zhang et al., 2017) | 89.19 | 69.26 | 80.65 | 68.50 | 84.23 | 83.83 | 44.57 | 74.32±0.84 | 61.77±1.53 |
| | MT (Tarvainen & Valpola, 2017) | 87.83 | 77.09 | 81.65 | 77.16 | 84.78 | 90.88 | 69.46 | 81.27±0.57 | 69.49±0.65 |
| | UA-MT (Yu et al., 2019) | 89.50 | 77.67 | 84.84 | 78.30 | 86.10 | 92.02 | 69.35 | 82.54±0.84 | 71.28±0.97 |
| | SASSnet (Li et al., 2020) | 89.66 | 77.62 | 85.61 | **80.92** | 85.43 | 90.99 | 59.20 | 81.35±0.45 | 70.15±0.34 |
| | DTC (Luo et al., 2021a) | 89.29 | 77.83 | 84.94 | 78.70 | 86.24 | 90.23 | 71.74 | 82.71±0.50 | 71.32±0.57 |
| | CPS (Chen et al., 2021) | 90.10 | 79.00 | 85.45 | 80.84 | 87.98 | 90.71 | 70.82 | 83.56±0.22 | 72.63±0.40 |
| | CLD (Lin et al., 2022) | 90.82 | 80.88 | 88.70 | 78.94 | 87.22 | 92.49 | 70.48 | 84.22±0.80 | 73.84±0.85 |
| | DHC (Wang & Li, 2023) | 90.40 | 77.14 | 85.87 | 78.64 | **88.91** | 89.99 | 74.71 | 83.67±0.80 | 72.83±1.13 |
| | MagicNet(Chen et al., 2023a) | 87.99 | 74.14 | 85.03 | 75.38 | 83.93 | 90.03 | 59.51 | 79.43±0.67 | 67.56±1.11 |
| w/ VCL | UGPCL (Chen et al., 2021) | 90.70 | 78.41 | 84.58 | 78.33 | 86.86 | 91.81 | 77.05 | 83.96±0.36 | 73.12±0.48 |
| | U²PL (Wang et al., 2022c) | 90.16 | 80.52 | 87.50 | 77.29 | 88.38 | 93.00 | 73.41 | 84.32±0.18 | 73.70±0.26 |
| | BaCon (Feng et al., 2024) | 90.80 | 79.82 | 85.64 | 80.49 | 87.21 | 91.76 | 73.54 | 84.18±0.12 | 73.48±0.11 |
| | CCL (Deng et al., 2024) | 90.61 | 80.04 | 87.41 | 80.87 | 87.25 | 92.16 | 68.11 | 83.78±0.31 | 73.16±0.35 |
| | **Ours** | **91.21** | **80.98** | **88.80** | 80.77 | 88.35 | **93.25** | **80.92** | **86.33±0.28** | **76.46±0.45** |
| | *50% labeled data (labeled:unlabeled=6:6)* | | | | | | | | | |
| w/o VCL | DAN (Zhang et al., 2017) | 90.46 | 82.08 | 87.21 | 77.20 | 87.64 | 89.35 | 73.16 | 83.87±0.80 | 73.11±1.10 |
| | MT (Tarvainen & Valpola, 2017) | 90.57 | 84.49 | 88.27 | 80.82 | 88.31 | 92.68 | 77.65 | 86.11±0.20 | 76.18±0.15 |
| | UA-MT (Yu et al., 2019) | 90.94 | 84.59 | 88.47 | 82.27 | 88.36 | 93.12 | 77.30 | 86.43±0.07 | 76.71±0.19 |
| | SASSnet (Li et al., 2020) | 88.73 | 79.02 | 85.89 | 79.30 | 85.60 | 91.14 | 71.59 | 83.04±0.40 | 71.82±0.60 |
| | DTC (Luo et al., 2021a) | 91.20 | 84.23 | 88.03 | 80.95 | 88.36 | 90.79 | 77.35 | 85.84±0.22 | 75.81±0.39 |
| | CPS (Chen et al., 2021) | **91.42** | 84.40 | 88.90 | 82.10 | **88.93** | 92.99 | 76.89 | 86.52±0.15 | 76.84±0.15 |
| | CLD (Lin et al., 2022) | 91.12 | 85.49 | 90.54 | 79.98 | 88.92 | 91.37 | 80.63 | 86.87±0.33 | 77.29±0.46 |
| | DHC (Wang & Li, 2023) | 90.75 | 84.93 | 89.34 | 81.83 | 88.69 | 93.72 | 79.23 | 86.93±0.14 | 77.43±0.23 |
| | MagicNet(Chen et al., 2023a) | 91.27 | 83.58 | 87.75 | 79.07 | 89.21 | 90.12 | 66.07 | 83.87±0.65 | 73.52±0.57 |
| w/ VCL | UGPCL (Chen et al., 2021) | 91.76 | 85.35 | 89.56 | 80.13 | 88.04 | 93.36 | 76.89 | 86.44±0.47 | 76.72±0.67 |
| | U²PL (Wang et al., 2022c) | 91.07 | 85.99 | 90.81 | 81.91 | 88.14 | 93.66 | 75.17 | 86.68±0.07 | 77.10±0.11 |
| | BaCon (Feng et al., 2024) | 91.26 | 84.91 | 89.42 | 82.12 | 88.05 | 93.13 | 74.98 | 86.27±0.10 | 76.44±0.14 |
| | CCL (Deng et al., 2024) | 91.02 | 84.44 | 87.51 | 81.43 | 87.60 | 89.76 | 77.54 | 85.62±0.66 | 75.31±0.93 |
| | **Ours** | 91.31 | **86.19** | **91.00** | **82.58** | 88.37 | **94.30** | **82.03** | **87.97±0.03** | **78.92±0.06** |

# G   MORE EXPERIMENT RESULTS ON MMWHS.

Fig.7 and Table.6 show the qualitative results, where our method provides better segmentation performance compared to all other approaches. This clearly demonstrates the superiority of our model.

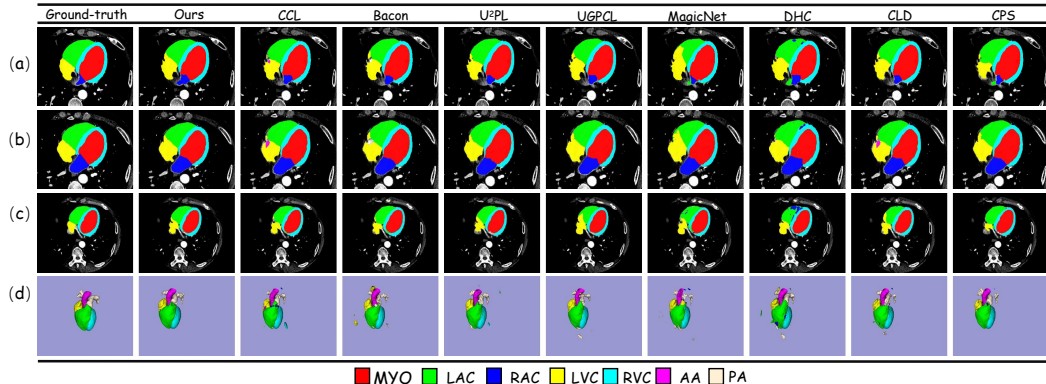

Figure 7: Visualization of the segmentation results for the BTCV dataset. (a-c) Segmentation results for one case of three transverse sections and (d) 3D segmentation views.

Table 7: Quantitative results on two settings of BTCV dataset. 'w/o VCL' or 'w/ VCL' indicates whether the SSL methods combined with CL or not. Best results are boldfaced.

| | Methods | Spl | R.kid | L.kid | Gal | Eso | Liv | Sto | Aor | IVC | P&S | Pan | RAG | LAG | Mean Dice | Mean Jaccard |
|---|---|---|---|---|---|---|---|---|---|---|---|---|---|---|---|---|
| | *10% labeled data (labeled:unlabeled=2:12)* | | | | | | | | | | | | | | | |
| w/o VCL | 3D U-Net (Çiçek et al., 2016) | 0.25 | 31.44 | 30.99 | 0.00 | 0.00 | 47.78 | 3.76 | 41.07 | 9.94 | 3.67 | 0.41 | 0.00 | 0.02 | 13.03±0.15 | 8.29±0.10 |
| | DAN (Zhang et al., 2017) | 59.98 | 82.65 | 77.08 | 4.36 | 0.71 | 80.22 | 8.69 | 46.88 | 60.62 | 30.47 | 11.36 | 18.65 | 8.30 | 37.69±1.48 | 29.06±1.11 |
| | MT (Tarvainen & Valpola, 2017) | 64.00 | 82.25 | 71.56 | 16.80 | 27.09 | 85.93 | 8.64 | 80.29 | 60.62 | 37.63 | 29.79 | 14.90 | 7.52 | 45.16±1.94 | 35.20±1.42 |
| | UA-MT (Yu et al., 2019) | 75.46 | 80.28 | 73.22 | 11.95 | 14.47 | 84.74 | 9.89 | 79.79 | 70.66 | 37.14 | 20.86 | 14.36 | 11.39 | 44.94±1.17 | 35.69±0.84 |
| | SASSnet (Li et al., 2020) | 74.17 | 80.94 | 80.02 | 16.25 | 6.37 | 85.70 | 6.28 | 77.26 | 69.04 | 40.79 | 32.16 | 29.83 | 1.67 | 46.19±1.09 | 36.98±0.88 |
| | DTC (Luo et al., 2021a) | 76.05 | 78.89 | 78.16 | 10.25 | 35.50 | 86.14 | 7.05 | 84.24 | 70.72 | 32.19 | 22.22 | 36.71 | 3.30 | 47.80±1.42 | 38.19±1.10 |
| | CPS (Chen et al., 2021) | 75.56 | 77.34 | 77.33 | 8.90 | 18.14 | 86.16 | 13.48 | 82.98 | 67.81 | 43.29 | 21.24 | 31.48 | 15.97 | 47.67±0.76 | 37.86±0.41 |
| | CLD (Lin et al., 2022) | 69.84 | 82.48 | 71.53 | 23.56 | 9.92 | 86.14 | 14.41 | 83.15 | 69.04 | 43.71 | 31.03 | 38.94 | 13.13 | 48.99±0.92 | 38.73±0.81 |
| | DHC (Wang & Li, 2023) | 72.12 | 80.80 | **80.79** | 16.38 | 24.63 | 85.61 | **15.29** | 83.02 | 62.29 | 43.46 | 29.70 | 35.78 | 7.65 | 49.04±0.35 | 38.68±0.17 |
| | MagicNet (Chen et al., 2023a) | 71.86 | 82.00 | 79.35 | 16.30 | 34.69 | 86.18 | 12.00 | 83.93 | 64.56 | 42.66 | **30.60** | 37.61 | 10.26 | 50.15±0.58 | 39.50±0.54 |
| w/ VCL | UGPCL (Chen et al., 2021) | 75.43 | 81.72 | 78.66 | 13.43 | 6.89 | 87.01 | 9.04 | 84.46 | 72.62 | 34.85 | 28.03 | 29.84 | 3.73 | 46.59±1.29 | 37.72±1.06 |
| | U²PL (Wang et al., 2022c) | 72.48 | 80.69 | 79.94 | 10.21 | 30.93 | 86.93 | 12.96 | 84.98 | 64.67 | 40.14 | 24.53 | 28.54 | 12.53 | 48.42±0.68 | 38.44±0.23 |
| | BaCon (Feng et al., 2024) | 76.02 | 82.64 | 74.78 | 15.67 | 22.33 | 84.35 | 9.73 | 82.58 | 69.35 | **44.20** | 25.56 | 32.21 | 3.26 | 47.90±1.50 | 38.20±1.12 |
| | CCL (Deng et al., 2024) | **76.20** | 80.37 | 77.95 | 13.63 | 29.25 | 85.58 | 12.22 | 81.36 | 65.88 | 37.94 | 29.18 | 26.14 | 8.08 | 47.98±1.43 | 37.84±1.08 |
| | **Ours** | 74.70 | **83.40** | 73.42 | **24.74** | **46.93** | **87.07** | 11.65 | **85.79** | **74.03** | 41.74 | 26.37 | **40.25** | **14.74** | **52.68±0.39** | **41.95±0.75** |
| | *50% labeled data (labeled:unlabeled=12:12)* | | | | | | | | | | | | | | | |
| w/o VCL | 3D U-Net (Çiçek et al., 2016) | 61.26 | 75.08 | 71.21 | 8.08 | 24.06 | 81.37 | 10.69 | 78.87 | 55.11 | 32.56 | 11.04 | 2.66 | 6.96 | 39.92±1.17 | 30.70±1.23 |
| | DAN (Zhang et al., 2017) | 74.07 | 88.8 | 88.56 | 20.19 | 26.10 | 85.69 | 36.81 | 78.79 | 59.77 | 52.95 | 50.12 | 48.31 | 40.16 | 57.72±1.21 | 45.94±1.15 |
| | MT (Tarvainen & Valpola, 2017) | 77.10 | 88.55 | 87.19 | 11.92 | 66.83 | 86.01 | 31.71 | 87.84 | 77.10 | 47.28 | 42.45 | 36.18 | 41.94 | 60.16±0.95 | 48.98±0.46 |
| | UA-MT (Yu et al., 2019) | 75.34 | 86.48 | 87.71 | 12.40 | 57.77 | 87.63 | 28.83 | 88.30 | 75.48 | 44.36 | 37.65 | 38.01 | 33.42 | 57.95±0.80 | 47.05±0.50 |
| | SASSnet (Li et al., 2020) | 76.48 | 90.28 | 87.96 | 12.01 | 56.46 | 87.64 | 35.56 | 89.29 | 78.34 | 51.25 | 51.73 | 42.97 | 44.87 | 61.91±0.12 | 50.75±0.12 |
| | DTC (Luo et al., 2021a) | 77.85 | 86.85 | 89.45 | 12.17 | 64.13 | **88.81** | 29.97 | 89.02 | 74.62 | 46.98 | 41.73 | 52.61 | 35.89 | 60.77±0.96 | 49.71±0.67 |
| | CPS (Chen et al., 2021) | 76.54 | 82.93 | 87.82 | 6.66 | 63.06 | 88.00 | 31.44 | 89.13 | 77.26 | 49.79 | 45.48 | 49.78 | 44.73 | 60.97±0.43 | 49.74±0.51 |
| | CLD (Lin et al., 2022) | 75.96 | 88.27 | 90.20 | 15.20 | 54.09 | 82.62 | 33.63 | 87.43 | 76.16 | 49.84 | 50.15 | **58.00** | 37.02 | 61.42±0.84 | 49.82±0.80 |
| | DHC (Wang & Li, 2023) | 76.65 | 88.71 | 87.19 | 10.00 | 69.03 | 80.67 | 33.06 | 86.43 | 73.24 | 48.67 | 47.97 | 53.97 | 40.26 | 61.22±1.57 | 49.43±1.42 |
| | MagicNet (Chen et al., 2023a) | 77.01 | 88.72 | 86.56 | 17.47 | 67.24 | 76.52 | 37.11 | 87.36 | 76.74 | 47.95 | 47.63 | 48.95 | 41.73 | 61.61±0.34 | 49.34±0.66 |
| w/ VCL | UGPCL (Chen et al., 2021) | 75.69 | 85.70 | 88.82 | 5.86 | 61.98 | 86.13 | 36.38 | 88.80 | 74.13 | 46.61 | 34.14 | 38.84 | 44.39 | 59.04±0.84 | 48.08±0.79 |
| | U²PL (Wang et al., 2022c) | 76.59 | 87.67 | 87.72 | 19.44 | 63.14 | 87.98 | 27.53 | **89.49** | 76.45 | 47.06 | 44.70 | 44.66 | 44.93 | 61.34±0.27 | 50.13±0.33 |
| | BaCon (Feng et al., 2024) | 75.91 | 88.17 | 87.04 | 11.15 | 66.55 | 87.77 | 34.52 | 88.14 | 75.76 | 45.18 | 37.69 | 52.99 | 43.18 | 61.08±0.04 | 49.71±0.10 |
| | CCL (Deng et al., 2024) | 77.14 | 87.77 | 89.53 | 5.81 | 67.04 | 86.75 | 31.91 | 88.54 | **78.35** | 43.50 | 36.69 | 53.75 | 38.41 | 60.40±0.89 | 49.40±0.86 |
| | **Ours** | **78.02** | **88.84** | **90.33** | **28.77** | 67.32 | 85.95 | **42.81** | 87.93 | 75.76 | **57.10** | **56.44** | 50.51 | **44.95** | **65.75±0.19** | **53.73±0.37** |

## H  MORE EXPERIMENT RESULTS ON BTCV.

Table.7 shows qualitative and quantitative results, where our method provides better segmentation performance compared to all other approaches. This clearly demonstrates the superiority of our model.

## I  ABLATION STUDY FOR HYPER-PARAMETERS.

To validate the robustness of our method, we implement ablation studies on hyper-parameters, including the variance factor $\alpha$, temperature coefficient $\tau$, number of negative candidates $N$, anchor voxels $M$, Scalar $\beta$, contrastive loss weight $\lambda_c$, and dimensionality of voxel-level features $\mathcal{V}$. The quantitative results for the different hyper-parameters are presented in Fig.8.

**Variance Factor $\alpha$.** In the entropy-based selection module, $\alpha$ is the variance factor influences the proportion of reliable voxels. To determine the value of $\alpha$, we train and test our method with

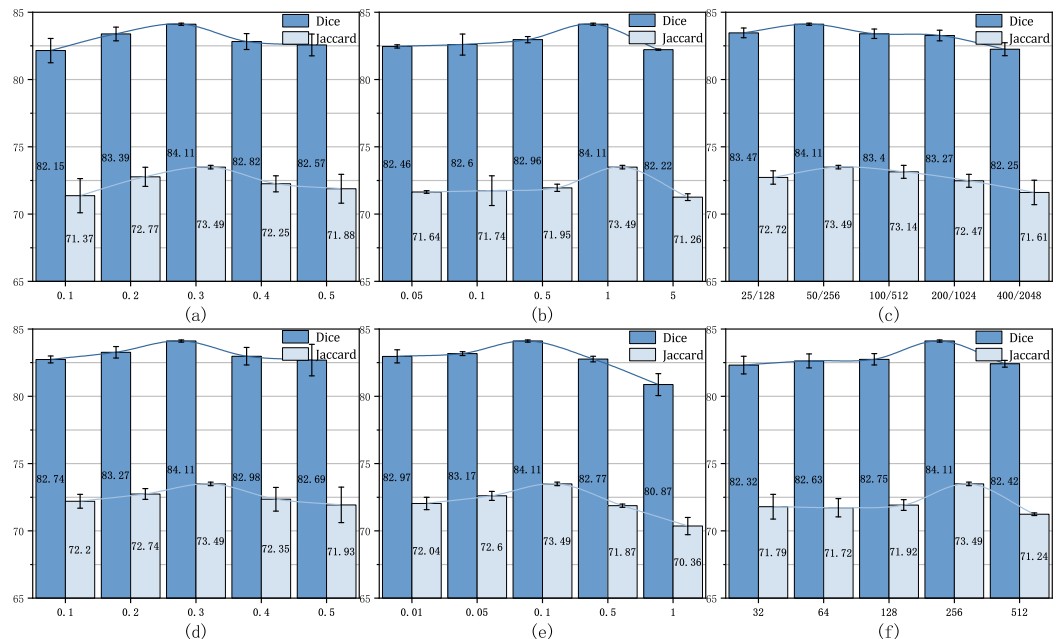

Figure 8: Quantitative comparisons of different hyper-parameters on FLARE 2022 dataset. (a-f) present mean Dice and mean Jaccard generated by the our method trained with different $\alpha, \tau, M/N, \beta, \lambda_c$, and $\mathcal{V}$, respectively.

different $\alpha$ values. As shown in Fig.8(a), our method trained with $\alpha = 0.5$ obtains the highest mean Dice and mean Jaccard.

**Temperature Coefficient $\tau$.** In the calculation of UVCL loss, $\tau$ plays a significant role in adjusting the emphasis on challenging samples. To evaluate the effect of different $\tau$ values, ablation experiments are performed using $\tau \in \{0.05, 0.1, 0.5, 1, 5\}$. The results (Fig.8(b)) indicate that $\tau = 1$ achieves optimal segmentation performance.

**Number of Negative Candidates $N$ and Anchor Voxels $M$.** $N$ and $M$ are used to calculate the HqC loss. As shown in Fig.8(c), the HqC loss trained with $N = 50$ and $M = 256$ achieves the best segmentation results.

**Scalar $\beta$.** Scalar $\beta$ is used to balance the HqC loss and the DVCL loss. As shown in Fig.8(d), $\beta = 0.3$ obtains the highest mean Dice and mean Jaccard.

**Contrastive Loss Weight $\lambda_c$.** $\lambda_c$ determines the contribution of contrastive loss to the total loss. As shown in Fig.8(e), $\lambda_c = 0.1$ obtains the highest mean Dice and mean Jaccard.

**Dimensionality of Voxel-level Features $\mathcal{V}$.** $\mathcal{V}$ determines the dimensionality of voxel-level features generated by the feature head. A small $\mathcal{V}$ may fail to capture sufficient information, whereas a large $\mathcal{V}$ may introduce noise. Considering this problem, experiments are conducted using $\mathcal{V} \in \{32, 64, 128, 256, 512\}$. The results (Fig.8(f)) demonstrate that $\mathcal{V} = 256$ achieves the optimal segmentation performance.

## J   THE COMPLETE DERIVATION OF THE FORMULA.

$$\psi(\mathcal{C}_m, \mathcal{D}_m) = -\log \frac{\mathcal{N}(\mathcal{C}_m \mid \mathbf{r}_m, \theta_\mathcal{S})}{\mathcal{N}(\mathcal{D}_m \mid \mathbf{r}_m, \theta_\mathcal{S})} = -\log \frac{\prod_{\mathbf{r}_n \in \mathcal{C}_m} \mathcal{O}_{m,n}}{\prod_{\mathbf{r}_k \in \mathcal{D}_m} \mathcal{O}_{m,k}} = -\log \prod_{\mathbf{r}_n \in \mathcal{C}_m} \mathcal{O}_{m,n} + \log \prod_{\mathbf{r}_k \in \mathcal{D}_m} \mathcal{O}_{m,k}$$

Since the logarithm of a product is equal to the sum of the logarithms, we have

$$-\log\prod_{\mathbf{r}_n\in\mathcal{C}_m}\mathcal{O}_{m,n}+\log\prod_{\mathbf{r}_k\in\mathcal{D}_m}\mathcal{O}_{m,k}=-\sum_{\mathbf{r}_n\in\mathcal{C}_m}\log(\mathcal{O}_{m,n})+\sum_{\mathbf{r}_k\in\mathcal{D}_m}\log(\mathcal{O}_{m,k})$$

$$=-\sum_{\mathbf{r}_n\in\mathcal{C}_m}\log(\frac{e^{p_m^T p_n}}{\sum_{\mathbf{r}_q\in U_{\mathcal{R}}}e^{p_m^T p_q}})+\sum_{\mathbf{r}_k\in\mathcal{D}_m}\log(\frac{e^{p_m^T p_k}}{\sum_{\mathbf{r}_q\in U_{\mathcal{R}}}e^{p_m^T p_q}})$$

$$=-\sum_{\mathbf{r}_n\in\mathcal{C}_m}[p_m^T p_n-\log(\sum_{\mathbf{r}_q\in U_{\mathcal{R}}}e^{p_m^T p_q})]+\sum_{\mathbf{r}_k\in\mathcal{D}_m}[p_m^T p_k-\log(\sum_{\mathbf{r}_q\in U_{\mathcal{R}}}e^{p_m^T p_q})] \quad (21)$$

$$=-\sum_{\mathbf{r}_n\in\mathcal{C}_m}p_m^T p_n+\sum_{\mathbf{r}_k\in\mathcal{D}_m}p_m^T p_k+(|\mathcal{C}_m|-|\mathcal{D}_m|)\log(\sum_{\mathbf{r}_q\in U_{\mathcal{R}}}e^{p_m^T p_q})$$

Consider the last term of Eq.21:

$$(|\mathcal{C}_m|-|\mathcal{D}_m|)\log(\sum_{\mathbf{r}_q\in U_{\mathcal{R}}}e^{p_m^T p_q})=(|\mathcal{C}_m|-|\mathcal{D}_m|)\log(|U_R|\cdot\frac{\sum_{\mathbf{r}_q\in U_{\mathcal{R}}}e^{p_m^T p_q}}{|U_R|})$$

$$=(|\mathcal{C}_m|-|\mathcal{D}_m|)(\log|U_R|+\log(\frac{\sum_{\mathbf{r}_q\in U_{\mathcal{R}}}e^{p_m^T p_q}}{|U_R|}))$$

Using *Jensen Inequality* $\log(\mathbb{E}[X])\geq\mathbb{E}[\log(X)]$, we have

$$(|\mathcal{C}_m|-|\mathcal{D}_m|)(\log|U_R|+\log(\frac{\sum_{\mathbf{r}_q\in U_{\mathcal{R}}}e^{p_m^T p_q}}{|U_R|}))\leq(|\mathcal{C}_m|-|\mathcal{D}_m|)(\log|U_{\mathcal{R}}|+\sum_{\mathbf{r}_q\in U_{\mathcal{R}}}\frac{p_m^T p_q}{|U_R|})$$

The expression $p_m^T p_q$ represents the inner product between the probability vector of $\mathbf{r}_m$ and the probability vectors of all $\mathbf{r}_n$ in $U_{\mathcal{R}}$. Since $\mathbf{r}_m$ and $\mathbf{r}_n$ are not identical, it follows that $p_m^T p_q<1$, and thus $\sum_{\mathbf{r}_q\in U_{\mathcal{R}}}\frac{p_m^T p_q}{|U_{\mathcal{R}}|}<1$. We have

$$(|\mathcal{C}_m|-|\mathcal{D}_m|)(\log|U_{\mathcal{R}}|+\sum_{\mathbf{r}_q\in U_{\mathcal{R}}}\frac{p_m^T p_q}{|U_{\mathcal{R}}|})<(|\mathcal{C}_m|-|\mathcal{D}_m|)(\log|U_{\mathcal{R}}|+1) \quad (22)$$

Substituting Eq.22 into Eq.21 gives

$$\psi(\mathcal{C}_m,\mathcal{D}_m)$$

$$=-\sum_{\mathbf{r}_n\in\mathcal{C}_m}p_m^T p_n+\sum_{\mathbf{r}_k\in\mathcal{D}_m}p_m^T p_k+(|\mathcal{C}_m|-|\mathcal{D}_m|)\log(\sum_{\mathbf{r}_q\in U_{\mathcal{R}}}e^{p_m^T p_q})$$

$$\leq-\sum_{\mathbf{r}_n\in\mathcal{C}_m}p_m^T p_n+\sum_{\mathbf{r}_k\in\mathcal{D}_m}p_m^T p_k+(|\mathcal{C}_m|-|\mathcal{D}_m|)(\log|U_{\mathcal{R}}|+\sum_{\mathbf{r}_q\in U_{\mathcal{R}}}\frac{p_m^T p_q}{|U_{\mathcal{R}}|})$$

$$<-\sum_{\mathbf{r}_n\in\mathcal{C}_m}p_m^T p_n+\sum_{\mathbf{r}_k\in\mathcal{D}_m}p_m^T p_k+(|\mathcal{C}_m|-|\mathcal{D}_m|)(\log|U_{\mathcal{R}}|+1)$$

$$=\overline{\psi}(\mathcal{C}_m,\mathcal{D}_m)$$

# K   VISUALIZATION ON FEATURE SPACE.

To better understand the effectiveness of DVCL, we provide visualizations of the feature space, showing the t-SNE plots (van der Maaten & Hinton, 2008) generated after applying VCL, Complementary Label VCL, and our proposed DVCL. The experiments are conducted on the test set using the FLARE 2022 dataset, trained under a 10% labeled data setting. To more intuitively illustrate the changes among unreliable voxels and reduce the influence of a large number of voxel on the observation, we downsample the original patch size (80, 160, 160) by a factor of 1/4 (20, 40, 40) during test inference and apply dimensionality reduction to the output of the feature head. The visualization results are shown in Fig.9(a), node2 and node13 represent reliable voxels, while node4, node5, node6, and node12 represent unreliable voxels that definitely do not belong to category 2 in the Complementary Label VCL. The numbers indicate the true classes of the features. Additionally, we calculate both inter-class and intra-class distances for the features. The inter-class distance is obtained by computing the Euclidean distance between the feature centers of two categories, while the

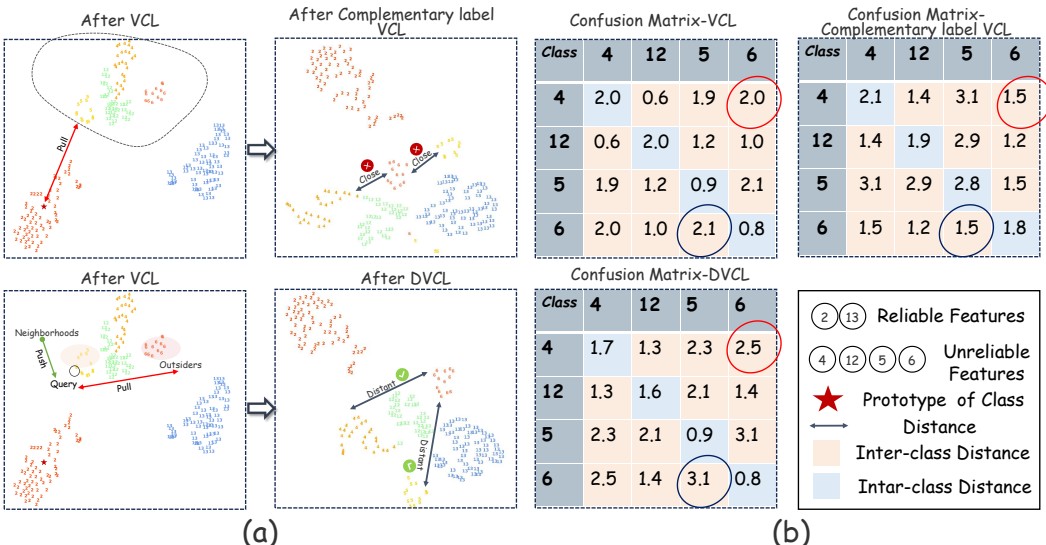

Figure 9: (a) Visualization of the feature spaces learned by VCL, Complementary Label VCL, and our proposed DVCL using t-SNE (van der Maaten & Hinton, 2008) on the FLARE 2022 dataset. (b)The confusion matrices represent the inter-class and intra-class distances.

intra-class distance is defined as the Euclidean distance between the two farthest features within the same category. These distances are organized in a matrix, where the diagonal represents intra-class distances and the off-diagonal represents inter-class distances, as shown in Fig. 9(b).

After applying VCL, it can be observed that its strength lies in forming highly structured features with clear boundaries between different categories, which is a notable advantage of VCL (Wang et al., 2021; Alonso et al., 2021). As shown in Fig 9(a)(top left), features with high classification confidence, such as node2 and node13, are treated as reliable voxels. However, voxels like node12, node4, node5, and node6, which are prone to classification confusion and have low confidence, tend to cluster together in the feature space without forming well-defined feature boundaries. As a result, they are typically classified as unreliable voxels. When Complementary Label VCL is introduced, these unreliable voxels, although prone to classification confusion, can be confidently identified as not belonging to category 2. This causes their features to be pushed away from the feature center of node2. However, this random repulsion operation can disrupt semantic relationships. For example, after applying VCL, node4 and node12 may be confused, but node4 and node6 should be distinguishable. Yet, with Complementary Label VCL, node4 and node6 are unintentionally drawn closer together, as shown in Fig 9(a)(top right). The same applies to node5 and node6. This misalignment may increase the burden on contrastive learning, potentially undermining its effectiveness.

To address the aforementioned issues, DVCL introduces the concepts of "neighbors" and "outsiders," which ensure semantic separation between node4 and node6 while maintaining the semantic consistency of node4. Through the visualization of feature maps and the changes in inter-class distances, it can be observed that after applying DVCL, the semantic relationships between previously unreliable voxels in VCL are preserved, and the feature classification becomes clearer. Additionally, we observe that, compared to Complementary Label VCL, the intra-class distance between unreliable voxels significantly decreases after applying DVCL, resulting in more distinct and distinguishable feature classification boundaries.

## L  STATISTICAL RESULT EVALUATION.

We conducted three repeated experiments for all methods and evaluated the stability of the segmentation results for each organ using box plots. We calculated the Dice values for each organ across all images in the test set for the three experiments. The test set includes 14 images, so for each organ,

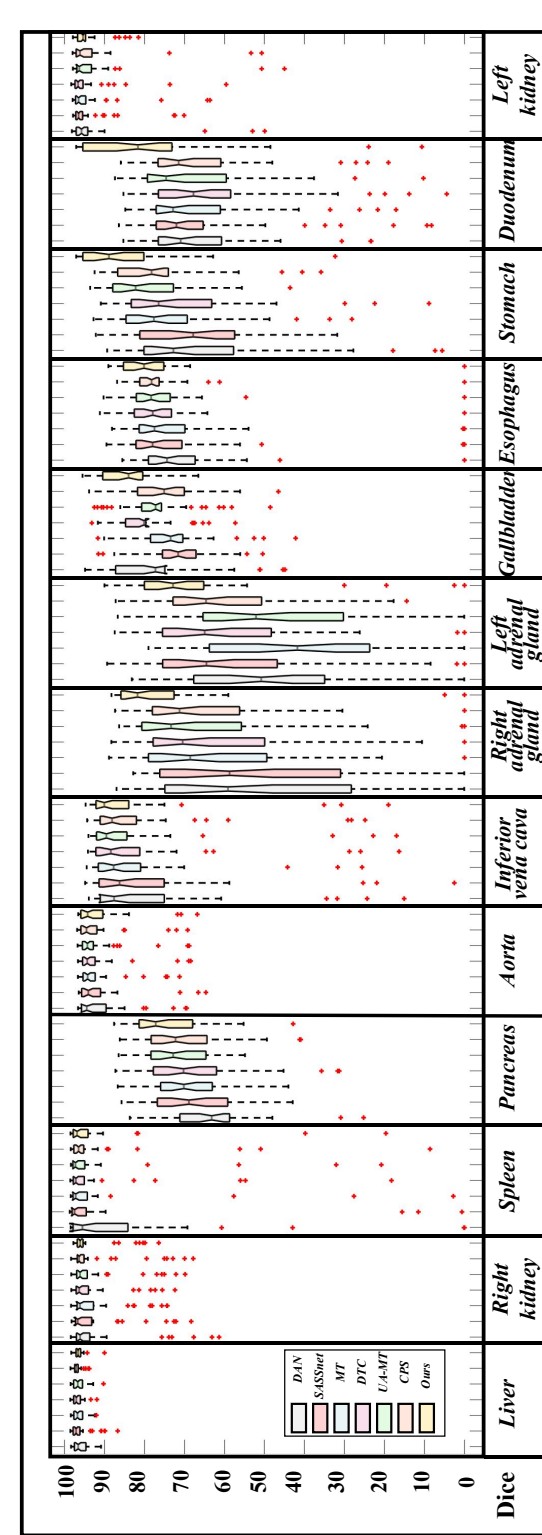

Figure 10: Dice comparison in box plots on FLARE 2022 dataset with 10% labeled images.

the box plot is based on 3×14=42 Dice values. It is noteworthy that for some challenging organs, such as the right adrenal gland and left adrenal gland, our method achieved the best and most stable segmentation performance, with the least fluctuation in the results.

Table 8: Ablation studies of the contrastive learning strategy are conducted, comparing several methods, including InfoNCE (Oord et al., 2018), SimCLR (Chen et al., 2020), KL divergence (Pérez-Cruz, 2008), and Log-likelihood (Wu et al., 2018). Best results are boldfaced.

| Method | Term | Mean Dice | Mean Jaccard |
|---|---|---|---|
| InfoNCE | $-\log \frac{\exp(\mathbf{r}_m \cdot \mathbf{r}_n/\tau)}{\exp(\mathbf{r}_m \cdot \mathbf{r}_n/\tau) + \sum_{\mathbf{r}_n \in \mathcal{D}_m} \exp(\mathbf{r}_m \cdot \mathbf{r}_k/\tau)}$ | 83.05±0.28 | 73.21±0.31 |
| SimCLR | $-\log \frac{\exp(\mathbf{r}_m \cdot \mathbf{r}_n/\tau)}{\sum_{\mathbf{r}_n \in \mathcal{C}_m} \exp(\mathbf{r}_m \cdot \mathbf{r}_n/\tau)} + \log \frac{\exp(\mathbf{r}_m \cdot \mathbf{r}_k/\tau)}{\sum_{\mathbf{r}_k \in \mathcal{D}_m} \exp(\mathbf{r}_m \cdot \mathbf{r}_k/\tau)}$ | 82.77±0.15 | 72.45±0.33 |
| KL | $-\frac{1}{|\mathcal{C}_m|} \sum_{\mathbf{r}_n \in \mathcal{C}_m} \mathrm{KL}(\mathbf{r}_m \| \mathbf{r}_n) + \lambda \frac{1}{|\mathcal{D}_m|} \sum_{\mathbf{r}_k \in \mathcal{D}_m} \mathrm{KL}(\mathbf{r}_m \| \mathbf{r}_k)$ | 82.47±0.31 | 72.01±0.40 |
| Log-likelihood | $-\log \frac{\mathcal{N}(\mathcal{C}_m | \mathbf{r}_m, \theta_\mathcal{S})}{\mathcal{N}(\mathcal{D}_m | \mathbf{r}_m, \theta_\mathcal{S})},$ | **84.11±0.08** | **73.49±0.13** |

## M  ABLATION STUDIE OF THE CONTRASTIVE LEARNING STRATEGY.

We chose to use the likelihood of class relationships, rather than directly pulling together or pushing apart voxel features as in traditional contrastive learning, because we argue that introducing contrastive learning between two feature sets goes beyond the limitations of traditional positive-negative sample comparisons. This approach enables us to more effectively capture distributional differences between groups, thereby providing a clearer understanding of the semantic structural relationships between samples. We adopted negative log-likelihood (NLL) as the loss function, inspired by self-supervised learning (Wu et al., 2018). To further validate the effectiveness of our strategy, we conducted comparative experiments with different contrastive learning strategies (including InfoNCE, SimCLR, and KL divergence). The results demonstrate that our strategy achieves the best performance, as shown in Table.8 .

