# OpenReview forum: "Keep Your Friends Close, and Your Enemies Farther: Distance-aware Voxel-wise Contrastive Learning for Semi-supervised Multi-organ Segmentation"
_ICLR.cc/2025/Conference — Submitted to ICLR 2025_

### Official Review · Reviewer_D2CU · 2024-10-31

**Soundness:** 2
**Presentation:** 2
**Contribution:** 2
**Rating:** 3
**Confidence:** 5

**Summary:**

The authors propose an entropy-weighted cross pseudo supervision loss (a variant of the cross pseudo supervision loss), a high-quality voxel-wise contrastive learning loss (a variant of contrastive learning loss), and a distance-aware voxel-wise contrastive learning loss, focusing particularly on the latter for semi-supervised medical image segmentation. Experiments were conducted on four medical imaging datasets.

**Strengths:**

Performance evaluation and comparison were conducted on four datasets, with both quantitative and qualitative results presented.

**Weaknesses:**

1. The authors address semi-supervised segmentation, motivated by the high cost and expertise needed for manual annotation. While this is a valid concern for large-scale datasets, it is less compelling for smaller datasets, as used in this paper. In the experiments, the maximum number of labeled samples is only 90 (in AMOS), raising questions about the method’s applicability to real-world applications. Would the proposed approach demonstrate improved performance with hundreds or thousands of labeled images for training? Additionally, it remains unclear how much annotation effort is saved (i.e., the percentage of labeled data needed with this method to match the performance of fully supervised models).
2. The method design appears to contradict its intended objectives (details below).
3. Certain parts of the paper lack clarity.
4. Language issues are frequent.

**Questions:**

1. How would the proposed method perform with a larger number of labeled images for training? While this study is not alone in using a small dataset for model training and evaluation, such a setup may fall short of addressing the core challenges that motivate semi-supervised segmentation research. In fact, the substantial gap between fully supervised and semi-supervised learning is expected to narrow significantly when either the training or testing set is increased.
2. Why discriminative features for segmentation can be learned by maintaining useful semantic relationships among unreliable voxels? I think they contradict with each other, because discriminativeness means features from different classes are separated, but the proposed distance-aware voxel-based contrastive learning loss may pull close together voxels that are semantically similar but belong to different classes.
3. What are the thresholds used in Fig 1a? It is surprising to see that the ratios of reliable labels are all lower than 1%.
4. In Fig 1b, A and B are apparently closer to prototype 1, and C and D are closer to prototype 2. However, in the text, it is mentioned that A $\notin$ 1, B $\notin$ 1, C $\notin$ 2, and D $\notin$ 2 (Line 99). Shouldn't it be A $\notin$ 2, B $\notin$ 2, C $\notin$ 1, and D $\notin$ 1? Additionally, using an illustrative figure as evidence (Line 96) is unconvincing.
5. In Table 2, is EBL equivalent to entropy-weighted cross pseudo supervision loss? If so, why the proposed high-quality voxel-wise contrastive learning loss is missing?
6. The performance of different methods shows considerable variability. However, does the small number of test images provide sufficient statistical power?
7. In Equation 3, do you use a fixed $\alpha$ throughout training? Additionally, for some easy samples, e.g., image patches that is entirely from the background region, the model can make confident and correct predictions. Will the proposed strategy incorrectly treat a part of the voxels as unreliable voxels?
8. It is surprising to see the optimal value of $\tau$ is 1. With $\tau = 1$, will the high-quality contrastive loss reach 0 as the training goes?
9. In Equation 9, are $r_{c}$ L2-normalized? If so, why the feature center of all anchor voxels can be used for computing contrastive loss? If not, please also explain.
10. The K nearest voxels are selected as neighbors, and K furthest as outsiders, for calculating the distance-aware voxel-wise contrastive learning loss. Do these straightforward choices risk making model training too easy to achieve and thus this loss terms make no differences?
11. Language issues are common, e.g., in Line 338, "fearest" should be "furthest".

---

> ### Author Response · Authors · 2024-11-26
> **Response to Reviewer D2CU (1/5)**
>
> ### Thank you for your detailed and constructive comments! Below is our response to your comments.
> ___
> ### First, we give our response to the weaknesses you mentioned:
>
> > #### **W1:** The authors address semi-supervised segmentation, motivated by the high cost and expertise needed for manual annotation. While this is a valid concern for large-scale datasets, it is less compelling for smaller datasets, as used in this paper. In the experiments, the maximum number of labeled samples is only 90 (in AMOS), raising questions about the method’s applicability to real-world applications. Would the proposed approach demonstrate improved performance with hundreds or thousands of labeled images for training? Additionally, it remains unclear how much annotation effort is saved (i.e., the percentage of labeled data needed with this method to match the performance of fully supervised models).
>
> #### **Response:** The annotation of 3D volumetric multi-organ medical images is challenging due to the overlap and interlacing of multiple organs, and typically requires collaboration from multiple experts. As a result, annotating 3D multi-organ medical images involves numerous challenges and difficulties, requiring significant time and resources to produce high-quality annotations. The AMOS dataset, which contains 300 fully annotated 3D CT images, is the largest 3D multi-organ dataset within our knowledge. We split the dataset into training, validation, and testing sets in a 6:2:2 ratio (180 for training, 60 for validation, and 60 for testing). For the AMOS dataset, we analyzed the performance using fully labeled data (180 images) and observed that with only 10% labeled data, our method performed nearly as well as when using the full labeled dataset , as shown in Tab.a.
>
> #### **Table a : Quantitative results on three settings of AMOS dataset. Best results are boldfaced.**
> | Model          | Ours(18 labeled images) | Ours(90 labeled images) | Fully(180 labeled images)  |
> |----------------|----------|----------|---------------
> | Dice (%)       | 71.09 ± 0.51     | 72.84 ± 0.23     | **76.55 ± 0.47**      |
> | JACCARD (%)        | 59.34 ± 0.46      | 61.38 ± 0.23    | **65.33 ± 0.42**   |
>
> > #### **W2:** The method design appears to contradict its intended objectives (details below).
>
> #### **Response:** Thank you for your comments. We will provide a detailed response later.
>
> > #### **W3:** Certain parts of the paper lack clarity.
>
> #### **Response:** Thank you for your feedback. We apologize for any lack of clarity in certain sections of the paper. We will review these parts carefully and revise them to improve the clarity.
>
> > #### **W4:** Language issues are frequent.
>
> #### **Response:** Thank you for pointing out the language issues. We sincerely apologize for any language-related shortcomings in the paper. We will thoroughly review the manuscript and make the necessary corrections to improve the language.
>
> ### Next, we give our response to your questions:
>
> > #### **Q1:** How would the proposed method perform with a larger number of labeled images for training? While this study is not alone in using a small dataset for model training and evaluation, such a setup may fall short of addressing the core challenges that motivate semi-supervised segmentation research. In fact, the substantial gap between fully supervised and semi-supervised learning is expected to narrow significantly when either the training or testing set is increased.
>
> #### **Response to "How would...with a larger number...for training?":**
> #### The annotation of 3D volumetric multi-organ medical images is challenging due to the overlap and interlacing of multiple organs, and typically requires collaboration from multiple experts. As a result, annotating 3D multi-organ medical images involves numerous challenges and difficulties, requiring significant time and resources to produce high-quality annotations. **The AMOS dataset, which contains 300 fully annotated 3D CT images, is the largest 3D multi-organ dataset within our knowledge. **
>
> #### **Response to "While this study...core challenges...research.":**
> #### In semi-supervised segmentation research, the key focus shifts to effectively leveraging unlabeled data to enhance segmentation model performance in the absence of abundant annotations. **We think the core challenge lies in how to effectively exploit the potential of unlabeled data while mitigating the adverse impact of noisy pseudo-labels.** To address this challenge, we proposed the DVCL method, which aims to achieve this goal by exploring and maintaining the semantic relationships between unreliable voxels. Furthermore, our experimental results demonstrate that for challenging classes in multi-organ segmentation—those with a high proportion of unreliable voxels—our method achieves remarkable performance, as illustrated in Fig.4.
> ___
> ### Part (1/5)

---

> > ### Comment · Reviewer_D2CU · 2024-11-26
> >
> > It is incorrect to say that the AMOS dataset is the largest 3D multi-organ dataset. In reality, the TotalSegmentator dataset surpasses AMOS not only in the number of annotated images (1000+ images) but also in the number of classes (100+ classes). Based on the authors’ response, it appears they have not fully acknowledged this limitation highlighted by the reviewer.

---

> > > ### Author Response · Authors · 2024-11-27
> > > **Response to Reviewer D2CU - Round 2**
> > >
> > > Thank you for your response. We sincerely apologize for the oversight in our dataset investigation. We will immediately begin conducting experiments on this dataset.  As the preprocessing process requires some time, we kindly request one to two days to complete it. Once we obtain the results, we will update them here at the earliest opportunity.

---

> > > ### Author Response · Authors · 2024-12-01
> > > **Response to Reviewer D2CU - Round 2**
> > >
> > > #### Thank you for your feedback. Following your suggestion, we utilized the **TotalSegmentator dataset version 2.0.1**. After decompressing the official archive, the dataset is organized as follows: it includes a meta.csv file and multiple subdirectories named sxxxx. Each sxxxx subdirectory contains a segmentations folder, which stores the segmentation files for various categories, along with a ct.nii.gz file.
> > > ```
> > > TotalSegmentator
> > > ├── meta.csv
> > > ├── s0000
> > > │   ├── segmentations
> > > │   │   ├── adrenal_gland_left.nii.gz
> > > │   │   ├── adrenal_gland_right.nii.gz
> > > │   │   ├── aorta.nii.gz
> > > │   │   └── ...
> > > │   ├── ct.nii.gz
> > > ├── s0001
> > > ├── s0002
> > > ├── s0003
> > > └── ...
> > > ```
> > > #### **The dataset consists of 1,228 CT images. Based on the meta.csv file, we divided the data into 1,082 training images, 57 validation images, and 89 test images.** To reduce the time spent on hyperparameter tuning, we integrated 13 organ labels from the FLARE and BTCV datasets and applied the same preprocessing steps for our experiments. The experiments were conducted under two settings: 10% labeled data (108 labeled + 974 unlabeled) and 50% labeled data (541 labeled + 541 unlabeled). Due to time constraints, we only compared our method with the U$^2$PL method, which is the best-performing approach in the FLARE dataset apart from ours. **The results show that our method consistently outperforms U$^2$PL across different settings. Notably, with only 10% labeled data, the mean Dice score achieved by our method is nearly comparable to the performance of the 3D U-Net trained with 100% labeled data.**
> > >
> > > #### **Table a : Quantitative results on two settings of TotalSegmentator dataset.**
> > > **Baseline** |LAG|RAG|Aor|Duo|Eso|Gal|IVC|L.Kid| R.Kid|Liv|Pan|Spl|Sto|Mean Dice|Mean Jaccard
> > > -|-|-|-|-|-|-|-|-|-|-|-|-|-|-|-
> > >  U$^2$PL(10%labeled)|72.95|77.23|94.97|77.11|78.44|51.54|86.65|96.74|92.27|97.59|81.91|97.36|90.90|84.28$\pm$0.4|73.29$\pm$0.18
> > >  Ours(10%labeled)|73.20|78.99|94.83|78.47|86.04|57.53|85.96|96.66|90.26|97.85|82.02|97.39|90.76|85.38$\pm$0.26|74.44$\pm$0.24
> > >  U$^2$PL(50%labeled)|72.71|78.07|95.69|78.33|83.56|62.24|87.26|96.28|88.44|97.89|81.64|97.14|88.83|85.24$\pm$0.19|74.17$\pm$0.34
> > >  Ours(50%labeled)|74.31|80.26|96.02|77.78|85.53|57.78|87.65|96.33|90.59|97.81|82.03|97.39|90.56|85.70$\pm$0.25|75.07$\pm$0.38
> > >  Fully(100%labeled)|76.43|77.57|95.92|80.68|86.42|60.47|87.79|96.98|92.65|98.11|83.54|97.76|92.75|86.70$\pm$0.67|76.44$\pm$0.77
> > >  ___
> > > ### Thank you for reading this response. We would be happy to discuss any further questions!

---

> ### Author Response · Authors · 2024-11-26
> **Response to Reviewer D2CU (2/5)**
>
> > #### **Q2:** Why discriminative features for segmentation can be learned by maintaining useful semantic relationships among unreliable voxels? I think they contradict with each other, because discriminativeness means features from different classes are separated, but the proposed distance-aware voxel-based contrastive learning loss may pull close together voxels that are semantically similar but belong to different classes.
>
> #### **Response:** We are sorry for the misunderstanding caused by our description. Regarding the reviewers' concern about the potential issue of pulling semantically similar but categorically different samples closer together, we address this by carefully selecting the value for the number of neighbors, $K$, to minimize such occurrences. By adjusting $K$, we can effectively reduce misclassifications and ensure that the selected neighbors are as likely as possible to belong to the same class. Inspired by [10] [11], our method selects the $K$ most similar voxels $(r_{n}| top-K(cos\left(r_m,r_{n}\right),\forall r_{n} \in U_R))$ as neighbors based on cosine similarity. We control the number of neighbors by adjusting the value of $K$, thereby selecting voxels that are more likely to belong to the same class. **By identifying as many same-class neighbors and different-class outsiders as possible, DVCL can learn discriminative features.**  We demonstrate this phenomenon through feature visualization of real data and statistical analysis of the inter-class distances of unreliable voxels (for details, please refer to Appendix K, lines 1214-1234). **As shown in Fig.9(a)(bottom right), discriminative features can be learned between unreliable voxels of different classes.**
>
>
> > #### **Q3:** What are the thresholds used in Fig 1a? It is surprising to see that the ratios of reliable labels are all lower than 1%.?
>
> #### **Response:** Fig. 1(a) are derived from the classic threshold-based semi-supervised method FixMatch[9], with a threshold set at $\tau = 0.95$. We apologize for the unclear labeling in the chart. In the Fig. 1(a), the Dice coefficient ranges from 0 to 1, while the "percentage of reliable voxels" should actually be represented as ($*10 ^2\%$), with a range from 0% to 100%. For instance, the reliable voxel percentage in the liver is close to 100%, while in the right adrenal gland, it is only 40%.
>
> > #### **Q4:** In Fig 1b, A and B are apparently closer to prototype 1, and C and D are closer to prototype 2. However, in the text, it is mentioned that A ∉ 1, B ∉ 1, C ∉ 2, and D ∉ 2 (Line 99). Shouldn't it be A ∉ 2, B ∉ 2, C ∉ 1, and D ∉ 1? Additionally, using an illustrative figure as evidence (Line 96) is unconvincing.
>
> #### **Response:** We apologize for the lack of precision in our illustration. The toy example in Fig.1 is intended to illustrate our motivation and the workings of the algorithm. Due to some oversight on our part, there was an error in the plotting. We demonstrate this phenomenon through feature visualization of real data (for details, please refer to Appendix K, Figure 9(a)).
> ___
> ### Part (2/5)

---

> ### Author Response · Authors · 2024-11-26
> **Response to Reviewer D2CU (3/5)**
>
> > #### **Q5:** In Table 2, is EBL equivalent to entropy-weighted cross pseudo supervision loss? If so, why the proposed high-quality voxel-wise contrastive learning loss is missing?
>
>  #### **Response:**  We sincerely apologize for the lack of clarity in our previous explanation. EBL (Entropy-based Learning) includes the ESM (Entropy-based Selection Module) for dynamically selecting reliable and unreliable voxels, as well as the ECPS (Entropy-weighted Cross Pseudo Supervision loss), which reweights the unsupervised loss based on the reliable and unreliable voxels selected by ESM. In the initial manuscript, DVCL in Table 2 referred to the combination of HqC (High-quality Voxel-wise Contrastive Learning) and DVCL (Distance-aware Voxel-wise Contrastive Learning). Following your suggesion, we have added experiments that separately evaluate HqC and DVCL and have redrawn the corresponding Tab.b.:
>
> #### **Table b : Ablation on component aspect. Best results are boldfaced.**
> **Baseline** | EBL| HqC| DVCL| Liv| Spl| Sto| L.kid| R.kid| Aor| Pan| IVC| Duo| Gal| Eso| RAG|LAG|Mean Dice|Mean Jaccard
> -|-|-|-|-|-|-|-|-|-|-|-|-|-|-|-|-|-|-
>  ✔||||96.92 | 91.86 | 77.02 | 92.70 | 92.71 | 92.25 | 69.39 | 81.91 | 65.94 | 75.15 | 72.78 | 63.56 | 58.96 | 79.32 ± 0.46   | 68.14 ± 0.61
>  ✔| ✔||| 96.58 | 90.56 | 82.80 | 93.77 | 93.35 | 92.23 | 76.59 | 83.93 | 69.09 | 82.71 | 73.43 | 67.04 | 62.64 | 82.36 ± 0.28   | 71.40 ± 0.48
>  ✔|| ✔|| 96.15 | 91.71 | 76.49 | 94.44 | **94.34** | 92.02 | 74.74 | 84.06 | 67.84 | 81.68 | 76.81 | 69.82 | 67.77 | 82.18 ± 0.34   | 71.61 ± 0.34
>  ✔ ||| ✔| 96.61 | 90.74 | 77.61 | 94.05 | 93.46 | 92.39 | 72.69 | 83.65 | 69.14 | 80.43 | 75.20 | 65.23 | 62.83 | 81.62 ± 0.10   | 70.76 ± 0.13
>  ✔|| ✔| ✔| 96.84 | **95.04** | 82.65 | 94.69 | 93.43 | 92.14 | 76.83 | 83.60 | 69.14 | 85.70 | 74.17 | **70.65** | 68.36 | 83.41 ± 0.31   | 72.50 ± 0.54
>  ✔ | ✔| ✔ | ✔| **97.37**|93.53|**83.47**|**94.86**|93.69|**92.52**|**76.98**|**84.43**|**70.97**|**87.71**|**75.69**|70.60| **71.67**|**84.11±0.08**|**73.49 ± 0.13**
>
> > ### **Q6:** The performance of different methods shows considerable variability. However, does the small number of test images provide sufficient statistical power?
>
> #### **Response:**
> * #### To assess the stability of the results, we used box plots as a substitute for these statistical measures. For further details, please refer to Fig.10 in Appendix L of the new version.
>
> * #### The annotation of 3D volumetric multi-organ medical images is challenging due to the overlap and interlacing of multiple organs, and typically requires collaboration from multiple experts. As a result, annotating 3D multi-organ medical images involves numerous challenges and difficulties, requiring significant time and resources to produce high-quality annotations. The AMOS dataset, which contains 300 fully annotated 3D CT images, is the largest 3D multi-organ dataset within our knowledge. We split the dataset into training, validation, and testing sets in a 6:2:2 ratio (180 for training, 60 for validation, and 60 for testing). For instance, the LA dataset [3] and NIHpancreas dataset [4], used by excellent semi-supervised algorithms like MCF [5] and AD-MT [6], only contain 20 test samples.
>
>
>
> > #### **Q7:** In Equation 3, do you use a fixed $\alpha$ throughout training? Additionally, for some easy samples, e.g., image patches that is entirely from the background region, the model can make confident and correct predictions. Will the proposed strategy incorrectly treat a part of the voxels as unreliable voxels?
>
> #### **Response:**
> * #### In Equation 3, we treat $\alpha$ as a fixed coefficient.
>
> * #### Regarding the extreme case mentioned by the reviewer, first, when all voxels belong to simple background voxels, the model is highly confident in classifying these voxels, resulting in very low entropy. However, for voxels with relatively high entropy, it is likely that the model encounters difficulties in distinguishing between background and foreground. Even if the model correctly classifies these voxels as background, the confidence may still be low. Since our partitioning is based on voxel entropy, it is correct to classify these low-confidence voxels as unreliable voxels. Furthermore, during training, we use a patch size of $128×128×80$, which occupies more than two-thirds of the total volume of the 3D data. As a result, it is highly unlikely to encounter a scenario where all patches consist solely of background voxels.
>  ___
> ### Part (3/5)

---

> > ### Comment · Reviewer_D2CU · 2024-12-02
> >
> > Based on the response to Q7, it appears that the proposed method lacks inherent robustness to variations in images and hyperparameters. Relying on the assumption that extreme cases will rarely occur under specific settings undermines the method’s soundness and should be avoided.

---

> > > ### Author Response · Authors · 2024-12-04
> > > **Response to Reviewer D2CU - Round 3**
> > >
> > > ####   Thank you for your suggestion. We will remove the assumption regarding "extreme cases rarely occurring" from our discussion. In response to Q7, we would like to address the reviewer's concern. For example, in cases where image patches entirely consist of background regions and the model makes correct predictions, our method would still identify some voxels as unreliable. The division of voxels in our method is primarily based on entropy values. Under such circumstances, voxels classified as unreliable typically exhibit higher entropy. Compared to other voxels, these higher-entropy voxels are more likely to encounter difficulties in distinguishing between background and foreground. Consequently, our method identifies a subset of these voxels within the reliable voxels as unreliable. **Even if these voxels appear reliable, their relatively higher entropy indicates they are less reliable than low-entropy voxels.** This approach enables our method to better address potential uncertainties, thereby improving the overall robustness of the model.

---

> ### Author Response · Authors · 2024-11-26
> **Response to Reviewer D2CU (4/5)**
>
> > #### **Q8:** It is surprising to see the optimal value of $\tau$ is 1. With $\tau = 1$ , will the high-quality contrastive loss reach 0 as the training goes?
>
> #### **Response:** Initially, we set the parameter $\tau = 0.5$ by default, following the latest studies[7],[8]. Since the aforementioned experiments were conducted on 2D natural image datasets, it is essential to consider that 3D medical images typically contain more noise and local deformations. A larger $\tau$ results in smoother gradients, reducing the intensity of interactions. This smoothing effect can effectively mitigate the impact of noisy samples or outliers on the learning process, thereby enhancing the robustness of the model. Subsequently, we conducted ablation experiments to adjust $\tau$ and found that $\tau = 1$ achieved the best performance under our experimental settings. Detailed results of the ablation experiments can be found in Fig. 8(b) (Appendix I) in our paper.
>
> > #### **Q9**: In Equation 9, are L2-normalized? If so, why the feature center of all anchor voxels can be used for computing contrastive loss? If not, please also explain.
>
> #### **Response:**
> * #### $\mathbf{r}_c$ represents the features of the anchor set in category $c$, and $\mathbf{r}_c\in \mathbb{R}^{d}$ is extracted by a feature head in the network. This feature head is composed of two "Conv-BN-ReLU-Dropout modules" and is connected by a $1×1×1$ convolution layer to adjust the output dimension $d$.
> * #### In the supervised contrastive learning framework, the true ground-truth masks provide category information for the anchor voxels, and anchor voxels of the same category are grouped into the same set. The feature center of all anchor voxels in this set (i.e., the category prototype) can represent the position of this category in the feature space and acts as a positive sample for all anchor voxels of that category in contrastive learning [7]. In the semi-supervised learning framework, due to the lack of true labels for unlabeled data, to maintain the quality of contrastive learning, we only select labeled voxels with accurate predictions and voxels with reliable guidance from unlabeled data as anchor voxels for a given category. This ensures that the feature center can accurately reflect the position of that category in the feature space, allowing it to serve as a positive sample in contrastive learning.
>
> > ### **Q10:** The K nearest voxels are selected as neighbors, and K furthest as outsiders, for calculating the distance-aware voxel-wise contrastive learning loss. Do these straightforward choices risk making model training too easy to achieve and thus this loss terms make no differences?
>
> #### **Response:** Neighbors typically exhibit high similarity, while outsiders represent the differences between categories. This selection helps the model better learn semantic relationships and enhances its ability to distinguish between different semantic classes. Although these selection methods may seem straightforward, they enable the model to capture finer and more robust semantic relationships between unreliable voxels in the feature space. Through the DVCL method, we are able to maintain meaningful semantic connections between unreliable voxels while fully leveraging the benefits of contrastive learning. Experimental results demonstrate the significant role these loss terms play in actual segmentation tasks, without making the model training overly simplistic, but instead providing effective guidance for the model’s learning process.  To further validate the effectiveness of our method, we conducted comparative experiments with alternative function designs. The results demonstrate that the Log-Likelihood achieves the best performance, as shown in Tab.b.
>
> #### **Table b: Ablation studies of the contrastive learning strategy. Best results are boldfaced.**
> |**Method**|**Term**|**Mean Dice**|**Mean Jaccard**|
> |-|-|-|-|
> |InfoNCE [6]| $-\log\frac{\exp\left(r_{m}\cdot r_{n}/\tau\right)}{\exp\left(r_{m}\cdot r_{n}/\tau\right)+\sum_{r_{n}\in D_{m}}\exp\left(r_{m}\cdot r_{k}/\tau\right)}$|83.05 ± 0.28|73.21 ± 0.31|
> |SimCLR [7]| $-\log\frac{\exp{(r_{m}\cdot r_{n}/\tau)}}{\sum_{r_{n}\in C_{m}}\exp{(r_{m}\cdot r_{n}/\tau)}} + \log\frac{\exp{(r_{m}\cdot r_{k}/\tau)}}{\sum_{r_{k}\in D_{m}}\exp{(r_{m}\cdot r_{k}/\tau)}}$|82.77 ± 0.15|72.45 ± 0.33|
> |KL [8]| $-\frac{1}{\mid C_m \mid}\sum_{r_n\in C_{m}} KL(r_{m}\mid\mid r_{n})+\lambda\frac{1}{D_{m}}\sum_{r_{k}\in D_{m}} KL(r_{m}\mid\mid r_{k})$|82.47 ± 0.31|72.01 ± 0.40|
> |Log-likelihood [5]|$-\log \frac{\mathcal{N}(C_m \mid r_m, \theta_S)}{\mathcal{N}(D_m \mid r_m, \theta_S)}$ |**84.11 ± 0.08**|**73.49 ± 0.13**|
>  ___
> ### Part (4/5)

---

> ### Author Response · Authors · 2024-11-26
> **Response to Reviewer D2CU (5/5)**
>
> > ### **Q11:** Language issues are common, e.g., in Line 338, "fearest" should be "furthest".
>
> #### **Response:** Thanks for pointing this out. We have fixed the typos we find.
>
>
> ### References
> #### [1] Chongjian G E, Wang J, Tong Z, et al. Soft Neighbors are Positive Supporters in Contrastive Visual Representation Learning[C]//The Eleventh International Conference on Learning Representations.
>
> #### [2] Dwibedi D, Aytar Y, Tompson J, et al. With a little help from my friends: Nearest-neighbor contrastive learning of visual representations[C]//Proceedings of the IEEE/CVF International Conference on Computer Vision. 2021: 9588-9597.
>
> #### [3] Xiong Z, Xia Q, Hu Z, et al. A global benchmark of algorithms for segmenting the left atrium from late gadolinium-enhanced cardiac magnetic resonance imaging[J]. Medical image analysis, 2021, 67: 101832.
>
> #### [4] Roth H R, Lu L, Farag A, et al. Deeporgan: Multi-level deep convolutional networks for automated pancreas segmentation[C]//Medical Image Computing and Computer-Assisted Intervention--MICCAI 2015: 18th International Conference, Munich, Germany, October 5-9, 2015, Proceedings, Part I 18. Springer International Publishing, 2015: 556-564.
>
> #### [5] Wang Y, Xiao B, Bi X, et al. Mcf: Mutual correction framework for semi-supervised medical image segmentation[C]//Proceedings of the IEEE/CVF conference on computer vision and pattern recognition. 2023: 15651-15660.
>
> #### [6] Zhao Z, Wang Z, Wang L, et al. Alternate diverse teaching for semi-supervised medical image segmentation[C]//European Conference on Computer Vision. Springer, Cham, 2025: 227-243.
>
> #### [7] Liu S, Zhi S, Johns E, et al. Bootstrapping Semantic Segmentation with Regional Contrast[C]//International Conference on Learning Representations.
>
> #### [8] Wang Y, Wang H, Shen Y, et al. Semi-supervised semantic segmentation using unreliable pseudo-labels[C]//Proceedings of the IEEE/CVF conference on computer vision and pattern recognition. 2022: 4248-4257.
>
> #### [9] Sohn K, Berthelot D, Carlini N, et al. Fixmatch: Simplifying semi-supervised learning with consistency and confidence[J]. Advances in neural information processing systems, 2020, 33: 596-608.
>
> #### [10] Yang S, Wang Y, Van De Weijer J, et al. Generalized source-free domain adaptation[C]//Proceedings of the IEEE/CVF international conference on computer vision. 2021: 8978-8987.
>
> #### [11] Cover T, Hart P. Nearest neighbor pattern classification[J]. IEEE transactions on information theory, 1967, 13(1): 21-27.
>
>  ___
> ### Part (5/5)
>
> ### Thank you for reading this response. We would be happy to discuss any further questions!

---

> ### Comment · Reviewer_D2CU · 2024-12-02
>
> 1. I still don't think $\tau$ = 1 makes sense to me. And I suspect the above choice for feature head affects the optimal $\tau$ found by the authors. Please investigate.
>
> 2. In Q10, I was asking if choosing the K nearest voxels as neighbors, and K furthest as outsiders will make the loss easily go to 0. My guess is that this choice makes the loss easily go to 0 and diminishes its effectiveness. The improved performance attributed by this loss term might be due to the weaker baseline.

---

> > ### Author Response · Authors · 2024-12-04
> > **Response to Reviewer D2CU - Round 3**
> >
> > ### Thank you for your insightful feedback! Below is our response to your comments.
> > ___
> >
> > > #### **Question1:** I still don't $\tau=1$ makes sense to me. And I suspect the above choice for feature head affects the optimal $\tau$ found by the authors. Please investigate.
> >
> > #### **Response:** Regarding the choice of the hyperparameter $\tau$, we followed the ablation study in U$^2$PL[1] and conducted a hyperparameter ablation study on our dataset. The experimental results indicate that our method achieves the best performance when $\tau = 1$.
> >
> > #### As for the feature head selection strategy, we also drew inspiration from U$^2$PL[1], with the only difference being that we replaced the 2D feature head with a 3D version. The specific 3D feature head code is as follows:
> > ```python
> > self.representation = nn.Sequential(
> >     nn.Conv3d(16, self_channels, kernel_size=3, stride=1, padding=1, bias=True),
> >     nn.BatchNorm3d(self_channels),
> >     nn.ReLU(inplace=True),
> >     nn.Dropout3d(0.1),
> >     nn.Conv3d(self_channels, self_channels, kernel_size=3, stride=1, padding=1, bias=True),
> >     nn.BatchNorm3d(self_channels),
> >     nn.ReLU(inplace=True),
> >     nn.Dropout3d(0.1),
> >     nn.Conv3d(self_channels, self_channels, kernel_size=1, stride=1, padding=0, bias=True),
> > )
> > ```
> > > #### **Question2:** In Q10, I was asking if choosing the K nearest voxels as neighbors, and K furthest as outsiders will make the loss easily go to 0. My guess is that this choice makes the loss easily go to 0 and diminishes its effectiveness. The improved performance attributed by this loss term might be due to the weaker baseline.
> >
> > #### **Response:** Regarding the reviewer's concern that selecting K nearest neighbors and K distant outsiders might make the training process too easy, we acknowledge that outsiders, due to their clear differences, may indeed introduce this potential risk. However, in our experiments, inspired by [2], we use cosine similarity to select the top-K most similar features from the feature head. It is important to note that our loss function is calculated based on the output probabilities of the segmentation head. **Crucially, feature similarity and classification probability similarity are not directly causally related, which helps prevent the model from prematurely converging to a simple solution due to the simplicity of the samples.**
> >
> > ####  Our loss calculation formula primarily relies on $\sum_{r_n \in C_m}p_m^Tp_n+\sum_{r_k \in D_m}p_m^Tp_k$, with gradients derived from $p_m$, $p_n$, and $p_k$. The first term enforces prediction consistency among neighbors, while the second term, in a straightforward interpretation, aims to disperse the predictions of outsiders. It is important to note that the dot product between two softmaxed predictions reaches its maximum value when both predictions align to the same class and are close to a one-hot vector.
> >
> > ####   Since the feature differences between $r_k$ (outsider) and $r_m$ are large, their inner product $\sum_{r_k \in D_m} p_m^T p_k$ is small, resulting in a reduced contribution from the second term of the loss function. Consequently, the loss value decreases quickly during training, and the model tends to learn simpler patterns. However, for $r_n$ (neighbor), since we use cosine similarity to select the top-K most similar features from the feature head, there is no direct causal relationship between feature similarity and classification probability similarity. Therefore, their inner product $\sum_{r_n \in C_m}p_m^Tp_n$ may not necessarily be the largest. With this approach, we believe the training process will not be overly easy, thus preventing the model from prematurely converging to a simplistic solution.
> >
> > ### References
> > [1] Wang Y, Wang H, Shen Y, et al. Semi-supervised semantic segmentation using unreliable pseudo-labels[C]//Proceedings of the IEEE/CVF conference on computer vision and pattern recognition. 2022: 4248-4257.
> >
> > [2] Cover T, Hart P. Nearest neighbor pattern classification[J]. IEEE transactions on information theory, 1967, 13(1): 21-27.
> >
> > ___
> > ### Finally, we would like to extend our heartfelt gratitude to the reviewers. Your meticulous and thoughtful feedback has been instrumental in enhancing the quality of our paper while offering us invaluable insights. We deeply appreciate your dedication and guidance throughout the review process.

---

### Official Review · Reviewer_qnby · 2024-11-04

**Soundness:** 1
**Presentation:** 3
**Contribution:** 2
**Rating:** 3
**Confidence:** 5

**Summary:**

This work presents a voxel-level contrastive learning (VCL) method for multi-organ semantic segmentation from medical images using semi-supervised learning (learning from small number of labeled and large number of unlabeled examples). Voxel-level contrasting learning for semi-supervised segmentation is not new by itself, where voxels with similar labels are considered positive examples and their representations are pulled closer, whereas voxels with dissimilar labels are considered negative examples and their representations pulled away from each other; various methods have shown applications for medical segmentation as listed in the references. A problem with these methods is the issue of handling poor pseudo labels (initial labels generated on unlabeled examples using model trained on labeled examples), which can result in noisy labels and hence poor contrastive training and algorithm performance. A known method to address this problem in semi-supervised learning literature is to use complementary labels, where the idea is to rely on the lowest predicted probability class and enforce that the voxel does not belong to that class rather than use the highest probability class with low confidence for contrastive learning. The paper claims that when using this method (complementary label) of contrastive learning on unreliable (poor confidence) pseudo labeled voxels, it can incorrectly move the voxel features that are semantically similar (close by in feature space) away from each other, while moving dissimilar voxel features incorrectly closer to each other. Hence, the main claimed contribution of the paper is a method to retain similar voxel features (closest neighbors in feature space) closer and dissimilar voxel features (farthest neighbors in feature space) far from each other. The second claimed contribution of the paper is a dynamic way to ascertain which voxels are unreliable in the image during each iteration of training using entropy of the predicted label distribution. If the entropy is higher than a certain threshold based on the average entropy of the voxels of the image in that iteration, it is deemed unreliable, and vice-versa. For training, the paper uses the Cross Pseudo Supervision (CPS) framework where consistency of predictions on same input using two similar but differently initialized networks is enforced in addition to supervised loss using labeled images, contrastive loss using labeled and unlabeled but reliable voxels, as well as the proposed complementary label contrastive loss method for unreliable voxels that claims to maintain semantic neighborhood relationships between voxel features. The evaluation and ablation experiments are performed on 4 open source CT image data sets and a small improvement in performance over other VCL methods is demonstrated.

**Strengths:**

1.	The area and the general problem of handling noisy pseudo labels in semi-supervised learning has high practical importance and would be of interest to the community.
2.	The dynamic entropy based method for ascertaining unreliable voxels proposed by the paper, while is not the primary contribution, seems original and a simple way to account for the varying confidence in the algorithm predictions over training iterations.
3.	The paper is written and presented well.
4.	The comparisons and ablations experiments are extensive.

**Weaknesses:**

1.	The paper does not clearly demonstrate the existence of the problem they are trying to solve, except for the cartoon illustration. It is not clear how often the claimed problem of semantically (dis)similar voxel features being (pulled)pushed (closer)away while using complementary label based VCL happens and by what degree, and how much does that affect the output. The paper needs to better illustrate the significance of the problem with real data. E.g., this could be demonstrated empirically in the following ways: a) Analyzing feature space visualizations before and after applying complementary label VCL; b) Quantifying how often semantically similar voxels end up far apart after complementary label VCL; c) Measuring the impact on segmentation performance when this issue occurs.
2.	A primary concern is the reasoning/motivation of the paper’s main contribution, which is to retain the neighborhood relationships of voxel features in the feature space as determined prior to applying VCL. The paper claims that these relationships should be maintained through the VCL process. But it is not clear why this should be the case. This unreliably assumes that the feature representations of voxels and their neighborhoods prior to VCL are indeed correct, which may not be the case. In fact, the purpose of complementary label VCL methods is to address the potential incorrectness in the feature representations in the first place. It may as well be the case that the initial feature representations (prior to VCL) were incorrect, and that their neighborhood relationships should not be maintained during the VCL training process. The paper needs to present data on this assumption. E.g., how often does the initial neighborhood relationships of unreliable pseudo labeled voxels align with ground truth labels on labeled data?
3.	A second concern with the paper’s main contribution is that even if assuming that the voxel representations prior to VCL are reliable, it may be the case that voxels that are neighbors in the feature space may in fact happen to be from different organs with different labels. I.e., CT image voxels from different organs may have ended up being neighbors in the feature space. The current method of the paper will then incorrectly pull them together and assign the same label, whereas they should be assigned different labels. It is not clear how the method accounts for the above two scenarios.
4.	The paper doesn’t clearly describe or refer to prior work specifically on complementary label based VCL methods for semi-supervised segmentation. It needs to better describe what methods exist currently, their performance, merits and drawbacks, and present the claimed novelty of the paper in this prior context. a) Please include a paragraph in the related work section summarizing key complementary label VCL methods for segmentation; b) create a table compare features and performance of existing complementary label VCL methods for segmentation to the proposed approach.
5.	The presented method improves over existing method only by a small margin. While this is not the reason for the decision recommendation, it is necessary to evaluate statistical significance and confidence intervals of the results.

**Questions:**

Questions, clarification and corrections that need to be addressed in the paper:

1.	In the introduction, the sentence “Experiments demonstrate that VCL based on complementary labels helps to learn good voxel representations.” is not substantiated. There are no references. This is where it is important for the paper to clarify what already exists with complementary label VCL methods and present their contribution in this context.

2.	In the introduction, the sentence “Despite the successes of this technique, we have identified a hidden drawback. This approach can disrupt the relationships among some unreliable voxels, and these relationships are helpful when learning features.” needs to be substantiated well with data. This is missing and is a major drawback currently.

3.	How are the cases where the nearest neighbors that still happen to be far distance wise in the feature space handled? Would the method still force them to belong to same category? Just relying on the neighbors irrespective of how far they are can lead the algorithm astray. Is there a distance threshold in the feature space that is used? If not, how would the differences in distances of neighbors across voxels be handled? Please explain the rationale and discuss potential limitations. Consider adding an ablation study examining the impact of different distance thresholds.

4.	This statement, “This approach helps maintain useful semantic relationships among unreliable voxels while still enjoying the advantages of contrastive learning.” needs to be substantiated well with reasoning and data. In fact, using a similar idea to impose this kind of constraint on physically closer voxels in the image (rather than in feature space) seems more intuitive. Neighbors typically belong to the same category, except at the boundaries. But the paper imposes the constraint in the feature space, and it is not clear why this should always be satisfied in unreliable pseudo labeled voxels.

5.	“Then, after DVCL, voxel A is pulled to the same cluster of B, while voxel C is pushed
away from it.” => But in this process, C also happens to be pushed closer to 2, which is the incorrect category. How are these kinds of scenarios handled?

6.	In equation (4), perhaps the expression for W_A is incorrectly swapped between unreliable and reliable pseudo labels? I.e., M_r and M_u? Please check.

7.	In section 3.3. sentence “The negative candidates are selected from labeled voxels.” Why are reliable pseudo labeled voxels not used for negative candidates?

8.	Why are the distributions in equation (14) denoted as normal distributions?

9.	Regd. the method for distance aware contrastive learning, why are likelihoods of class relationships used instead of just minimizing (bringing closer) or maximizing (moving them farther) distances in the feature space? The motivation for the presented method is not clearly explained. Also, using class relationships doesn’t give leeway for the features to belong to nearby but different classes, which minimizing distances may.

10.	This sentence: “where ||pTm − pn||2 represents the l2-norm between the query feature and the features in the close” seems incorrect since pTm and pn are probabilities and cannot be L2 norm between features. Please clarify.

11.	The methodology using bounds is not clearly explained since the final equation (17) still seems to depend on Ur. A better description of the problem that is being solved here needs and how it is solved is needed.

12.	 In this sentence “In Table 4, we compare DVCL with existing nearest neighbor VCL methods (Wu et al., 2023; Dwibedi et al., 2021; Chongjian et al.) and find that DVCL exhibits superior performance. This improvement can be attributed to two main factors. First, for unreliable voxels, DVCL not only considers..”. Why are these references not included earlier in the introduction and related sections? These are important to place the contributions of the paper in prior research context. Please discuss them there as well.


Things to improve the paper that did not impact the score

1.	Reference Wang et al., 2022a seems incorrect since it does not even seem to talk about complementary label contrastive learning.

2.	In figure 2: Visual difference b/w nearest neighbors (solid lines) and farthest outsiders (dashed lines) doesn’t come out well. Please use color coding or other ways that make they easy to distinguish visually.

3.	In framework overview under methods, the paper needs to describe and motivate the CPS framework they use. Why was this framework chosen and not others? The paper doesn’t motivate the choice well.

4.	In Table 2, Ablation experiments on component aspect: (1) EBL; (2) DVCL, it appears including either EBL or dvcl already takes the result closer to when including both as compared to the baseline. A discussion on this would be good to add to the paper.

---

> ### Author Response · Authors · 2024-11-26
> **Response to Reviewer qnby (1/7)**
>
> ### Thank you for your detailed and constructive comments! Below is our response to your comments.
> ___
> ### First, we give our response to the weaknesses you mentioned:
> > #### **W1:** The paper does not clearly demonstrate the existence of the problem they are trying to solve, except for the cartoon illustration. It is not clear how often the claimed problem of semantically (dis)similar voxel features being (pulled)pushed (closer)away while using complementary label based VCL happens and by what degree, and how much does that affect the output. The paper needs to better illustrate the significance of the problem with real data. E.g., ...
>
> #### **Response:** We sincerely appreciate the constructive suggestions provided by the reviewers. Following your suggestion, **we demonstrate the existence of the problem through feature visualization of real data and statistical analysis of the inter-class distances of unreliable voxels (for details, please refer to Appendix K, lines 1214-1227).** The experiments are conducted on the test set using the FLARE 2022 dataset, trained under a 10% labeled data setting. The visualization results show that Complementary Label VCL only considers the mutual exclusion between unreliable voxels and class prototypes, while neglecting the relationships among unreliable voxels. This leads to random exclusion, which disrupts the original semantic relationships between unreliable voxels. For example, as shown in Fig.9(a)(top left) of Appendix.K in the new version, after applying VCL, node4 and node12 may be confused, but node4 and node6 should be distinguishable. However, with Complementary Label VCL, as shown in Fig.9(a)(top right)of Appendix.K, node4 and node6 are unintentionally drawn closer together.  **We further demonstrate this change by calculating the inter-class relationships among unreliable voxels. Additionally, we compare the performance of the Complementary Label VCL method (including U$^2$PL, BaCon, and CCL) with the method we propose across four datasets, as shown in Tables 1, 5, 6, and 7.** The results indicate that our method shows performance improvements.
>
> > #### **W2:** A primary concern is the reasoning/motivation of the paper’s main contribution, which is to retain the neighborhood relationships of voxel features in the feature space as determined prior to applying VCL. The paper claims that these relationships should be maintained through the VCL process. But it is not clear why this should be the case. This unreliably assumes that the feature representations of voxels and their neighborhoods prior to VCL are indeed correct, which may not be the case. In fact, the purpose of complementary label VCL methods is to address the potential incorrectness in the feature representations in the first place. It may as well be the case that the initial feature representations (prior to VCL) were incorrect, and that their neighborhood relationships should not be maintained during the VCL training process. The paper needs to present data on this assumption. E.g.,  ...
>
> #### **Response:** We would like to provide further clarification: **in the feature space after applying VCL, it is essential to preserve the neighborhood relationships of voxel features.** This is because the core advantage of VCL [15][16] lies in maximizing the similarity between samples of the same class while minimizing the similarity between samples of different classes, thereby enhancing the discriminative power of feature representations and promoting the formation of a well-clustered structure in the feature space. We validated this point on the FLARE 2022 dataset used in our experiments through feature visualization. For example, as shown in Fig.(a)(top left)of Appendix.K in the new version, after applying VCL, the reliable voxels (those in classes 2 and 13) are clearly separated into two distinct clusters. Conversely, some voxels (those in classes 4, 12, 5, and 6) are classified as unreliable due to their high feature similarity and susceptibility to confusion. **However, these unreliable voxels still adhere to the principle that "features that are close or distant in the feature space should produce consistent or inconsistent predictions," meaning their neighborhood relationships are still approximately preserved.**
> #### In contrast, the Complementary Label VCL method attempts to optimize by pushing unreliable voxels away from class prototypes they are unlikely to belong to. However, this approach fails to fully consider the neighborhood relationships among unreliable voxels, and may even disrupt these relationships(for details, please refer to Appendix K, lines 1223-1227). **Therefore, although the Complementary Label VCL method is built upon VCL and aims to fully leverage unreliable voxels, its neglect of the neighborhood relationships between unreliable voxels may limit its ability to achieve the intended effect. This highlights a key improvement in the design of our method.**
> ___
> ### Part (1/7)

---

> > ### Comment · Reviewer_qnby · 2024-12-03
> >
> > Weakness 1 (W1): The authors have included some real data examples to illustrate the problem that is being claimed to be addressed by the paper. But the significance is still a bit unclear. E.g., how often is this problem seen in real data? This can be quantified by measuring how often the top k nearest neighbors of a voxel feature do not all have the same label for different k. The ground truth labels can be used for this.
> >
> > Weakness 2 (W2): The author’s comments do not exactly address the raised key concern here. The assumption that is being made in the paper is that the neighborhood relationships of unreliable voxel features should be maintained during VCL. But that is exactly the kind of confirmation bias that the paper is claiming to address in the first place. Can the authors answer exactly why the initial features of the unreliable voxels should be trusted at all in the first place? These initial features will change during VCL and complementary VCL processes to some other value, and the purpose of complementary VCL is in fact to influence the change in the right direction using the more reliable complementary label relationships. These feature changes will effect changes in neighborhood relationships. So, it is still not clear why the initial feature neighborhood relationships should be trusted or maintained during the process?
> >
> > Additional comment: Based on additional reading of related work, I would like to correct my previous incorrect comment that the entropy based unreliable voxel determination presented in this paper is original, which in fact was already described in the complementary VCL paper by Wang et. Al references here.

---

> ### Author Response · Authors · 2024-11-26
> **Response to Reviewer qnby (2/7)**
>
> > #### **W3:** A second concern with the paper’s main contribution is that even if assuming that the voxel representations prior to VCL are reliable, it may be the case that voxels that are neighbors in the feature space may in fact happen to be from different organs with different labels. I.e., CT image voxels from different organs may have ended up being neighbors in the feature space. The current method of the paper will then incorrectly pull them together and assign the same label, whereas they should be assigned different labels. It is not clear how the method accounts for the above two scenarios.
>
> #### **Response:** Although voxels from different organs might become neighbors due to similar features in the feature space, by carefully choosing the value for the number of neighbors, $K$, we can effectively reduce such misclassifications and ensure that the selected neighbors are as likely as possible to belong to the same class. For ablation studies on $K$, please refer to Tab.3 in the new revision. Inspired by [12] [13], our method selects the $K$ most similar voxels $(r_{n}| top-K(cos\left(r_m,r_{n}\right),\forall r_{n} \in U_R))$ as neighbors based on cosine similarity. We control the number of neighbors by adjusting the value of $K$, thereby selecting voxels that are more likely to belong to the same class. Additionally, Following Dwibedi et al.[14], we use queues as candidate sets $\mathcal{C}_m$, where each element represents a feature. During network training with sample batches, we store features in $\mathcal{C}_m$, updating the candidate sets using a first-in-first-out strategy.
>
> > #### **W4:** The paper doesn’t clearly describe or refer to prior work specifically on complementary label based VCL methods for semi-supervised segmentation. It needs to better describe what methods exist currently, their performance, merits and drawbacks, and present the claimed novelty of the paper in this prior context. a) Please include a paragraph in the related work section summarizing key complementary label VCL methods for segmentation; b) create a table compare features and performance of existing complementary label VCL methods for segmentation to the proposed approach.
>
> #### **Response:** Thank you for your helpful suggestion. We have refined the descriptive details of various complementary label VCL methods for semi-supervised segmentation tasks in Section 2(for details, please refer to lines 158-159). Additionally, we compared the performance of complementary label VCL methods (including U$^2$PL, BaCon, and CCL) with our proposed approach across four datasets, as presented in Tab.1, Tab.5, Tab.6, and Tab.7. The results demonstrate that our method consistently outperforms complementary label VCL methods in terms of performance.
>
> > #### **W5:** The presented method improves over existing method only by a small margin. While this is not the reason for the decision recommendation, it is necessary to evaluate statistical significance and confidence intervals of the results.
>
> #### **Response:** Thank you for your helpful suggestion. To assess the stability of the results, we used box plots as a substitute for these statistical measures. Our method achieved the best and most stable segmentation performance, with the least fluctuation in the results. For further details, please refer to Fig.10 in Appendix L of the new version. Since validating statistical significance and confidence intervals requires a large number of repeated experiments, we conducted only 3 repetitions for each method. Instead, we used box plots to evaluate and demonstrate the stability of the results.
>
> ___
> ### Part (2/7)

---

> > ### Comment · Reviewer_qnby · 2024-12-03
> >
> > Weakness 3 (W3): The paper should acknowledge and discuss this as a limitation. If this is not a problem, the paper should demonstrate that by quantifying the number of times the voxels may potentially have different ground truth labels but close by features for different values of k. A future iteration of the method can account for the potential similarity in features across different labeled regions.
> >
> > Weakness 5 (W5): The authors can use methods like Monte Carlo dropout to compute uncertainties over segmentation predictions.

---

> ### Author Response · Authors · 2024-11-26
> **Response to Reviewer qnby (3/7)**
>
> ### Next, we give our response to your questions:
>
> > #### **Q1:** In the introduction, the sentence “Experiments demonstrate that VCL based on complementary labels helps to learn good voxel representations.” is not substantiated. There are no references. This is where it is important for the paper to clarify what already exists with complementary label VCL methods and present their contribution in this context.
>
> #### **Response:** We sincerely apologize for the confusion caused to the reviewers due to the missing citation. We have completed the references here(lines 93-94). In fact, our argument here is based on the references mentioned in line 86–87 of the manuscript, which primarily focus on complementary label VCL methods and their variants. Following your suggestion, we have summarized the contributions of these methods individually. Please refer to the new revision for details(lines 158-159).
>
> > #### **Q2:** In the introduction, the sentence “Despite the successes of this technique, we have identified a hidden drawback. This approach can disrupt the relationships among some unreliable voxels, and these relationships are helpful when learning features.” needs to be substantiated well with data. This is missing and is a major drawback currently.
>
> #### **Response:** Following your suggestion, we support our argument through feature visualization of real data and statistical analysis of the inter-class distances of unreliable voxels (for details, please refer to Appendix K, lines 1214-1227). The experiments are conducted on the test set using the FLARE 2022 dataset, trained under a 10% labeled data setting. **The visualization results show that Complementary Label VCL only considers the mutual exclusion between unreliable voxels and class prototypes, while neglecting the relationships among unreliable voxels. This leads to random exclusion, which disrupts the original semantic relationships between unreliable voxels.** For example, as shown in Fig.9(a)(top left) of Appendix.K in the new version, after applying VCL, node4 and node12 may be confused, but node4 and node6 should be distinguishable. However, with Complementary Label VCL, as shown in Fig.9(a)(top right)of Appendix.K, node4 and node6 are unintentionally drawn closer together. We further demonstrate this change by calculating the inter-class relationships among unreliable voxels.
>
> > #### **Q3:** How are the cases where the nearest neighbors that still happen to be far distance wise in the feature space handled? Would the method still force them to belong to same category? Just relying on the neighbors irrespective of how far they are can lead the algorithm astray. Is there a distance threshold in the feature space that is used? If not, how would the differences in distances of neighbors across voxels be handled? Please explain the rationale and discuss potential limitations. Consider adding an ablation study examining the impact of different distance thresholds.
>
> #### **Response:** **Instead of using a distance threshold to account for variations in the distances between neighbors, inspired by [12] [13], our method selects the $K$ most similar voxels $(r_{n}| top-K(cos\left(r_m,r_{n}\right),\forall r_{n} \in U_R))$ as neighbors based on cosine similarity.** After selecting the $K$ neighbors with the highest cosine similarity, these neighbors are uniformly regarded as belonging to the same category as the query voxel. We adjust the number of neighbors by varying the value of $K$. When $K$ is small, the query voxel has fewer neighbors. Although these neighbors have high similarity, they may only contain limited semantic information. Conversely, when $K$ is large, the number of neighbors increases, potentially introducing misclassified samples with high similarity but belonging to different categories. For ablation studies on $K$, please refer to Tab.3 in the new revision.
> ___
> ### Part (3/7)

---

> > ### Comment · Reviewer_qnby · 2024-12-03
> >
> > Question 3 (Q3): The authors need to discuss the limitation and impact of the variability in what near means for different voxels. For some voxels, the closest neighbors may still be far away compared to some other voxels whose neighbors may be much closer. It is not clear how using the same k for all voxels affects the results.

---

> ### Author Response · Authors · 2024-11-26
> **Response to Reviewer qnby (4/7)**
>
> >#### **Q4:** This statement, “This approach helps maintain useful semantic relationships among unreliable voxels while still enjoying the advantages of contrastive learning.” needs to be substantiated well with reasoning and data. In fact, using a similar idea to impose this kind of constraint on physically closer voxels in the image (rather than in feature space) seems more intuitive. Neighbors typically belong to the same category, except at the boundaries. But the paper imposes the constraint in the feature space, and it is not clear why this should always be satisfied in unreliable pseudo labeled voxels.
>
> #### **Response to "This statement...needs to be substantiated well with reasoning and data.":**
> #### Thank you for your helpful suggestion. To validate the statement, “This approach helps...”, we conducted t-SNE feature visualization of real data and statistical analysis of the inter-class distances of unreliable voxels (for details,please refer to Appendix K,Fig.9). The experiments are conducted on the test set using the FLARE 2022 dataset, trained under a 10% labeled data setting. **The advantage of applying VCL lies in maximizing the similarity between samples of the same class while minimizing the similarity between samples of different classes, thereby enhancing the discriminative power of feature representations and promoting the formation of a well-clustered structure in the feature space[15][16].** Although unreliable voxels may suffer from classification confusion, some semantic relationships in the feature space can be effectively leveraged, as shown in Fig.9(a)(top left). However, Complementary Label VCL only considers the mutual exclusion between unreliable voxels and class prototypes, while neglecting the relationships among unreliable voxels. **This leads to random exclusion, which disrupts the original semantic relationships between unreliable voxels**(for details,please refer to Appendix K,lines 1214-1227). **To address the above issues, DVCL introduces the concepts of "neighbors" and "strangers," which maintain useful semantic relationships among unreliable voxels while still enjoying the advantages of contrastive learning.**(for details,please refer to Appendix K,lines 1228-1233).
>
> #### **Response to "In fact, using a similar idea to...except at the boundaries":**
> #### We acknowledge the reviewer’s point that adjacent elements at the image level are more likely to belong to the same category, except at the boundaries. **However, since the core objective of our design is to focus on unreliable voxels, which are typically located in the boundary regions where multiple organs overlap in multi-organ tasks, they may not conform to this assumption.** In multi-organ tasks, directly using image-level neighbors to handle unreliable voxels may have limited effectiveness.
>
> #### **The reason we choose to search for "neighbors" and "outsiders" in the feature space is that, although voxel positions of different organs may overlap in image space, the model can effectively distinguish them in the feature space due to differences in local context.** For example, convolutional layers can learn subtle differences between organs, such as variations in texture, edges, or density, allowing these voxel to be separated in the feature space. Therefore, we enhance the model’s learning ability for unreliable voxels by selecting "neighbors" and "outsiders" based on similarity in the feature space. Additionally, following the reviewer’s suggestion, we conducted further experiments on selecting neighbors for unreliable voxels at the image level, and the results are shown in Tab.a.
>
> #### **Table a: Ablation studies of the select strategy of the "neighbors" and "outsiders". $N$ represents the number of "neighbors" in the image space based on Euclidean distance, while $O$ denotes the number of "outsiders" in the same space. Best results are boldfaced.**
> |Methods|Mean Dice|Mean Jaccard|
> |-|-|-|
> |**Ours**|**84.11 $\pm$ 0.08**|**73.49$\pm$0.13**|
> |**$N=5, O=5$**|82.22 $\pm$ 0.15|71.11 $\pm$ 0.21|
> |**$N=5, O=10$**|82.48 $\pm$ 0.32|70.77 $\pm$ 0.41|
> |**$N=10, O=10$**|81.65 $\pm$ 0.13|70.07 $\pm$ 0.16|
>
> #### **Response to "But the paper...satisfied in unreliable pseudo labeled voxels.":**
> #### **We argue that all voxels, even unreliable ones, should ideally satisfy this property.** This phenomenon can be observed in Fig.9(a)(top left) of Appendix K in the new version. Although unreliable voxels are prone to confusion, they still intuitively follow this pattern in the feature space. Deep learning models optimize the loss function using contrastive learning strategies that minimize intra-class variance and maximize inter-class variance, causing feature vectors of the same class to cluster and those of different classes to separate, enabling effective classification in high-dimensional feature space.
> ___
> ### Part (4/7)

---

> ### Author Response · Authors · 2024-11-26
> **Response to Reviewer qnby (5/7)**
>
> > #### **Q5:** “Then, after DVCL, voxel A is pulled to the same cluster of B, while voxel C is pushed away from it.” => But in this process, C also happens to be pushed closer to 2, which is the incorrect category. How are these kinds of scenarios handled?
>
> #### **Response:** We apologize for the lack of precision in our illustration. The toy example in Fig.1 is intended to illustrate our motivation and the workings of the algorithm. Due to some oversight on our part, there was an error in the plotting. We demonstrate this phenomenon through feature visualization of real data (for details, please refer to Appendix K, Figure 9(a)). As shown in Fig.9(a)(bottom right), although our method increases the distance between unreliable voxels and their outsiders, these unreliable voxels are not pushed toward category 2, which they do not belong to.
>
> > #### **Q6:** In equation (4), perhaps the expression for W_A is incorrectly swapped between unreliable and reliable pseudo labels? I.e., M_r and M_u? Please check.
>
> #### **Response:** We apologize for the mix-up between M_r and M_u. Following your suggestion, we have made it clear in equation (4) in the new revision.
>
> > #### **Q7:** In section 3.3. sentence “The negative candidates are selected from labeled voxels.” Why are reliable pseudo labeled voxels not used for negative candidates?
>
> #### **Response:** For reliable pseudo-labeled voxels, **due to the absence of ground-truth labels,** we can only determine that a voxel is highly likely to belong to a certain category without knowing its exact category. Consequently, we do not include unlabeled data in the discussion of negative samples. We treat the misclassified voxels of the model as negative samples for the corresponding category. For instance, if a voxel actually belongs to category A but is misclassified as category B, we aim for this voxel to be far from category B in the feature space, thereby considering it as a negative sample for category B.
>
> > #### **Q8:** Why are the distributions in equation (14) denoted as normal distributions?
>
> #### **Response:** We are sorry for the misunderstanding caused by our description. **$\mathcal{N}(C_m\mid\theta_S)$ represents a joint probability rather than a normal distribution.** It calculates $O_{m,n}$ for each voxel $r_n$ in the set $C_{m}$ and takes their product, thereby reflecting the overall similarity score of $C_{m}$. Since $C_{m}$ is a neighborhood set, we aim to continually increase its similarity score, which will result in a gradual reduction of the negative log-likelihood value of $\mathcal{N}(C_m\mid\theta_S)$.
>
> > #### **Q9:** Regd. the method for distance aware contrastive learning, why are likelihoods of class relationships used instead of just minimizing (bringing closer) or maximizing (moving them farther) distances in the feature space? The motivation for the presented method is not clearly explained. Also, using class relationships doesn’t give leeway for the features to belong to nearby but different classes, which minimizing distances may.
>
> #### **Response to "Regd. the method for ...in the feature space ":**
> #### **We chose to use the likelihood of class relationships, rather than directly pulling together or pushing apart voxel features as in traditional contrastive learning, because we argue that introducing contrastive learning between two feature sets goes beyond the limitations of traditional positive-negative sample comparisons. This approach enables us to more effectively capture distributional differences between groups, thereby providing a clearer understanding of the semantic structural relationships between samples.** We adopted negative log-likelihood (NLL) as the loss function, inspired by self-supervised learning [5]. To further validate the effectiveness of our strategy, we conducted comparative experiments with different contrastive learning strategies (including InfoNCE, SimCLR, and KL divergence). The results demonstrate that our strategy achieves the best performance, as shown in Tab. b.
> #### **Table b: Ablation studies of the contrastive learning strategy. Best results are boldfaced.**
> |**Method**|**Term**|**Mean Dice**|**Mean Jaccard**|
> |-|-|-|-|
> |InfoNCE [6]| $-\log\frac{\exp\left(r_{m}\cdot r_{n}/\tau\right)}{\exp\left(r_{m}\cdot r_{n}/\tau\right)+\sum_{r_{n}\in D_{m}}\exp\left(r_{m}\cdot r_{k}/\tau\right)}$|83.05 ± 0.28|73.21 ± 0.31|
> |SimCLR [7]| $-\log\frac{\exp{(r_{m}\cdot r_{n}/\tau)}}{\sum_{r_{n}\in C_{m}}\exp{(r_{m}\cdot r_{n}/\tau)}} + \log\frac{\exp{(r_{m}\cdot r_{k}/\tau)}}{\sum_{r_{k}\in D_{m}}\exp{(r_{m}\cdot r_{k}/\tau)}}$|82.77 ± 0.15|72.45 ± 0.33|
> |KL [8]| $-\frac{1}{\mid C_m \mid}\sum_{r_n\in C_{m}} KL(r_{m}\mid\mid r_{n})+\lambda\frac{1}{D_{m}}\sum_{r_{k}\in D_{m}} KL(r_{m}\mid\mid r_{k})$|82.47 ± 0.31|72.01 ± 0.40|
> |Log-likelihood [5]|$-\log \frac{\mathcal{N}(C_m \mid r_m, \theta_S)}{\mathcal{N}(D_m \mid r_m, \theta_S)}$ |**84.11 ± 0.08**|**73.49 ± 0.13**|
> ___
> ### To be continued
> ### Part (5/7)

---

> > ### Comment · Reviewer_qnby · 2024-12-03
> >
> > Question 5 (Q5): Please correct the cartoon figure. This gives an incorrect idea to the reader.
> >
> > Question 7 (Q7): It is still not clear why reliable pseudo labels cannot be used for negatives during contrastive learning, similar to how it is done in VCL methods.
> >
> > Question 8 (Q8): Please make the appropriate notational correction in the paper.

---

> ### Author Response · Authors · 2024-11-26
> **Response to Reviewer qnby (6/7)**
>
> #### **Response to "Also, using class relationships doesn’t give leeway ...":**
> #### In highly similar regions, the representations in the feature space may overlap, making it difficult for the model to distinguish boundaries accurately. To address this issue, we control the number of neighbors, selecting samples with the most similar semantic information as neighbors. For outsiders, the selection is more relaxed, but to reduce computational cost and ensure loss stability, we choose a number of outsiders only slightly larger than the number of neighbors.
>
> > #### **Q10:** This sentence: “where ||pTm − pn||2 represents the l2-norm between the query feature and the features in the close” seems incorrect since pTm and pn are probabilities and cannot be L2 norm between features. Please clarify.
>
> #### **Response:** Thank you for your helpful suggestions! We have rectified the errors in the formula and revised Equation (16) to enhance its clarity. The details of our proof are provided in Appendix J in the new revision.
>
> > #### **Q11:** The methodology using bounds is not clearly explained since the final equation (17) still seems to depend on Ur. A better description of the problem that is being solved here needs and how it is solved is needed.
>
> #### **Response:** We apologize for any inconvenience caused by our lack of clarity in the explanation. Specifically, we optimize $\psi(C_m,D_m)$ into $\overline{\psi}(C_m,D_m)$ and reduce the computation of $\sum_{r_q \in U_R}\frac{p_m^T p_q}{\mid U_R \mid}$ after optimization. 	Using Jensen Inequality $\log\left(\mathbb{E}[X]\right)\geq\mathbb{E}[\log(X)]$, we have
> $$
> \psi(C_m,D_m)=-\sum_{r_{n} \in C_{m}}p_{m}^{T}p_{n}+\sum_{r_{k} \in D_{m}}p_{m}^{T}p_{k}+(|C_{m}|-|D_{m}|)\log(\sum_{r_{q}\in U_{R}}e^{p_{m}^{T}p_{q}})\leq-\sum_{r_n \in C_m}p_m^Tp_n+\sum_{r_k \in D_m}p_m^Tp_k+(|C_m|-|D_m|)(\log|U_R|+\sum_{r_q\in U_R}\frac{p_m^Tp_q}{|U_R|})
> $$
> #### The expression $p_{m}^{T}p_{q}$ represents the inner product between the probability vector of $r_m$ and the probability vectors of all $r_n$ in $U_{R}$. Since $r_m$ and $r_n$ are not identical, it follows that $p_{m}^{T}p_{q} < 1$, and thus$\sum_{r_q \in U_R}\frac{p_m^Tp_q}{|U_R|} < 1$. We have
> $$
> -\sum_{r_n \in C_m}p_m^Tp_n+\sum_{r_k \in D_m}p_m^Tp_k+(|C_m|-|D_m|)(\log|U_R|+\sum_{r_q\in U_R}\frac{p_m^Tp_q}{|U_R|})<-\sum_{r_n \in C_m}p_m^Tp_n+\sum_{r_k \in D_m}p_m^Tp_k+(|C_m|-|D_m|)(\log|U_R|+1)=\overline{\psi}(C_m,D_m)
> $$
> #### This reduces the time complexity from $O$($n^2$) to $O(1)$, making the strategy more efficient when applied to large-scale datasets.
>
> > #### **Q12:** In this sentence “In Table 4, we compare DVCL with existing nearest neighbor VCL methods (Wu et al., 2023; Dwibedi et al., 2021; Chongjian et al.) and find that DVCL exhibits superior performance. This improvement can be attributed to two main factors. First, for unreliable voxels, DVCL not only considers..”. Why are these references not included earlier in the introduction and related sections? These are important to place the contributions of the paper in prior research context. Please discuss them there as well.
>
> #### **Response:** Thank you for your helpful suggestion. We have discussed these self-supervised learning methods in section.2 (Related Work lines 189-193) in the new revision.
>
> ### Next, we give our response to the things to improve the paper that did not impact the score:
>
> > #### **Thing1:** Reference Wang et al., 2022a seems incorrect since it does not even seem to talk about complementary label contrastive learning.
>
> #### **Response:** We sincerely apologize for the errors caused by our oversight and have made the necessary corrections [1] in the new version (line 86).
>
> > #### **Thing2:** In figure 2: Visual difference b/w nearest neighbors (solid lines) and farthest outsiders (dashed lines) doesn’t come out well. Please use color coding or other ways that make they easy to distinguish visually.
>
> #### **Response:** Thank you for your suggestions! Following your suggestion, we have updated Fig.2 in the new revision to make it more clear.
>
> ___
> ### Part (6/7)

---

> ### Author Response · Authors · 2024-11-26
> **Response to Reviewer qnby (7/7)**
>
> > #### **Thing3:** In framework overview under methods, the paper needs to describe and motivate the CPS framework they use. Why was this framework chosen and not others? The paper doesn’t motivate the choice well.
>
> #### **Response:** **We chose CPS as our baseline because it combines two conventional methods in the current semi-supervised learning field (Consistency learning and self-training), outperforming the traditional mean teacher model, making it the benchmark we selected.** Semi-supervised segmentation methods mainly include Self-training [10] and Consistency learning [9]. Self-training is an offline process that involves three steps: training a model on labeled data, generating pseudo-labels for unlabeled data using the pre-trained model, and retraining the model with both ground truth and pseudo-labels. Consistency learning, on the other hand, is an online process that encourages the model to produce consistent outputs for different augmentations of the same input.
>
> > #### **Thing4:** In Table 2, Ablation experiments on component aspect: (1) EBL; (2) DVCL, it appears including either EBL or dvcl already takes the result closer to when including both as compared to the baseline. A discussion on this would be good to add to the paper.
>
> #### **Response:** Thank you for your suggestions! The ECPS in EBL effectively utilizes unreliable voxels at the image level, while DVCL handles unreliable voxels more effectively at the feature level. These strategies significantly improve the performance of organs with a large number of unreliable voxels, such as Gal, RAG, and LAG. The major performance improvement comes from these organs, and when combined, they achieve the best overall performance.
>
> ### References
> [1] Wang Y, Wang H, Shen Y, et al. Semi-supervised semantic segmentation using unreliable pseudo-labels[C]//Proceedings of the IEEE/CVF conference on computer vision and pattern recognition. 2022: 4248-4257.
>
> [2 ]Du Z, Jiang X, Wang P, et al. LION: Label Disambiguation for Semi-supervised Facial Expression Recognition with Progressive Negative Learning[C]//IJCAI. 2023: 699-707.
>
> [3]Feng Q, Xie L, Fang S, et al. BaCon: Boosting Imbalanced Semi-supervised Learning via Balanced Feature-Level Contrastive Learning[C]//Proceedings of the AAAI Conference on Artificial Intelligence. 2024, 38(11): 11970-11978.
>
> [4]Deng Q, Guo Y, Yang Z, et al. Boosting semi-supervised learning with Contrastive Complementary Labeling[J]. Neural Networks, 2024, 170: 417-426.
>
> [5]Wu Z, Xiong Y, Yu S X, et al. Unsupervised feature learning via non-parametric instance discrimination[C]//Proceedings of the IEEE conference on computer vision and pattern recognition. 2018: 3733-3742.
>
> [6]Oord A, Li Y, Vinyals O. Representation learning with contrastive predictive coding[J]. arXiv preprint arXiv:1807.03748, 2018.
>
> [7]Chen T, Kornblith S, Norouzi M, et al. A simple framework for contrastive learning of visual representations[C]//International conference on machine learning. PMLR, 2020: 1597-1607.
>
> [8]Cover T, Hart P. Nearest neighbor pattern classification[J]. IEEE transactions on information theory, 1967, 13(1): 21-27.
>
> [9] Tarvainen A, Valpola H. Mean teachers are better role models: Weight-averaged consistency targets improve semi-supervised deep learning results[J]. Advances in neural information processing systems, 2017, 30.
>
> [10] Amini M R, Feofanov V, Pauletto L, et al. Self-training: A survey[J]. arXiv preprint arXiv:2202.12040, 2022.
>
> [11] Chen X, Yuan Y, Zeng G, et al. Semi-supervised semantic segmentation with cross pseudo supervision[C]//Proceedings of the IEEE/CVF conference on computer vision and pattern recognition. 2021: 2613-2622.
>
> [12] Yang S, Wang Y, Van De Weijer J, et al. Generalized source-free domain adaptation[C]//Proceedings of the IEEE/CVF international conference on computer vision. 2021: 8978-8987.
>
> [13] Cover T, Hart P. Nearest neighbor pattern classification[J]. IEEE transactions on information theory, 1967, 13(1): 21-27.
> [14] Dwibedi D, Aytar Y, Tompson J, et al. With a little help from my friends: Nearest-neighbor contrastive learning of visual representations[C]//Proceedings of the IEEE/CVF International Conference on Computer Vision. 2021: 9588-9597.
>
> [15] Wang W, Zhou T, Yu F, et al. Exploring cross-image pixel contrast for semantic segmentation[C]//Proceedings of the IEEE/CVF international conference on computer vision. 2021: 7303-7313.
>
> [16] Alonso I, Sabater A, Ferstl D, et al. Semi-supervised semantic segmentation with pixel-level contrastive learning from a class-wise memory bank[C]//Proceedings of the IEEE/CVF international conference on computer vision. 2021: 8219-8228.
>
> ___
> ### Part (7/7)
>
> ### Thank you for reading this response. We would be happy to discuss any further questions!

---

> ### Author Response · Authors · 2024-12-04
> **Response to Reviewer qnby - Round 2 (1/2)**
>
> ### Thank you for your insightful feedback! Below is our response to your comments.
> ___
> > #### **Weakness 2 (W2):** The author’s comments do not exactly address the raised key concern here. The assumption that is being made in the paper is that the neighborhood relationships of unreliable voxel features should be maintained during VCL. ...  So, it is still not clear why the initial feature neighborhood relationships should be trusted or maintained during the process?
>
> #### **Response:**
>
> * ####  **Why can certain semantic relationships of unreliable voxels after VCL still be trusted?** We think that by applying VCL, even unreliable voxels are guided by the overall optimization objective, gradually aligning their feature distribution towards a more reasonable semantic structure in the feature space. This is primarily due to the core advantage of VCL [2][3], which enhances the discriminative ability of feature representations and fosters the formation of well-defined clustering structures. **While unreliable voxels may exhibit slightly weaker global discriminative properties, their local semantic information (e.g., neighborhood features or partial category feature aggregation) remains preserved, benefiting from VCL's optimization of feature clustering.** This is evident in Fig. 9(a) (top left) of Appendix K: even though some categories of unreliable voxels fail to form clear category boundaries, the overall trend of intra-class cohesion and inter-class separation is still maintained. This suggests that after applying VCL, unreliable voxels retain partially trustworthy semantic relationships in the feature space.
>
> * #### **Which feature relationships should be maintained?** Fig. 9(a) (top left) in Appendix K provides an illustrative example: there is confusion between categories 5 and 12, as well as between categories 12 and 6. However, certain specific categories of voxels, such as categories 5 and 6, exhibit significant differences and are unlikely to belong to the same category. Therefore, we think it is crucial to preserve certain potentially accurate feature neighborhood relationships of unreliable voxels, as this helps to more precisely reflect the true semantic associations. At the same time, we aim to maintain intra-class consistency, meaning that even as the feature space changes, unreliable voxels should still maintain a high similarity with other unreliable voxels that potentially belong to the same category, thereby ensuring intra-class consistency.
>
> * #### **Why does Complementary VCL disrupt these semantic relationships?** Complementary VCL is an outstanding work that has provided us with significant insights. It can be seen as an extension of VCL applied to labeled data and reliable voxels, further enhancing feature learning by introducing a complementary mechanism for unreliable voxels. We agree with the reviewers’ perspective that the goal of complementary VCL is to guide features toward the correct direction by leveraging more reliable complementary label relationships. However, the complementary mechanism in this method primarily relies on the mutual exclusion relationship between unreliable voxels and class prototypes. This mutual exclusion relationship only involves the association between a single unreliable voxel and a single class prototype, lacking additional constraints. **Such a design might disrupt the previously mentioned feature neighborhood relationships and fails to fully account for the intra-class consistency formed by VCL.** For example, in Fig. 9(a) (top right) in Appendix K, as mentioned above, categories 5 and 6 should be distinguishable. However, with Complementary Label VCL, categories 5 and 6 are unintentionally drawn closer together. At the same time, within categories 5 and 6, we observe a dispersion effect caused by the complementary mechanism. We observed that although these voxels may belong to the same category, they are dispersed in the feature space due to the mutual exclusion operation. This dispersion could even undermine the advantages brought by contrastive learning. **Therefore, avoiding disruption to the potentially correct neighborhood relationships of unreliable voxels is a key direction for our further optimization.**
> * #### Confirmation bias refers to the scenario where VCL fully trusts unreliable pseudo-labels, leading to noise due to the incorrect pseudo-labels, and ultimately learning incorrect knowledge. However, we do not choose to trust unreliable pseudo-labels. Instead, we aim to use these unreliable voxels more reasonably. We note that the neighborhood relationships between unreliable voxels might be correct and thus worth considering. For example, as shown in the top left of Fig. 9a, there is confusion between categories 5 and 12, and between categories 12 and 6. However, for categories 5 and 6, we can clearly determine that they do not belong to the same category.
> ___
> ### Part (1/2)

---

> ### Author Response · Authors · 2024-12-04
> **Response to Reviewer qnby - Round 2 (2/2)**
>
> > #### **W1:** The authors have included some real data examples to illustrate the problem that is being claimed to be addressed by the paper. But the significance is still a bit unclear. E.g., how often is this problem seen in real data? This can be quantified by measuring how often the top k nearest neighbors of a voxel feature do not all have the same label for different k. The ground truth labels can be used for this.
>
> #### **Response:** We evaluated the frequency of this issue by analyzing changes in inter-class and intra-class distances of unreliable voxels, as shown in Fig. 9(b) in Appendix K. Taking the first row of the confusion matrix VCL as an example, the off-diagonal entries represent the inter-class distances between category 4 and categories 12, 5, and 6, respectively. After applying VCL, the inter-class distance ranking for category 4 (from largest to smallest) should be 6>5>12. However, after applying complementary label VCL, the ranking changed to 5>6>12. By comparing the rankings before and after, we estimate the probability of semantic relationship alterations to be $\frac{2}{3} $. Extending this analysis to the inter-class distances of all four categories, the frequency of the issue occurring is  $\frac{9}{12} $, or 75%. This suggests that the probability of semantic relationship changes among unreliable voxel categories is approximately 75%. The diagonal region represents intra-class distance. By comparing the confusion matrices of VCL and Complementary Label VCL, it can be observed that the intra-class distance increases, which violates the principle of intra-class consistency. According to statistical analysis, the probability of intra-class consistency being disrupted is $\frac{3}{4} $.
> > #### **Additional comment:** Based on additional reading of related work, I would like to correct my previous incorrect comment that the entropy based unreliable voxel determination presented in this paper is original, which in fact was already described in the complementary VCL paper by Wang et. Al references here.
>
> #### **Response:** Thank you for your comments! We inspired by Wang et.[1] by using entropy as a criterion to distinguish between reliable and unreliable pseudo-labels, as explicitly mentioned and cited in line 243 of our paper. However, unlike Wang's method[1], we employed a mean-plus-standard strategy in the Entropy-based Selection Module to determine the threshold, eliminating the need to sort all voxels and search for the threshold based on percentiles. This reduces the time complexity from $O(logN)$ to $O(1)$, making the strategy more efficient when applied to large-scaledatasets.
>
> ### References
> [1] Wang Y, Wang H, Shen Y, et al. Semi-supervised semantic segmentation using unreliable pseudo-labels[C]//Proceedings of the IEEE/CVF conference on computer vision and pattern recognition. 2022: 4248-4257.
>
> [2] Cover T, Hart P. Nearest neighbor pattern classification[J]. IEEE transactions on information theory, 1967, 13(1): 21-27.
>
> [3] Dwibedi D, Aytar Y, Tompson J, et al. With a little help from my friends: Nearest-neighbor contrastive learning of visual representations[C]//Proceedings of the IEEE/CVF International Conference on Computer Vision. 2021: 9588-9597.
>
> ___
> ### Finally, we would like to extend our heartfelt gratitude to the reviewers. Your meticulous and thoughtful feedback has been instrumental in enhancing the quality of our paper while offering us invaluable insights. We deeply appreciate your dedication and guidance throughout the review process.
>
> ___
> ### Part (2/2)

---

> ### Author Response · Authors · 2024-12-04
> **Response to Reviewer qnby - Round 2**
>
> > #### **Weakness 3 (W3):** The paper should acknowledge and discuss this as a limitation. If this is not a problem, the paper should demonstrate that by quantifying the number of times the voxels may potentially have different ground truth labels but close by features for different values of k. A future iteration of the method can account for the potential similarity in features across different labeled regions.
>
> #### **Response:** We agree with the reviewer’s perspective on this issue and acknowledge that it is a limitation of our approach. We currently use cosine similarity to select $K$ features that are most similar, but it is unavoidable that voxels from different organs in CT images may become neighbors in the feature space. As a result, our method might incorrectly group these voxels together and assign them the same label. To mitigate this issue, we have conducted ablation studies on the value of $K$ and selected the parameter that yields the best performance. However, we are aware of the limitations of this approach. In future research, we plan to further optimize our algorithm and explore more intelligent mechanisms to select an appropriate $K$, thereby reducing the occurrence of such cases and enhancing the robustness of the method.
>
> > #### **Weakness 5 (W5):** The authors can use methods like Monte Carlo dropout to compute uncertainties over segmentation predictions.
>
> #### **Response:** Based on your suggestions, we have made every effort to assess the stability of our results through additional experiments. However, due to the tight schedule (with only one day remaining for rebuttal submission), we were unable to complete experiments related to Monte Carlo dropout. We are committed to conducting further experiments in future research to comprehensively evaluate our results. Once again, thank you for your valuable feedback!

---

> ### Author Response · Authors · 2024-12-04
> **Response to Reviewer qnby - Round 2**
>
> > #### **Question 3 (Q3):** The authors need to discuss the limitation and impact of the variability in what near means for different voxels. For some voxels, the closest neighbors may still be far away compared to some other voxels whose neighbors may be much closer. It is not clear how using the same k for all voxels affects the results.
>
> #### **Response:** Thank you for your valuable feedback. Our work is inspired by self-training methods, where neighbor selection is performed solely based on top-k cosine similarity. Regarding the reviewer's comment that two feature vectors may have strong directional similarity but still large actual distances (e.g., Euclidean distance), we acknowledge that we have not fully addressed this limitation. In future work, we will conduct more experiments to mitigate this issue, such as adjusting thresholds or introducing other distance metrics to handle cases where feature similarity is high but distances are large. We also recognize the limitations of our current strategy for selecting the value of k. In future research, we plan to further optimize the algorithm and explore more intelligent mechanisms for selecting an appropriate k to reduce the occurrence of this issue and improve the robustness of the method.

---

> ### Author Response · Authors · 2024-12-04
> **Response to Reviewer qnby - Round 2**
>
> > #### **Question 5 (Q5):** Please correct the cartoon figure. This gives an incorrect idea to the reader.
>
> #### **Response:** We appreciate your attention to detail and will make the necessary corrections to avoid any misleading interpretations.
>
> > #### **Question 6 (Q6):** It is still not clear why reliable pseudo labels cannot be used for negatives during contrastive learning, similar to how it is done in VCL methods.
>
> #### **Response:** Since we aim to move the voxels away from incorrectly classified categories in the feature space, we treat the misclassified voxels as negative samples for the corresponding category. Unlike traditional VCL methods, we do not treat voxels that are reliable for a particular category as negative samples for other categories.
>
> > #### **Question 8 (Q8):** Please make the appropriate notational correction in the paper.
>
> #### **Response:** Thank you for your valuable feedback. We will make the appropriate notational corrections in the paper as suggested.

---

### Official Review · Reviewer_3LE6 · 2024-11-04

**Soundness:** 3
**Presentation:** 3
**Contribution:** 3
**Rating:** 6
**Confidence:** 4

**Summary:**

This paper introduces a semi-supervised learning method called Distance-aware Voxel-wise Contrastive Learning (DVCL) to enhance multi-organ segmentation (MoS) in medical imaging. Traditional voxel-wise contrastive learning (VCL) aligns features based on initial pseudo-labels, which can cause confirmation bias and poor segmentation in complex anatomical contexts with large organ component variations. Distance awareness allows DVCL to cluster local voxels and push away distant ones, ensuring semantic consistency among uncertain pseudo-labels. An entropy-based selection module (ESM) classifies pseudo-labels as reliable or unreliable using a reweighting mechanism to maximize contrastive learning. In multi-organ segmentation tasks, DVCL outperforms state-of-the-art algorithms in four datasets.

**Strengths:**

1. The paper’s DVCL approach is innovative in the context of contrastive learning for segmentation, particularly in preserving neighborhood relationships within noisy pseudo-labels.
2. The method demonstrates robust quantitative improvements across four datasets, showing consistent performance gains over previous state-of-the-art models. The ablation studies and comparisons in Table 1 also strengthen the claim that DVCL’s modifications lead to meaningful improvements.
3. The paper explains complex concepts such as "neighbors" vs. "outsiders" in the feature space well, making the rationale behind DVCL accessible. The illustrations, especially Fig. 1(b), effectively contrast DVCL with existing methods, helping readers understand the unique aspects of the approach.

**Weaknesses:**

1.The entropy-based selection mechanism is crucial for DVCL's performance, however its selection threshold values (τ) are not thoroughly investigated across many datasets. While this may appear to be a minor drawback, a more transparent discussion or empirical study of threshold selection and any dataset-specific adjustments will improve clarity and allow for greater replication of results.

2.The DVCL's dependence on neighbors and outsiders to manage semantic ties between voxels presupposes that these associations remain stable during training iterations. However, for complicated anatomical structures, boundary voxels may present a barrier since their classification as "neighbors" or "outsiders" varies. An examination of how DVCL handles such boundary scenarios, particularly in places where organs overlap, would provide more insight into the method's semantic maintenance resilience.

**Questions:**

1.How are entropy thresholds for reliable and unreliable pseudo-labels (τ) determined? Are they dynamically modified between datasets or fixed based on previous experiments?

2.How do changing K and K' values affect DVCL's segmentation accuracy for anatomically complex or tiny organs? Further clarification on this topic may provide useful insights for enhancing DVCL across a variety of organ segmentation tasks.

---

> ### Author Response · Authors · 2024-11-26
> **Response to Reviewer 3LE6 (1/1)**
>
> ### Thank you for your detailed and constructive comments! Below is our response to your comments.
> ___
> ### First, we give our response to the weaknesses you mentioned:
>
> > #### **Weakness1**: The entropy-based selection mechanism is crucial for DVCL's performance, however its selection threshold values (τ) are not thoroughly investigated across many datasets. While this may appear to be a minor drawback, a more transparent discussion or empirical study of threshold selection and any dataset-specific adjustments will improve clarity and allow for greater replication of results.
>
> #### **Response:** We sincerely appreciate the reviewer's suggestion, and we conducted hyperparameter experiments on $\alpha$ across four datasets. The results are shown in in Tab. a. The threshold $\tau$ selection is primarily based on entropy, which serves as an indicator of the model's confidence in its predictions, reflecting the certainty of the model’s prediction for the current voxel.
> #### **Table a: Ablation studie of Variance factor α cross four datasets（Dice）. Best results are boldfaced.**
> | Datasets |$\alpha = 0.1$|$\alpha = 0.2$ |$\alpha = 0.3$|$\alpha = 0.4$|$\alpha = 0.5$|
> |-|-|-|--|-|-|
> | **FLARE (10% Labeled data)**|82.15 $\pm$ 0.90|83.39 $\pm$ 0.51|**84.11 $\pm$ 0.08**|83.82 $\pm$ 0.59| 82.57 $\pm$ 0.81|
> | **AMOS (10% Labeled data)**|70.22 $\pm$ 0.73 | 70.88 $\pm$ 0.36|**71.09 $\pm$ 0.51**|71.03 $\pm$ 0.44| 70.10 $\pm$ 0.12|
> | **BTCV (10% Labeled data)**|50.99 $\pm$ 0.86|51.10 $\pm$ 0.75|52.68 $\pm$ 0.39|52.65 $\pm$ 0.27|**52.88 $\pm$ 0.41**|
> | **MMWHS (10% Labeled data)**|85.07 $\pm$ 0.21|85.45 $\pm$ 0.42|**86.33 $\pm$ 0.28**| 86.01 $\pm$ 0.14|85.32 $\pm$ 0.33|
> > #### **Weakness2**: The DVCL's dependence on neighbors and outsiders to manage semantic ties between voxels presupposes that these associations remain stable during training iterations. However, for complicated anatomical structures, boundary voxels may present a barrier since their classification as "neighbors" or "outsiders" varies. An examination of how DVCL handles such boundary scenarios, particularly in places where organs overlap, would provide more insight into the method's semantic maintenance resilience.
>
> #### **Response:** We fully agree with the reviewer's point that voxels in boundary regions are often prone to being selected as incorrect neighbors. **Therefore, instead of selecting "neighbors" and "outsiders" based on spatial location, we focus on selecting voxels with similar features at the feature level, thereby reducing errors caused by boundary effects.** The reason we choose to search for "neighbors" and "outsiders" in the feature space is that, although voxel positions of different organs may overlap in image space, the model can effectively distinguish them in the feature space due to differences in local context. For example, convolutional layers can learn subtle differences between organs, such as variations in texture, edges, or density, allowing these voxel to be separated in the feature space. Therefore, we enhance the model’s learning ability for unreliable voxels by selecting "neighbors" and "outsiders" based on similarity in the feature space.
>
> ### Next, we give our response to your questions:
> >#### **Question1:** How are entropy thresholds for reliable and unreliable pseudo-labels ($\tau$) determined? Are they dynamically modified between datasets or fixed based on previous experiments?
>
> #### **Response:**
> #### **Our threshold $\tau$ dynamically changes across different datasets and even between training iterations to adapt to the model's confidence in its predictions.** The threshold $\tau$ is primarily determined by entropy, which reflects the model's confidence in its prediction for the current voxel. When the model is more confident in its prediction for a voxel, the entropy value is lower. Therefore, we use the mean plus standard deviation approach, allowing the threshold to dynamically adjust according to the model's confidence during training. Since the model's confidence varies across different datasets, the threshold also adjusts adaptively.
>
> > #### **Question2:** How do changing K and K' values affect DVCL's segmentation accuracy for anatomically complex or tiny organs? Further clarification on this topic may provide useful insights for enhancing DVCL across a variety of organ segmentation tasks.
>
> #### **Response:** $K$ and $K'$ represent the number of neighbors and outsiders, respectively, for an unreliable voxel. When $K$ and $K'$ are too small, it may prevent more unreliable voxels from being involved in training, making it difficult to mine the semantic relationships between unreliable voxels. On the other hand, when K is too large, in cases with complex or tiny organs, the neighbors may include many semantically similar but class-different voxels, which could have a negative impact.
> ___
> ### Thank you for reading this response. We would be happy to discuss any further questions!

---

### Official Review · Reviewer_thnD · 2024-11-04

**Soundness:** 2
**Presentation:** 2
**Contribution:** 2
**Rating:** 5
**Confidence:** 4

**Summary:**

The proposed method introduces a voxel-wise contrastive learning approach for semi-supervised medical image segmentation. This approach aims to move voxels with the same pseudo-labels closer to their respective prototypes while separating those with different labels. While this idea has been explored in previous research, the novelty of the proposed method appears limited. Additionally, the authors have not sufficiently clarified how their approach stands out from existing methods. Despite these limitations, the proposed method demonstrated strong performance on multi-organ segmentation tasks.

**Strengths:**

1. Voxel-wise contrastive learning is an interesting concept, even though it has been explored in previous works. I agree with the effectiveness of separating negative labels while bringing positive labels closer together.

2. The proposed work is well written and easy to understand.'

3. The proposed works aims to solve the meaningful problem of semi-supervised medical image segmentation task.

**Weaknesses:**

1. The concept of voxel-wise contrastive learning has been widely used in previous research, and the reviewer finds limited novelty in this approach.
2. The evaluation of this work is limited to the FLARE 2022 dataset. Additional experiments are recommended to better assess the generalization capability of the proposed approach.
3. Visualizations of the feature space or clustering results would be beneficial to observe the distribution and patterns of different prototypes.

**Questions:**

1. Despite differences in the tasks to which it is applied, what are the technical difference between the proposed method and existing approaches?
2. Organ segmentation is relatively simpler compared to other medical segmentation tasks, such as those involving pathology or vessel structures. Has the proposed method been shown to be effective on these more complex tasks?
3. The definition of a prototype is unclear, and it would be helpful for the authors to include a figure to illustrate its meaning and visualize the results in the feature space.

---

> ### Author Response · Authors · 2024-11-26
> **Response to Reviewer thnD (1/2)**
>
> ### Thank you for your detailed and constructive comments! Below is our response to your comments.
> ___
> ### First, we give our response to the weaknesses you mentioned:
> > #### **W1:** The concept of voxel-wise contrastive learning has been widely used in previous research, and the reviewer finds limited novelty in this approach.
>
> #### **Response:** **To fully leverage unreliable voxels in semi-supervised VCL, our method offers a significant advantage over Complementary Label VCL(lines 82-94) by maintaining useful semantic relationships among unreliable voxels while still enjoying the advantages of contrastive learning.** We intuitively validate this idea through feature visualization of real data and statistical analysis of the inter-class distances of unreliable voxels (for details, please refer to Appendix K, lines 1214-1227). Additionally, we compare the performance of the Complementary Label VCL method (including U$^2$PL, BaCon, and CCL) with the method we propose across four datasets, as shown in Tables 1, 5, 6, and 7. The results indicate that our method shows performance improvements.
>
> > #### **W2:** The evaluation of this work is limited to the FLARE 2022 dataset. Additional experiments are recommended to better assess the generalization capability of the proposed approach.
>
> #### **Response:**  We are sorry for the misunderstanding caused by our description. We would like to clarify that our experiments cover four datasets: FLARE 2022, AMOS, BTCV, and MMWHS. Due to space limitations, the experimental results on the AMOS, BTCV, and MMWHS datasets are provided in Appendix F, Appendix G, and Appendix H, respectively.
>
> > #### **W3:** Visualizations of the feature space or clustering results would be beneficial to observe the distribution and patterns of different prototypes.
>
> #### **Response:** We sincerely appreciate the constructive suggestions provided by the reviewers. Following your suggestion, **we support our idea through feature visualization of real data and statistical analysis of the inter-class distances of unreliable voxels (for details, please refer to Appendix K, lines 1214-1227).** We can further validate our motivation and idea by observing the changes in the feature distribution of unreliable voxels under different methods. The prototype can be approximated as the feature center.
>
> ___
> ### Part (1/2)

---

> ### Author Response · Authors · 2024-11-26
> **Response to Reviewer thnD (2/2)**
>
> ### Next, we give our response to your questions:
>
> > #### **Q1:** Despite differences in the tasks to which it is applied, what are the technical difference between the proposed method and existing approaches?
>
> #### **Response:** **Our method offers a significant advantage over Complementary Label VCL(lines 82-94) by maintaining useful semantic relationships among unreliable voxels while still enjoying the advantages of contrastive learning.** To validate this idea, we conducted a t-SNE visualization analysis, as shown in Fig.9(a) of Appendix.K in the new version. The experiments are conducted on the test set using the FLARE 2022 dataset, trained under a 10% labeled data setting. The advantage of applying VCL lies in maximizing the similarity between samples of the same class while minimizing the similarity between samples of different classes, thereby enhancing the discriminative power of feature representations and promoting the formation of a well-clustered structure in the feature space[15][16]. Although unreliable voxels may suffer from classification confusion, some semantic relationships in the feature space can be effectively leveraged. However, Complementary Label VCL only considers the mutual exclusion between unreliable voxels and class prototypes, while neglecting the relationships among unreliable voxels. This leads to random exclusion, which disrupts the original semantic relationships between unreliable voxels(for details, please refer to Appendix K, lines 1214-1227). To address the above issues, DVCL introduces the concepts of "neighbors" and "strangers," which maintain the semantic relationships between unreliable voxels(for details, please refer to Appendix K, lines 1227-1231). **This approach helps maintain useful semantic relationships among unreliable voxels while still enjoying the advantages of contrastive learning**
>
> > #### **Q2:** Organ segmentation is relatively simpler compared to other medical segmentation tasks, such as those involving pathology or vessel structures. Has the proposed method been shown to be effective on these more complex tasks?
>
> #### **Response:** Thank you for your comments. The challenge in multi-organ segmentation lies in improving the segmentation accuracy of difficult organs. Due to significant morphological differences between organs, such as the tubular structures of the esophagus and pancreas, these disparities increase segmentation difficulty and affect the performance of difficult organ segmentation. Through experiments, we found that in semi-supervised multi-organ segmentation tasks, difficult organs often contain a large number of unreliable voxels (lines 77-81). Therefore, our method significantly improves the segmentation accuracy of difficult organs by effectively utilizing the information from unreliable voxels (as shown in Fig.4). While pathology image segmentation and vessel image segmentation are typically single-class tasks, these tasks may face greater challenges in segmentation accuracy due to high detail requirements, structural complexity, and noise interference. The challenges may primarily lie in the data preprocessing and post-processing stages, rather than in the innovation of the algorithm itself. In the future, we will further validate the effectiveness of our method through experiments on pathology or vessel images.
>
> > #### **Q3:** The definition of a prototype is unclear, and it would be helpful for the authors to include a figure to illustrate its meaning and visualize the results in the feature space.
>
> #### **Response:** We are sorry for the misunderstanding caused by our description. A prototype refers to the most representative sample within a given class, serving as the typical representative that embodies the core features of that class. In the context of VCL, the prototype is often approximated as the feature center, which is the mean of the features within the same class. For visual details, please refer to Appendix K, Fig.9 (a).
>
> ___
> ### Part (2/2)
> ### Thank you for reading this response. We would be happy to discuss any further questions!

---

### Author Response · Authors · 2024-12-01
**General Response to All Reviewers**

#### Dear Reviewers,

#### We would like to thank all reviewers for their detailed and valuable comments. We have addressed each comment individually and made the necessary revisions. The manuscript has been updated accordingly, with the new changes highlighted in blue. The main updates can be summarized as follows:

* #### In Appendix K, we validate our research motivation and the effectiveness of the proposed method by presenting the t-SNE visualization of feature distributions on the FLARE 2022 real dataset, along with statistics on inter-class and intra-class distances, as shown in Fig.9 of Appendix K.
* #### In Appendix A, we provide a more detailed discussion of related work on the Complementary Label VCL method and Self-supervised Learning methods.
* #### We are making every effort to implement the additional results and analyses suggested by the reviewers. For example, to demonstrate the stability of the evaluation results, we have plotted Fig.10, and an ablation study on the contrastive learning strategy is provided in Tab.8 of Appendix M.
* #### We have corrected the error in Equation 16 and provided a more detailed theoretical proof of the formula in Appendix J.
* #### We have added the missing details throughout the paper and corrected several statements, such as the explanation of the ablation study on the hyperparameter $K$ in Tab.4.
* #### We modify the use of some symbols (e.g., Equation 4) to make the text more fluent and understandable.
* #### We refresh Fig.2 to make them more clear and comprehensible.

#### The valuable suggestions from the reviewers have greatly improved our paper. We are pleased to engage in further discussions to address any remaining concerns. We intend to incorporate all experiments conducted in the rebuttal phase into a revised version of the paper, and we are committed to making all the source code publicly available in the future.

#### Thanks again.

---

### Comment · Area_Chair_WzBB · 2024-12-03

Could the reviewers check whether the authors' response addresses your concerns?

---

### Meta-Review · Area_Chair_WzBB · 2024-12-16

**Metareview:**

This paper proposes a novel voxel-wise contrastive learning framework for semi-supervised multi-organ segmentation, which aims to maintain semantic relationships among voxels and also employ the advantages of voxel-wise contrastive learning via pulling neighbors and pushing outsiders in the feature space. This paper is well-written and easy to follow. Experimental results show promising improvement over compared baselines.

This paper received 1x marginally above the acceptance threshold, 1x marginally below the acceptance threshold, and 2x reject from reviewers. The major concerns raised by reviewers centered around (1) the motivation of the proposed DVCL, (2) requirements for additional analysis for DVCL, and (3) the interpretation of distinguishment between DVCL and existing works. While I appreciate the efforts the authors have made to reply to questions point-by-point, some key concerns are not thoroughly addressed. For instance, reviewer qnby noted that after the author's discussion, "It is still not clear why these initial features of the unreliable voxels and their neighborhood relationships should be trusted at all in the first place."

In sum, although comprehensive experiments are conducted and promising results are shown, the soundness of this work needs further improvement, including the motivation and rationale of the proposed methodology. Thus, the decision is to reject the paper this time.

**Additional Comments On Reviewer Discussion:**

Reviewers raised questions regarding the motivation, methodology design, and experimental analysis for this work. The authors thoroughly addressed most of the concerns by polishing the manuscript, adding external analysis, as well as detailed explanation point-by-point. However, some key aspects are not fully addressed, leaving space for further improvement of this work.

---

### Decision · Program_Chairs · 2025-01-22

Reject